# Metabolic mapping of the human solute carrier superfamily

Tabea Wiedmer [1,11], Shao Thing Teoh [1,11], Eirini Christodoulaki [1], Gernot Wolf [1], Chengzhe Tian [1], Vitaly Sedlyarov [1], Abigail Jarret[1], Philipp Leippe [1], Fabian Frommelt [1], Alvaro Ingles-Prieto [1], Sabrina Lindinger [1], Barbara M G Barbosa [1], Svenja Onstein [1], Christoph Klimek [1], Julio Garcia [1], Iciar Serrano [1], Daniela Reil[1], Diana Santacruz[2], Mary Piotrowski[3], Stephen Noell [3], Christoph Bueschl [1], Huanyu Li [4], Gamma Chi [4], Stefan Mereiter [5,6], Tiago Oliveira [5,6], Josef M Penninger [5,6,7,8], David B Sauer [4], Claire M Steppan [3], Coralie Viollet[2], Kristaps Klavins [1], J Thomas Hannich [1], Ulrich Goldmann [1] & Giulio Superti-Furga [1,9,10 ✉]

## Abstract

Solute carrier (SLC) transporters govern most of the chemical exchange across cellular membranes and are integral to metabolic regulation, which in turn is linked to cellular function and identity. Despite their key role, individual functions of the SLC superfamily members were not evaluated systematically. We determined the metabolic and transcriptional profiles upon SLC overexpression in knock-out or wild-type isogenic cell backgrounds for 378 SLCs and 441 SLCs, respectively. Targeted metabolomics provided a fingerprint of 189 intracellular metabolites, while transcriptomics offered insights into cellular programs modulated by SLC expression. Beyond the metabolic profiles of 102 SLCs directly related to their known substrates, we identified putative substrates or metabolic pathway connections for 71 SLCs without previously annotated bona fide substrates, including SLC45A4 as a new polyamine transporter. By comparing the molecular profiles, we identified functionally related SLC groups, including some with distinct impacts on osmolyte balancing and glycosylation. The assessment of functionally related human genes presented here may serve as a blueprint for other systematic studies and supports future investigations into the functional roles of SLCs.

**Keywords** Membrane Transporters; Metabolism; Metabolomics; Solute Carriers; Transcriptomics
**Subject Category** Metabolism

See also: F Frommelt et al, G Wolf et al & U Goldmann et al

## Introduction

Despite our rapidly increasing familiarity with the human genome and its variant forms, most large-scale studies assessing the genome's functional properties focus on the contributions of a few genes to specific functional readouts (Kampmann, 2020; Bock et al, 2022). There is a need to complement this approach with strategies allowing the functional comparison of many genes, measured with the same readout(s), in parallel, under controlled conditions. This strategy is particularly suitable for comparing gene families or genes encoding proteins with shared properties since it also generates data for genes that do not emerge in stochastic screens due to redundancy, mildness of phenotype, or specificity of the readout. A particularly worthy area of investigation concerns membrane transporters, responsible for the chemical exchange of biological systems with their environment and the repartition of solutes within biological systems. Not only cellular metabolism depends on the influx and efflux of chemical molecules, but the plethora of physiological processes is chemically integrated and thus also heavily dependent on transporters (Wu et al, 2011; Sahoo et al, 2014; Alam et al, 2023). As the characteristics of the transported molecule(s) are often unknown, and known substrates are chemically very heterogenous, broadly applicable assays have been hard to establish, resulting in many uncharacterized membrane transporters throughout the human genome (Dvorak et al, 2021; Saier et al, 2016; Fagerberg et al, 2010). Therefore, we consider it urgent to better understand this class of proteins (César-Razquin et al, 2015; Superti-Furga et al, 2020).

Of more than 1500 genes potentially encoding for transmembrane transporters, we opted to focus on the Solute Carrier (SLC) superfamily as it represents the largest, with ~450 members (Ye et al, 2014; Ferrada and Superti-Furga, 2022; Gyimesi and Hediger,

[1]CeMM Research Center for Molecular Medicine of the Austrian Academy of Sciences, 1090 Vienna, Austria. [2]Boehringer Ingelheim Pharma GmbH & Co. KG, 88400 Biberach, Germany. [3]Pfizer Worldwide Research and Development, Groton, CT 06340, USA. [4]Centre for Medicines Discovery, Nuffield Department of Medicine, University of Oxford, Oxford, UK. [5]Department of Laboratory Medicine, Medical University of Vienna, 1090 Vienna, Austria. [6]Institute of Molecular Biotechnology of the Austrian Academy of Sciences (IMBA), 1030 Vienna, Austria. [7]Helmholtz Centre for Infection Research, 38124 Braunschweig, Germany. [8]Department of Medical Genetics, Life Sciences Institute, University of British Columbia, V6T 1Z3 Vancouver, Canada. [9]Center for Physiology and Pharmacology, Medical University of Vienna, 1090 Vienna, Austria. [10]Fondazione Ri.MED, Palermo, Italy. [11]These authors contributed equally: Tabea Wiedmer, Shao Thing Teoh. ✉E-mail: gsuperti@cemm.oeaw.ac.at

2022) and is exceptionally asymmetrical in the distribution of knowledge among its members, with a few very well-studied and many still uncharacterized (Hediger et al, 2013; Oprea et al, 2018; César-Razquin et al, 2015). The SLC superfamily is divided into >65 families based on their sequence and functional similarity and its members transport a large variety of molecules such as amino acids, inorganic ions, carbohydrates, lipids, neurotransmitters, nucleosides/nucleotides, vitamins, peptides and xenobiotics (Fredriksson et al, 2008; Meixner et al, 2020). SLCs are localized to the plasma membrane or to different intracellular membranes (Meixner et al, 2020; Giacomini et al, 2022). At the plasma membrane, they control metabolite transport at the cellular level, mediating not only nutrient uptake and waste excretion but also communication and metabolic interaction of the cell with its environment (Pizzagalli et al, 2021; Alam et al, 2023). Intracellularly localized SLCs are crucial for the creation of different chemical and metabolic environments in the context of compartmentalization of biochemical processes within membrane-surrounded organelles (Bar-Peled and Kory, 2022).

SLCs function as facilitative or secondary active transporters, either equilibrating solute gradients or concentrating solutes by coupling the transport of several substrates through symport or antiport (Colas et al, 2016; Drew and Boudker, 2024). The ability to facilitate transport of different molecules or ions along or against a concentration gradient allows transporters to simultaneously control multiple biochemical pathways and the intracellular environment. While some SLCs are very specific for a given substrate, others are promiscuous and able to transport a range of compounds with different chemical structures. This is mirrored from the substrate-centric perspective; while certain compounds are transported by a single SLC, others are substrates for multiple SLCs. Though certain SLCs transport the same substrate, they can still differ in terms of their substrate affinity, specificity, and transport capacity (Meixner et al, 2020; Gauthier-Coles et al, 2021; Bröer, 2023).

Beyond controlling intracellular metabolite concentrations, SLCs influence metabolism by regulating metabolic signals on different levels. This can be at the posttranslational level as, for example, glutamine transporters at the plasma membrane regulate mTOR and autophagy, while lysosomal SLC38A9 is involved in the recruitment and regulation of mTORC1, a master regulator of cell growth, by sensing cellular nutrient availability (Nicklin et al, 2009; Rebsamen et al, 2015; Wang et al, 2015). At the transcriptional level, depletion of individual amino acids causes differential transcriptional responses of transporter expression, enabling cells to overcome conditions of nutrient limitation (Rebsamen et al, 2022; Chidley et al, 2024). For example, nucleoside transporters from SLC families 28, 29, and 35 modulate BRD4-dependent epigenetic states and transcriptional regulation (Li et al, 2021). Furthermore, metabolites themselves have regulatory functions and act as important signaling molecules driving feedforward and feedback mechanisms (Cable et al, 2021; Baker and Rutter, 2023). SLC-mediated metabolite exchange is not confined to individual cells but plays a critical role in the systemic interactions among distant metabolic organs via the circulatory system. For example, transporters of the SLC2 and SLC5 families are crucial for the uptake and distribution of glucose across the organism (Klip et al, 2024). A prime example of SLC regulation is seen in glucose homeostasis, where insulin stimulates the translocation of the glucose transporter SLC2A4 (GLUT4) from intracellular vesicles to the plasma membrane in muscle and fat cells, and its dysregulation may contribute to insulin resistance and type-2 diabetes (Zisman et al, 2000).

Understanding how SLCs participate in the regulation of metabolic processes is of therapeutic importance as alteration of transporter function is at the basis of many metabolic perturbations observed both in rare monogenic disorders and common diseases (Zhang et al, 2019; Alam et al, 2023). Targeting SLCs has already been successful, for example, in the treatment of type-2 diabetes by inhibiting SLC5A2 (SGLT2), a sodium-dependent glucose transporter responsible for glucose reabsorption in the kidney (Lin et al, 2015). With metabolic reprogramming being a hallmark of cancer (Hanahan, 2022), SLCs are also involved in tumorigenesis, cancer cell growth, and metastasis. Cancer cells frequently upregulate glucose and glutamine transporters to meet their increased energetic and biosynthetic requirements (Finley, 2023). SLC-mediated transport also enables adaptation of cancer cells to changes in nutrient availability during the metastatic process (Christen et al, 2016; Elia et al, 2019; Rinaldi et al, 2021; Bian et al, 2020). This illustrates how altered metabolism in disease mechanistically converges with metabolite uptake and provides evidence for the therapeutic potential to target SLCs in cancer, though metabolic adaptation is considered a major challenge. Therefore, it is important to understand the interplay between SLC expression and metabolism better.

Addressing the functional complexities of the entire SLC superfamily, in the context of their critical roles in metabolism and the challenges posed by their diverse expression patterns and substrate specificities, necessitates approaches beyond the few traditional cognate assays. Advancements in gene-editing technologies, particularly CRISPR/Cas9, have provided a powerful toolset for selectively manipulating individual SLC genes within a controlled experimental environment. The RESOLUTE consortium (https://re-solute.eu/) was formed to obtain functional data on all SLC transporters encoded in the human genome and shed light on the interface between chemistry and biology that gatekeeps the traffic of chemical substances (Superti-Furga et al, 2020). We opted for a comparative, systems-level strategy that incorporates both loss-of-function and gain-of-function methodologies across the entire SLC superfamily and established a collection of genetically modified human cell lines. This methodical approach facilitated a precise assessment of individual SLC roles in steady-state cellular metabolism under standard cell culture conditions. Coordinated in the same large, concerted effort, involving dozens of academic and industry laboratories, are accompanying studies investigating the global SLC-interactome, systematic genetic interaction mapping and integration of data and knowledge domain in a unified, multimodal landscape (Frommelt et al, 2025; Wolf et al, 2025; Goldmann et al, 2025) (Fig. 1A).

In this study, the basis for the other three studies, we used transcriptomic and metabolomic profiles to establish comparable functional annotations and identify kinship among transporters. We leverage two omics techniques to capture a comprehensive survey of the repercussions of modulating SLC activity. Besides identifying metabolic signatures reflecting transported substrate or biological function for individual SLCs, the acquisition of omics profiles across the entire SLC superfamily allows for the discovery of shared and coordinated functions between groups of SLCs. Based

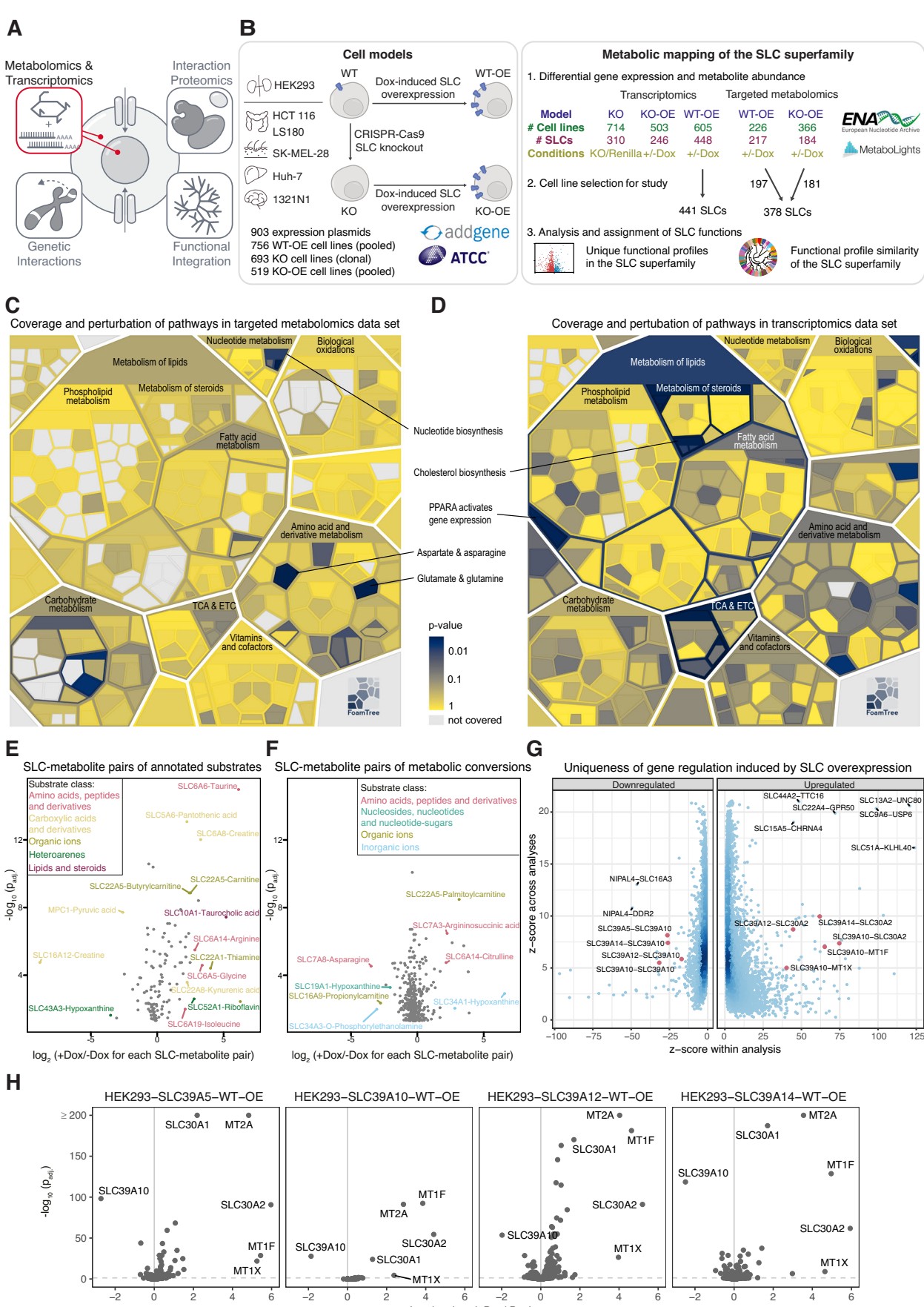

◀ **Figure 1.  Multi-omic analysis of SLC superfamily-mediated metabolic regulation.**

(A) Overview representation of the SLC superfamily-wide RESOLUTE paper collection. (B) Workflow for RESOLUTE cell line generation and acquisition and analysis of targeted metabolomics and RNA-sequencing data sets. For the generation of an SLC cell line collection, the Jump-In™ T-REx™ HEK293 cell line and a panel of five human cancer cell lines of different tissue origin (HCT 116/colon, LS180/colon, 1321N1/brain, SK-MEL-28/skin, and Huh-7/liver) were selected. A set of cell lines with doxycycline-controlled overexpression in wild-type Jump-In™ T-REx™ HEK293 was generated for all SLCs (WT-OE). Among the cancer cell lines, a parental cell line for each individual SLC gene was chosen based on its expression pattern across the panel, and two independent monoclonal knock-out (KO) cell lines generated. Subsequently, the genetically depleted genes were re-expressed with doxycycline-controlled expression vectors of the cognate codon-optimized cDNA to create two KO-overexpression (KO-OE) cell models. SLC expression in KO-OE and WT-OE cell lines was induced by overnight treatment with doxycycline prior to the collection of both untreated and treated samples (four biological replicates per condition for metabolomics, two biological replicates per condition for transcriptomics). Expression plasmids are available at Addgene (https://www.addgene.org/depositor-collections/re-solute/) and HCT 116 KO cell lines at ATCC (https://www.atcc.org/cell-products/cell-models/solute-transporter-carrier-cells). Raw data is available at public repositories (MetaboLights MTBLS10077; ENA PRJEB81360) and the differential analyses in interactive dashboards on the RESOLUTE webpage (https://re-solute.eu/resources/dashboards). (C) Frequency of metabolite abundance changes in Reactome metabolic pathways. For each pathway, the number of metabolomics analyses featuring differentially abundant metabolites that were matched to any reaction within the pathway were counted (metabolomics data set of 378 SLCs, metabolites with $P_{adj.}$ <0.05). Significance of the observed frequencies was tested by permutation, shuffling the identities of the quantified metabolites 200,000 times. Significance levels are visualized on a Voronoi treemap of the hierarchical structure of all sub-pathways of the human Metabolism pathway in Reactome, to show areas specifically affected by SLC overexpression. (D) Frequency of gene expression changes in Reactome metabolic pathways. For each pathway, the number of transcriptomics analyses featuring differentially expressed genes that were matched to any reaction within the pathway were counted (transcriptomics data set of 441 SLCs, genes with $P_{adj.}$ <0.05). Significance of the observed frequencies was tested by permutation, shuffling the identities of the quantified genes 200,000 times. Significance levels are visualized on a Voronoi treemap of the hierarchical structure of all sub-pathways of the human Metabolism pathway in Reactome, to show areas specifically affected by SLC overexpression. (E) Significant SLC-metabolite pairs of SLCs and their annotated substrate. $Log_2$ fold changes and adjusted $P$ values between $+/-$doxycycline (Dox) of 124 SLC-metabolite pairs of 57 SLCs are visualized. Significance for each SLC-metabolite pair was determined by contrasting each metabolite's normalized and batch-corrected values in doxycycline-induced samples to the corresponding uninduced samples (four replicates each) in a one-way ANOVA for WT-OE cell lines, or a two-way ANOVA for KO-OE cell lines, modeling the clone effect as an additional factor. $P$ values were adjusted using the Benjamini–Hochberg correction. (F) Significant SLC-metabolite pairs of SLCs and metabolic conversions of their annotated substrate. $Log_2$ fold changes and adjusted $P$ values between $+$Dox/-Dox of 382 SLC-metabolite pairs of 82 SLCs are visualized. Pairs are colored according to the substrate class of the respective substrate. Significance for each SLC-metabolite pair was determined by contrasting each metabolite's normalized and batch-corrected values in Dox-induced samples to the corresponding uninduced samples (four replicates each) in a one-way ANOVA for WT-OE cell lines, or a two-way ANOVA for KO-OE cell lines, modeling the clone effect as an additional factor. $P$ values were adjusted using the Benjamini–Hochberg correction. (G) Uniqueness of gene expression changes induced by SLC overexpression. Strong differential gene expression in specific HEK293 WT-OE cell lines ( $+$ Dox/$-$Dox) were identified by calculation of z-scores of the shrunken log fold change of each gene within a certain analysis and across all analyses. Genes were then further filtered for significance of differential expression ($P_{adj.}$ <0.05; statistical analysis using Wald test and $P$ value adjustment using the Benjamini–Hochberg correction) and for minimal signal (the gene had to have in one condition at least 50 read counts in both replica). Dots of selected pairs to illustrate the differential expression of zinc transporters and metallothioneins upon overexpression of SLC39 family members are colored in red. (H) Differential gene expression analysis between $+$Dox/$-$Dox samples of HEK293-SLC39A5-WT-OE, HEK293-SLC39A10-WT-OE, HEK293-SLC39A12-WT-OE, HEK293-SLC39A14-WT-OE cell lines. This analysis only considers endogenous transcripts and excludes transcripts of codon-optimized, overexpressed SLCs. Statistical analysis was performed using the Wald test. $P$ values were adjusted using the Benjamini–Hochberg correction. The dashed line indicates $P_{adj.}$ <0.05.

on the hypothesis that SLCs with similar omics profiles share functional properties, our data set also enables the assignment of novel functions to poorly studied SLCs using the guilt-by-association principle.

We published all generated data sets and developed protocols at open-access databases, and made most of our created plasmid vectors and cell lines publicly available through non-profit repositories (https://re-solute.eu/resources/reagents). This collection of data sets, reagents (>900 plasmids, >900 sgRNAs), and cell lines (>1000) will be an important tool for further elucidating SLC function by the scientific community.

## Results

### Pipeline for functional profiling of SLC family

For the RESOLUTE project we selected a panel of five human cancer cell lines of different tissue origin (HCT 116/colon, LS180/colon, 1321N1/brain, SK-MEL-28/skin, and Huh-7/liver), cumulatively expressing more than 75% of the SLC superfamily (Fig. EV1A, data from (Goldmann et al, 2025), ENA Project number PRJNA545487). To study SLC function systematically, we used a controlled cellular system (Fig. 1B). We first selected one parental cell line for each individual SLC gene based on its expression pattern across the panel and generated two independent

monoclonal knock-out (KO) cell lines. HCT 116 was chosen as the primary cell model due to its suitability for deriving monoclonal cell lines. If an SLC was not or relatively low expressed in the HCT 116 cell line, we selected another cell line from the panel with the highest expression for the respective SLC. KO cell line generation was not always successful, for instance, generation of KO clones failed for five (SLC25A26, MTCH2, SLC30A9, SLC25A3, and SLC35B1) out of 6 SLCs found essential in HCT 116 cells (Wolf et al, 2025). Subsequently, we re-expressed the genetically depleted genes with doxycycline-controlled expression vectors of the cognate codon-optimized cDNA to create two KO-overexpression (hereafter referred to as KO-OE) cell models per target (Methods). A Strep-HA tag was added to the codon-optimized SLC sequences at the C- or N-terminal. An additional set of cell lines with doxycycline-controlled overexpression in wild-type Jump-In™ T-REx™ HEK293 was generated for all SLCs (hereafter referred to as WT-OE). Overall, we generated 903 expression plasmids, 756 pooled WT-OE cell lines, 693 clonal KO cell lines and 519 pooled KO-OE cell lines. Expression plasmids are available through the public repository Addgene (https://www.addgene.org/depositor-collections/re-solute/) and HCT 116 KO cell lines through ATCC (https://www.atcc.org/cell-products/cell-models/solute-transporter-carrier-cells) (Fig. 1B).

SLC expression in KO-OE and WT-OE cell lines was induced by overnight treatment with doxycycline prior to the collection of both untreated and treated samples to increase the likelihood of

detecting metabolic or transcriptomic changes in response to SLC expression while enabling normalization to uninduced cells. RNA-Seq was performed for 605 WT-OE, 714 KO, and 503 KO-OE cell lines, and 448, 310, and 246 SLCs, respectively (Fig. 1B). For targeted metabolomics, we used an ion-pairing reversed-phase liquid chromatography-mass spectrometry method based on a commercially available kit (The Agilent Metabolomics Dynamic MRM Database and Method) as it met our requirements of broad metabolite coverage, robustness of data acquisition, and industry standards for comparability. It provides reproducible and quantitative measurement of 189 metabolites spanning different compound classes (e.g., nucleotides, amino acids, sugars), covering many central metabolic pathways including glycolysis, TCA cycle, oxidative phosphorylation, pentose phosphate pathway, purine and pyrimidine metabolism, amino acid metabolism, and one-carbon metabolism (Dataset EV1). For metabolomics, our goal was to obtain at least one measurement of the effects of SLC over-expression for each SLC, preferably using the KO-OE model to minimize background activity from endogenous transporters and to maintain a cellular environment that is as close to physiological conditions as possible. For SLCs without a KO-OE model available at the time of metabolomics profiling, we used the HEK293 WT-OE model. Targeted metabolomics was acquired for 226 WT-OE and 366 KO-OE cell lines, i.e., for 217 and 184 SLCs, respectively (Fig. 1B). Differential expression (transcriptomics) and metabolite abundance (metabolomics) between doxycycline-induced and uninduced conditions were analyzed for each SLC-OE cell line. Raw data for both –omics methods is available at public repositories (MetaboLights MTBLS10077, European Nucleotide Archive PRJEB81360), and the processed data is accessible through interactive dashboards on the RESOLUTE webpage (https://re-solute.eu/resources/dashboards).

Here, to analyze these –omics data sets across all measured SLCs, we selected for each SLC one cell line model and respective data sets to enable analyses and comparisons across the SLC superfamily. We aimed to obtain a broad and quantitative picture of the similarities and dissimilarities among transporters, as two transporters that elicit similar profiles are likely to be more functionally related than two that create dissimilar changes. In some cases, it may also be possible to directly infer the transported substrate from the profile changes (Fig. 1B). For transcriptomic analyses, RNA-Seq of WT-OE cell lines for 441 SLCs were included (Figs. 1B and EV1B; Dataset EV2), as this data set was the most complete, covering 99% of the SLC superfamily members and at least one member of each of the 69 SLC families (Fig. EV1C), and all obtained in the same cellular background. To assess the variation and potential patterns in the differential analyses of respective SLCs, we performed a principal component analysis. We observed high variance in the data set as the first two principal components explained only 6.8% and 5.5% of the variance. Furthermore, we did not observe a separation into distinct groups or according to the structural fold of the respective SLCs (Appendix Fig. S1A). For targeted metabolomic analyses, we included data sets for 378 SLCs from WT-OE and KO-OE cell lines (Figs. 1B and EV1B; Dataset EV2), i.e., 85% of the entire superfamily, covering 67 of the 69 SLC families (Fig. EV1D). We prioritized the selection of KO-OE cell lines over WT-OE cell lines for reasons mentioned earlier. We observed a separation of the differential analyses by cell line model and parental cell line (WT-OE and KO-OE; Jump-In™ T-

REx™ HEK293 and HCT 116), which explained only a small fraction of the variance (PC1 8.0%). The second principal component was only slightly lower and unrelated to the cell line model and parental cell line (PC2 7.6%). Similar to the RNA-seq data set, we did not observe a separation into distinct groups or according to the structural fold of the respective SLCs (Fig. EV1E; Appendix Fig. S1B).

To assess the coverage and complementarity of targeted metabolomics and transcriptomics data sets for the interrogation of metabolism, we analyzed which metabolic pathways were altered based on differential metabolite abundance and gene expression upon SLC overexpression. Using targeted metabolomics, we observed significant differences between doxycycline-induced and uninduced samples for 282 SLCs (75% of profiled cell lines, metabolites with adjusted $P$ value < 0.05; Dataset EV3). Across all profiled SLCs, the changes in metabolite abundance and gene expression impacted 257 and 308 pathways, respectively, out of a total of 328 metabolic pathways in the Reactome database (https://reactome.org/PathwayBrowser/#/R-HSA-1430728; (Milacic et al, 2024)) (Fig. 1C,D; Dataset EV4). Therefore, our approach enabled the interrogation of a considerable portion of the metabolic space. Permutation analysis and subsequent enrichment test showed that in the metabolomics data set, metabolites in carbohydrate, nucleotide, and amino acid metabolism were most frequently altered while in the transcriptomics data set genes in lipid metabolism, the TCA cycle, and the electron transport chain showed most frequent expression changes. With their different coverage and perturbation of metabolic pathways, the two data modalities complemented each other (Fig. 1C,D).

From a metabolite-centric perspective, we observed a wide spectrum in the frequency of significant changes across the 189 metabolites included in our method (Appendix Fig. S1C). We found that 15 metabolites were altered in more than 20% of the cell lines tested. To assess the influence of the LC–MS method on the detection of significant changes for different metabolites, we calculated the frequency of robust detection (defined as a measurement value above the minimum calibration point for that metabolite) and examined its relationship to the frequency of significant changes. We found that while the frequency of robust detection was weakly correlated with frequency of significant change (Spearman's $r$ 0.569), a number of metabolites including lysine, aminoadipic acid, citric/isocitric acid, aconitic acid and cis-aconitic acid were frequently changed despite being less robustly detected (Appendix Fig. S1D). Hence, robustness of metabolite detection is not the sole factor driving the detection of significant changes. We additionally compared the average magnitude of induced / uninduced fold change with the frequency of significant change as well as the frequency of robust detection (Fig. EV1F). As expected, metabolites with low frequency of robust detection tended to have inflated fold change magnitudes, likely due to low signal-to-noise ratio. However, the average magnitude of fold change was not correlated with the frequency of significant change (Spearman's $r$ −0.127).

Aconitate, aspartate, and citrate displayed the highest frequency of significant change, suggesting that changes in the TCA cycle may be a common effect of perturbing transport-related metabolism. This was in line with the transcriptomics data set in which the TCA cycle was frequently altered. Conversely, 15 metabolites were changed in only one, two or three cell lines and may indicate

unique substrates or their metabolic conversions, or reflect perturbations in unique metabolic pathways (Appendix Fig. S1C).

Due to the limited coverage of the targeted metabolomics method, we assessed its potential bias towards certain SLCs and substrates by using a literature-based and manually curated annotation and classification of SLC substrates (Goldmann et al, 2025). We grouped SLCs by substrate class and observed that at least 59% of the SLCs in each class showed significant effects, indicating that metabolic changes captured by our method were not biased towards certain SLC substrate classes (Fig. EV1G).

Even though our approach was not tailored towards the direct identification of novel substrates due to method coverage and metabolic adaptation upon substrate transport, we searched the differential metabolite abundance profiles for known substrates of SLCs to validate the workflow. In total, 279 of the 378 profiled SLCs had one or more annotated substrates and for 141 of them, our targeted metabolomics method covered at least one substrate. For 57 of those SLCs (40%) from different substrate classes, we identified significant changes of annotated cognate substrates, such as SLC5A6:pantothenic acid, SLC6A8:creatine, SLC22A5:carnitine, and SLC6A6:taurine pairs, for which we observed increased substrate levels. Conversely, we found decreased cellular levels of pyruvate upon overexpression of MPC1, consistent with pyruvate utilization in mitochondria (Fig. 1E). Since altered transport not only affects steady-state levels of transported substrates but also their metabolic derivatives, we expanded the analysis to metabolic conversions, i.e., metabolites which are either educts or products of reactions involving annotated SLC substrates. For this, we mapped annotated SLC substrates to Reactome pathway reactions and considered adjacent metabolites covered by our targeted metabolomics method. Of the 279 substrate-annotated SLCs included in this study, our method covered the measurement of such metabolic conversions for 233 SLCs. For 82 of those SLCs (35%), we found the levels of either an educt or a product of a metabolic reaction involving at least one of their annotated substrates significantly changed (Fig. 1F). For 37 SLCs we detected both an annotated substrate and a metabolic derivative. Annotated substrates and metabolic conversions were most frequently identified for the substrate class of *amino acid, peptides and derivatives* transporters. *Carboxylic acids and derivatives* and *nucleosides, nucleotides and nucleotide-sugars* transporters were the second most frequently identified substrate classes for annotated substrates and metabolic conversions, respectively. The least observable substrates classes were *transition element cations*, *inorganic ions* and *lipids and steroids*. The frequency of significant changes was significantly higher when the metabolite was either an annotated substrate or a metabolic conversion compared to the perturbation frequency of all metabolites across all cell lines (Fisher's exact test, both $P < 0.0001$) (Fig. EV1H). These results confirmed that our targeted metabolomics approach mainly served the molecular profiling of the functional consequences downstream of SLC expression, but that we nevertheless also measured direct or closely proximal substrate abundance changes for a proportion of SLCs (102 out of 279 profiled SLCs with substrate annotation).

Arguably, the transcriptional response to SLC overexpression is less suitable for the identification of direct chemical substrates of the SLCs under investigation. However, we reasoned that gene expression changes should be mainly elicited by the activity of the transporter under investigation, i.e., by changing the concentration of specific metabolites and ions. Importantly, the transcriptional profile is quantitative and robust, covers several thousand genes involved in virtually all cellular functions, has been the workhorse for differential molecular responses of cells to perturbations for decades, and is one of the most interoperable phenotypic descriptor of cell states (Lamb et al, 2006; Niepel et al, 2017; Keenan et al, 2018; Subramanian et al, 2017).

In our transcriptomic data set many genes were affected in multiple different cell lines, suggesting that they may reflect transport-independent effects, such as a general cellular stress response caused by high transgene expression levels. Nevertheless, the approach also identified genes that are uniquely affected by the expression of specific codon-optimized SLC sequences (Fig. 1G; Dataset EV5). For example, the genes for the metallothionein family members MT1F, MT1X and MT2A, that are known to protect against metal toxicity by binding heavy metals such as zinc and copper (Chen et al, 2024), were upregulated in cell lines overexpressing one of the zinc importers SLC39A5, SLC39A10, SLC39A12 or SLC39A14 (Fig. 1H). In addition, two zinc exporters, SLC30A1 and SLC30A2, were upregulated in these cell lines, likely to counter the excessive zinc influx, while the zinc importer SLC39A10 was downregulated (Fig. 1H). Altogether, these data indicated that the increased metal import activity in these cells triggered a transcriptional response aimed at lowering the concentration of unbound intracellular metals. Importantly, this exemplified how transcriptomics may inform on transport of substrates not covered by the targeted metabolomics method. Further, it suggested how superfamily-wide systematic perturbation of SLCs may provide insights into the function of individual SLCs through comparable transcriptional responses, conceptually validating the approach.

## Insights to SLC function by SLC-metabolite pairs

To identify strong metabolomic changes that could potentially lead to new substrate identification in validation studies, we used a rationale based on the benchmarking data set of annotated substrates and metabolic conversions for SLCs as described above. We grouped all measured SLC-metabolite relationships, that we call "pairs", into four categories: (i) "expected" pairs of SLCs with their annotated substrates ("annotated substrate"); (ii) "expected" pairs of SLCs and metabolic conversions of their substrate, i.e., an educt or product of a reaction with the substrate as described previously ("metabolic conversion"); (iii) "novel" SLC-metabolite pairs, consisting of metabolite hits not matching annotated substrates or their immediate metabolic conversions ("novel for non-orphan SLC"); (iv) metabolite hits for orphan SLCs ("novel for orphan SLC") (Fig. 2A; Dataset EV3). We compared the magnitude of change upon differential abundance analysis across the four categories and found that changes in metabolite levels of SLC-annotated substrate pairs were significantly larger than those of SLC-metabolic conversion pairs (mean $\log_2$ fold changes, Kruskal–Wallis test, $P < 0.0001$). This indicated a tendency for direct substrates to show larger abundance changes upon SLC overexpression, a finding that is not surprising, as metabolic conversion may be multidirectional and diffused. This also suggested that novel SLC-metabolite pairs with a relatively large $\log_2$ fold change may imply novel putative substrates for the respective SLCs (Fig. 2A). To that end, the data set was explored on

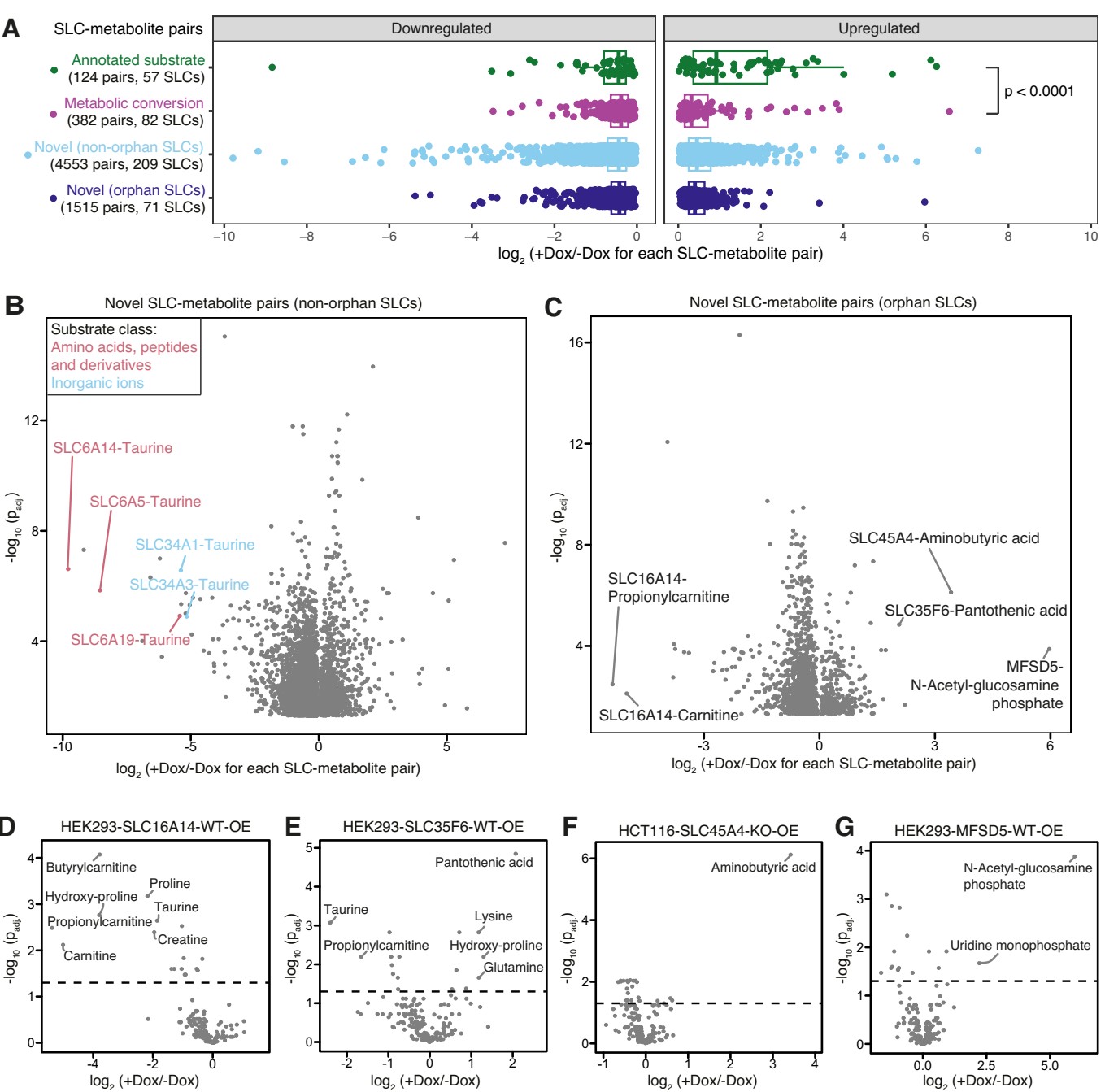

a per-metabolite basis, which illustrated on the one hand all SLC-metabolite pairs divided per metabolite and, on the other hand, the distribution of magnitude of change as well as the SLCs causing the largest changes (Fig. EV2A).

For 209 SLCs with annotated substrates in the knowledge domain, we found >4,500 SLC-metabolite pairs significantly changed that had not previously been annotated in connection with these SLCs (category iii, Fig. 2A). Taurine was among the most strongly altered metabolites with a more than 30-fold decrease upon expression of five SLCs: SLC6A5, SLC6A14, SLC6A19, SLC34A1 and SLC34A3 ($log_2$ fold change > 5). While these SLCs had diverse annotated substrates, such as different amino acids and

phosphate, all of them couple substrate transport to sodium ($Na^+$) (Fig. 2B, (Goldmann et al, 2025)). This suggested that SLC-induced taurine reduction may have been related to osmotic imbalance and/or sodium coupling.

Lastly, we found >1500 SLC-metabolite pairs for 71 orphan SLCs (category iv, Fig. 2A). Among these, the strongest changes were observed for SLC45A4:gamma aminobutyric acid (GABA), SLC35F6:pantothenic acid, MFSD5:n-acetylglucosamine phosphate, SLC16A14:propionylcarnitine and SLC16A14:carnitine (Fig. 2C). Therefore, these SLCs were considered for further experimental validation. SLC16A14 overexpression reduced intracellular carnitine, propionylcarnitine and butyrlcarnitine levels

**Figure 2. SLC-metabolite differential abundance pairs illuminate known and novel functional relationships.**

(A) Boxplot of all significant SLC-metabolite pairs divided into categories "annotated substrate", "metabolic conversion", "novel (non-orphan SLCs)", "novel (orphan SLCs)". The number of pairs and the respective number of SLCs are indicated per category. Kruskal–Wallis test was performed between all categories, and the $P$ value for the significant difference indicated between differential metabolite levels of "annotated substrate" and "metabolic conversion" categories ($P = 0.000004$). The $\log_2$ fold change minima, lower whisker, Q1 percentiles, center, Q3 percentiles, upper whisker, maxima for categories "annotated substrate"; "metabolic conversion"; "novel (non-orphan SLCs)"; "novel (orphan SLCs)" are for upregulated pairs 0.02, 0.02, 0.37, 0.92, 2.16, 4.85, 6.26; 0.02, 0.02, 0.15, 0.32, 0.71, 1.55, 6.57; 0.02, 0.02, 0.25, 0.45, 0.79, 1.59, 7.27; 0.04, 0.04, 0.25, 0.40, 0.64, 1.21, 5.97 and for downregulated pairs −8.84, −1.60, −0.80, −0.44, −0.27, −0.05, −0.05; −3.49, −1.27, −0.63, −0.38, −0.21, −0.04, −0.04; −9.79, −1.39, −0.71, −0.44, −0.26, −0.02, −0.02; −5.37, −1.13, −0.62, −0.43, −0.28, −0.03, −0.03, respectively. (B) Significant novel SLC-metabolite pairs for SLCs with an annotated substrate. Respective metabolites are not substrates or metabolic conversions. Pairs are colored according to the substrate class of the respective SLC substrate. $\log_2$ fold changes and adjusted $P$ values between +/−doxycycline (Dox) of 4553 SLC-metabolite pairs of 209 SLCs are visualized. Significance for each SLC-metabolite pair was determined by contrasting each metabolite's normalized and batch-corrected values in Dox-induced samples to the corresponding uninduced samples (four replicates each) in a one-way ANOVA for WT-OE cell lines, or a two-way ANOVA for KO-OE cell lines, modeling the clone effect as additional factor. $P$ values were adjusted using the Benjamini–Hochberg correction. (C) Significant novel SLC-metabolite pairs for orphan SLCs. $\log_2$ fold changes and adjusted $P$ values between +/−doxycycline (Dox) of 1515 SLC-metabolite pairs of 71 SLCs are visualized. Significance for each SLC-metabolite pair was determined by contrasting each metabolite's normalized and batch-corrected values in Dox-induced samples to the corresponding uninduced samples (four replicates each) in a one-way ANOVA for WT-OE cell lines, or a two-way ANOVA for KO-OE cell lines, modeling the clone effect as additional factor. $P$ values were adjusted using the Benjamini–Hochberg correction. (D–G) Differential metabolite abundance analysis between +/−doxycycline (Dox) samples of (D) HEK293-SLC16A14-WT-OE, (E) HEK293-SLC35F6-WT-OE, (F) HCT116-SLC45A4-KO-OE, (G) HEK293-MFSD5-KO-OE cell lines. Significance for each SLC-metabolite pair was determined by contrasting each metabolite's normalized and batch-corrected values in Dox-induced samples to the corresponding uninduced samples (four replicates each) in a one-way ANOVA for WT-OE cell lines, or a two-way ANOVA for KO-OE cell lines, modeling the clone effect as additional factor. $P$ values were adjusted using the Benjamini–Hochberg correction. The dashed line indicates $P_{adj.} < 0.05$.

(Fig. 2D), suggesting a possible functional overlap with the known carnitine effluxer SLC16A9 (Suhre et al, 2011). Other metabolite changes included decreased hydroxy-proline, proline, taurine, and creatine, all molecules with potential roles as osmolytes. Metabolite abundance changes for SLC35F6 partially overlapped with SLC16A14 in decreased taurine and propionylcarnitine, while several amino acids (lysine, glutamine and hydroxy-proline) as well as pantothenic acid were increased (Fig. 2E). SLC45A4 and MFSD5 displayed stronger and more selective changes among the metabolites covered by our method. SLC45A4 overexpression increased GABA by over 10-fold ($\log_2$ fold change 3.42) (Fig. 2F), and MFSD5 overexpression increased N-acetylglucosamine phosphate by over 60-fold ($\log_2$ fold change 5.97) and uridine monophosphate by 4.65-fold ($\log_2$ fold change 2.22) (Fig. 2G). Evaluating the frequency of significant change in the respective metabolites across all profiled SLCs, GABA was among the least frequently changed metabolites (changed only in six cell lines), while pantothenic acid, N-acetylglucosamine phosphate, propionylcarnitine and carnitine showed significant changes in 3–25% of cell lines (Fig. EV1F). Based on the strong differential change and the unique metabolite profile, we focused on the SLC45A4:GABA pair for further validation.

## SLC45A4 mediates the export of polyamines

Since HCT116-SLC45A4-KO-OE cells displayed Dox-induced increase in GABA (Fig. 2F), and immunofluorescence showed that HA-tagged SLC45A4 localized to the plasma membrane in these cells (Fig. EV3A), we investigated whether SLC45A4 mediates GABA uptake. GABA supplementation to culture media caused a time-dependent increase in intracellular GABA concentration of HCT 116 cells overexpressing the GABA transporter SLC6A1, but not SLC45A4, ruling out direct uptake as the mechanism of SLC45A4-mediated GABA increase (Fig. EV3B). We then investigated the possibility that transported compounds not covered in our initial metabolomics data set was feeding into GABA biosynthesis or otherwise perturbing GABA metabolism. Since the heterologous expression of mouse SLC45A4 in yeast cells has

been reported to mediate sucrose uptake (Bartölke et al, 2014), we measured sucrose uptake but did not observe increased uptake in SLC45A4 overexpressing cells (Fig. EV3C). We next examined the metabolic pathways involved in intracellular GABA production, i.e., through the GABA shunt via direct conversion of glutamic acid via glutamate decarboxylase, encoded by *GAD1* and *GAD2*, or through the polyamine synthesis pathway from putrescine by diamine oxidase (DAO), encoded by the *AOC1* gene (Amine Oxidase, Copper-containing, 1) (Fig. 3A). Using stable isotope labeling, we probed the contribution of these pathways to GABA production in our cellular system. Culturing HCT116-SLC45A4-KO-OE cells in $^{13}$C-glutamic acid did not lead to the production of labeled GABA (Fig. EV3D). However, GABA was labeled upon culturing the cells in media supplemented with $^{13}$C$^{15}$N-arginine, a precursor for polyamines. In addition, we observed substantial labeling of ornithine and the polyamines putrescine, spermidine and spermine, supporting the notion that GABA may originate from putrescine (Fig. 3B, upper row). Labeled putrescine and spermidine levels decreased intracellularly and were elevated in supernatants of Dox-induced cells, suggestive of increased export of polyamines (Fig. 3B, lower row). To confirm that increased GABA derived from putrescine, cells were treated with difluoromethylornithine (DFMO), an inhibitor of ornithine decarboxylase (ODC1), the enzyme that produces putrescine from ornithine. ODC1 inhibition caused a stark depletion in intracellular putrescine levels and blocked SLC45A4-mediated GABA accumulation (Fig. EV3E). Hence, SLC45A4 appeared to mediate an increase in putrescine-derived GABA, justifying an in-depth investigation of the pathway from putrescine to GABA.

AOC1, the human DAO responsible for converting putrescine to 4-aminobutyraldehyde (ABAL) for subsequent oxidation to GABA, is not expressed in HCT 116 cells (data from (Goldmann et al, 2025); ENA Project number PRJNA545487). In line with this, AOC1 KO did not affect putrescine export or GABA accumulation (Fig. 3C). Since fetal bovine serum (FBS), a component of regular cell culture media, contains DAO activity (Gahl and Pitot, 1979), we hypothesized that the putrescine to GABA conversion occurred through an extracellular pathway and that serum-derived DAO activity converted extracellular

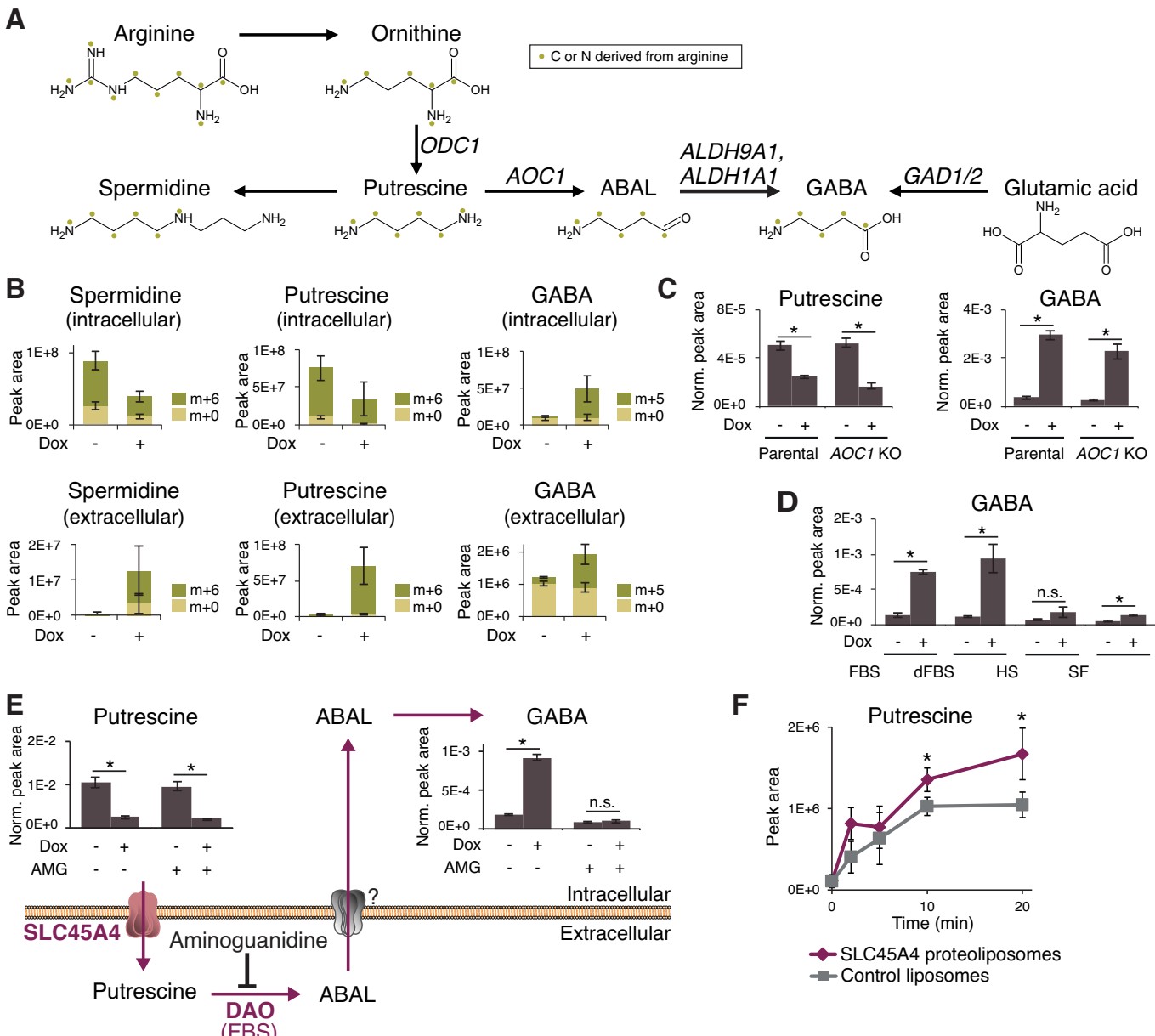

**Figure 3. The orphan SLC45A4 is a novel putrescine exporter.**

(A) Schematic representation of metabolic pathways that produce GABA. Carbon and nitrogen atoms potentially derived from arginine (and hence labeled from $^{13}C^{15}N$-arginine) are marked by small green dots. (B) Labeled (from $^{13}C^{15}N$-arginine) and unlabeled abundances of intracellular vs extracellular metabolites for HCT116-SLC45A4-KO-OE cells $+/-$ 24 h doxycycline (Dox) induction. Labeled species are as indicated for each metabolite. Bar heights represent means, error bars represent s.d. ($n = 6$). (C) Intracellular putrescine and GABA levels in HCT116-SLC45A4-KO-OE cells with and without AOC1 KO $+/-$ 24 h Dox induction. Bar heights represent means, error bars represent s.d. ($n = 3$). $P$ values (Welch's $t$ test): putrescine uninduced vs Dox parental cells, 2.34E-03; putrescine uninduced vs Dox AOC1 KO cells, 0.0183; GABA uninduced vs Dox parental cells, 8.73E-04; GABA uninduced vs Dox AOC1 KO cells, 6.83E-03. (D) Intracellular GABA levels in HCT116-SLC45A4-KO-OE cells cultured in regular FBS-containing media, media with dialyzed FBS (dFBS), media with horse serum (HS), or serum-free media (SF) $+/-$ 9 h Dox induction. Bar heights represent means, error bars represent s.d. ($n = 3$). $P$ values (Welch's $t$ test): FBS uninduced vs Dox, 1.23E-05; dFBS uninduced vs Dox, 0.0192; HS uninduced vs Dox, 0.0671; SF uninduced vs Dox, 1.37E-03. (E) Intracellular putrescine and GABA levels in HCT116-SLC45A4-KO-OE cells with and without DAO inhibition by aminoguanidine (AMG) $+/-$ 9 h Dox induction. Bar heights represent means, error bars represent s.d. ($n = 3$). Purple arrows show the proposed pathway leading to increased intracellular GABA accumulation in SLC45A4 overexpression cells: SLC45A4 mediates export of putrescine into the extracellular space, where it is then acted upon by serum-derived DAO activity to generate ABAL for GABA production. $P$ values (Welch's $t$ test): putrescine uninduced vs Dox without AMG, 2.74E-04; putrescine uninduced vs Dox with AMG, 4.99E-03; GABA uninduced vs Dox without AMG, 8.50E-06; GABA uninduced vs Dox with AMG, 0.505. (F) Putrescine quantification in SEC eluates of SLC45A4 proteoliposomes vs control liposomes over different uptake durations. Data points represent means, error bars represent s.d. ($n = 3$). $P$ values (Welch's $t$ test) for SLC45A4 proteoliposomes vs control liposomes comparisons: $t = 0$, 0.932; $t = 2$, 0.0610; $t = 5$, 0.588; $t = 10$, 0.0363; $t = 20$, 0.0371. Where presented, asterisks (*) and n.s. on significance bars indicate $P < 0.05$ and $P \geq 0.05$, respectively. (C–E) Metabolite peak areas were normalized to internal standard compound peak area and further normalized to cellular protein content as described in "Methods".

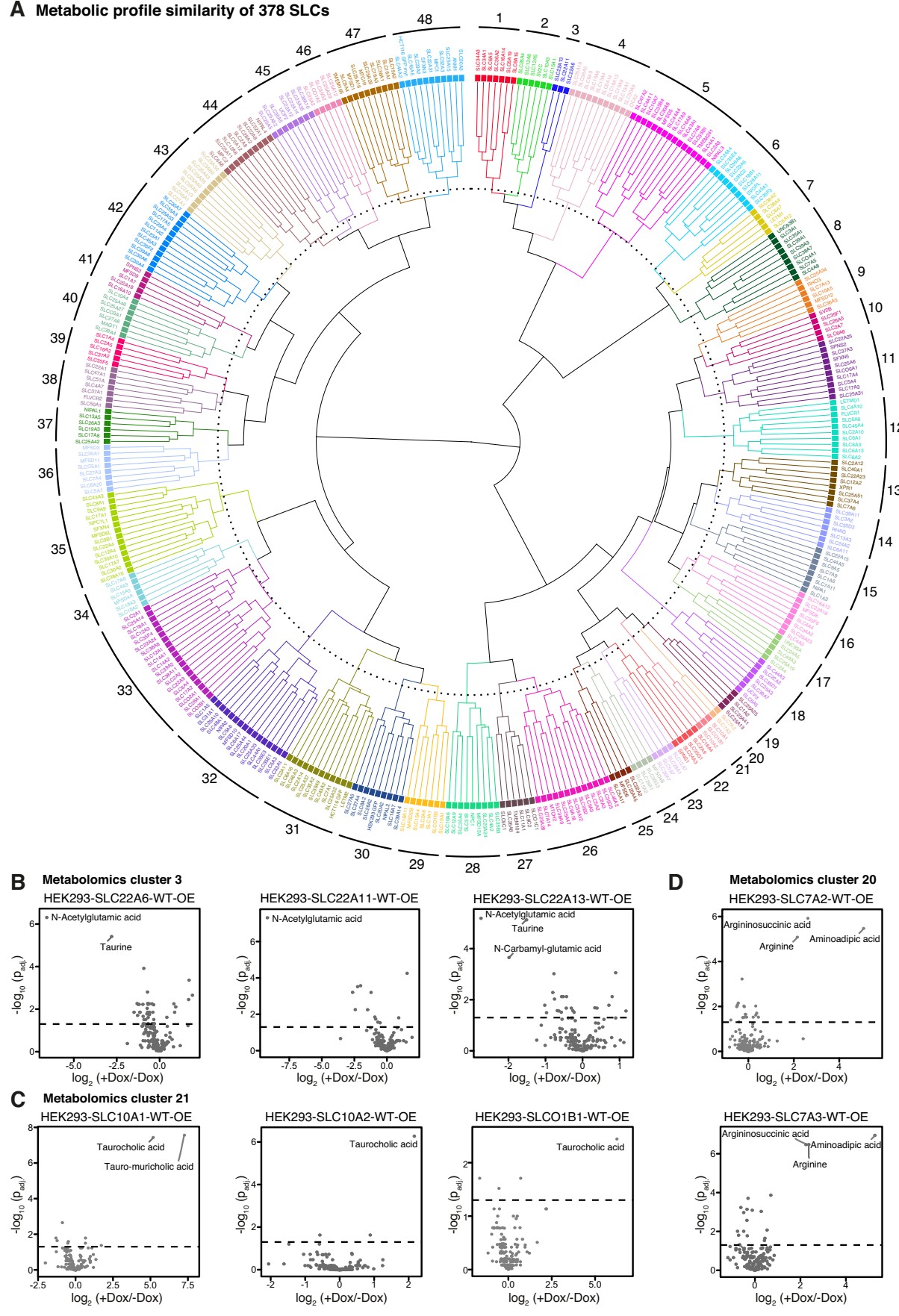

**A** Metabolic profile similarity of 378 SLCs

**B** Metabolomics cluster 3

**C** Metabolomics cluster 21

**D** Metabolomics cluster 20

Figure 4. Similarity clustering of metabolomic profiles reveals known and novel SLC functional relationships.

(A) Dendrogram of hierarchical clustering based on the Euclidean distance matrix of normalized metabolic profiles of 381 cell lines (metabolomics data set of 378 SLC cell lines and 3 GFP cell lines). (B) Differential metabolite abundance analysis upon $+/-$ doxycycline (Dox) induction of HEK293-SLC22A6-WT-OE, HEK293-SLC22A11-WT-OE and HEK293-SLC22A13-WT-OE cell lines. (C) Differential metabolite abundance analysis upon $+/-$ Dox induction of HEK293-SLC10A1-WT-OE, HEK293-SLC10A2-WT-OE and HEK293-SLCO1B1-WT-OE cell lines. (D) Differential metabolite abundance analysis upon $+/-$ Dox induction of HEK293-SLC7A2-WT-OE and HEK293-SLC7A3-WT-OE cell lines. $P$ values were adjusted using the Benjamini–Hochberg correction. The dashed line indicates $P_{adj.}$ <0.05.

putrescine to ABAL. We cultured HCT116-SLC45A4-KO-OE cells in different media serum conditions. Substitution of regular FBS with dialyzed FBS (dFBS), from which small molecules have been excluded, had no effect on SLC45A4-mediated GABA production. Conversely, SLC45A4 KO-OE cells cultured in horse serum (HS), which has low DAO activity (Kunimoto et al, 1985; Niskanen and Wharton, 1987), or serum-free media (SF) failed to accumulate GABA (Fig. 3D) while depletion of intracellular putrescine was unaffected (Fig. EV3F). This suggested that SLC45A4 exported putrescine and spermidine, while serum DAO activity converted exported putrescine into ABAL for subsequent GABA synthesis. Indeed, DAO inhibition by aminoguanidine (Caron et al, 1988) blocked GABA accumulation in SLC45A4 KO-OE cells (Fig. 3E). Since aminoguanidine is also a known inhibitor of nitric oxide synthase (Misko et al, 1993) and may have other cellular effects, we also directly probed the importance of serum DAO in GABA production by supplementation. While cells cultured in HS media accumulated significantly less GABA than in FBS, supplementation of porcine DAO to the media greatly increased the amount of GABA production for both FBS and HS conditions (Fig. EV3G), indicating that extracellular DAO activity was a limiting factor in the production of GABA from putrescine.

To assess whether SLC45A4 directly transports putrescine, we performed a cell-free uptake assay using proteoliposomes reconstituted with SLC45A4 protein. Following incubation in putrescine-containing assay buffer for defined time points, the SLC45A4 proteoliposomes or empty liposomes were separated from buffer by rapid centrifugation through size exclusion chromatography (SEC) spin columns (Fig. EV3H). Glycine was encapsulated within liposomes as an internal marker of liposome abundance. The external assay buffer contained putrescine, along with PIPES as an external marker of buffer carryover. Consistent with SLC45A4-catalyzed transport of putrescine, the putrescine/glycine and putrescine/PIPES ratios increased over time and were significantly higher in SLC45A4 proteoliposomes relative to control liposomes (Figs. 3F and EV3I). Hence, our data indicate that SLC45A4 directly transports putrescine.

## Similarity of omics profiles indicates functional relationships between SLCs

While the combined transcriptomics and metabolomics data allow for the generation of hypotheses on individual SLCs worth further investigation, we next explored the potential for more general observations. We expanded the analysis from individual SLC profiles to an SLC superfamily-wide analysis, aiming to identify metabolic and transcriptomic patterns among SLCs that would facilitate broad functional assignment through profile similarity. Employing hierarchical clustering, we grouped all overexpression cell lines based on the differential response of 14,187 genes and 142 metabolites between induced and uninduced samples. We selected

the number of clusters based on the mean silhouette width (Rousseeuw, 1987), aiming for an average cluster size of 5–10 cell lines (Fig. EV4A,B).

Metabolomic profiles of 378 SLC- and three GFP-expressing cell lines resulted in 48 clusters, based on their similarity in differential abundance of the 142 metabolites upon overexpression (Fig. 4A; Dataset EV6). Using a heatmap of the average metabolic profile per cluster, we identified key metabolites driving the similarities between clustered SLCs. This analysis led us to further explore selected clusters, their individual members, and the associated metabolite changes (Fig. EV2B). For instance, we found cluster 3 consisting of SLC22A6, SLC22A11, and SLC22A13, which all strongly reduced N-acetylglutamic acid levels (Fig. 4B). All three SLC22 members are annotated to transport urate and increased urate uptake is anticipated to inhibit N-acetylglutamic acid production due to urate's role as a N-acetylglutamate synthase inhibitor (Nissim et al, 2011), likely to explain the observed phenotype. Cluster 21 was characterized by the increase of taurocholic acid, an annotated substrate for all its members, SLC10A1, SLC10A2, and SLCO1B1 (Fig. 4C). Cluster 20 with SLC7A2 and SLC7A3 as members presented an example in which we observed similarity due to the differential signal of both the annotated substrate arginine as well as the metabolic products argininosuccinic acid and aminoadipic acid, which are downstream of arginine and lysine (another annotated substrate), respectively (Fig. 4D). Furthermore, hierarchical clustering of cluster-averaged metabolite profiles revealed co-regulation of several metabolite groups including nucleotide monophosphates, nucleotide triphosphates, and amino acids (Fig. EV4C). For instance, the SLCs within cluster 1 (SLC34A3, SLC34A1, SLC6A5, SLC36A2, SLC16A14, SLC6A19, SLC6A15) showed an upregulation of a group of metabolites consisting of isoleucine, tryptophan, tyrosine, histidine, leucine, citrulline, valine and methionine. Besides the upregulation of several amino acids, cluster 1 was characterized by a strong downregulation of taurine that likely stemmed from osmotic imbalance due to sodium-coupled transport, which had been reported for all SLCs in the cluster except SLC36A2 and SLC16A14 (Goldmann et al, 2025). Another commonality between the amino acid transporters of this cluster was their transport of glycine and/or proline, that may present an interesting starting point to study the regulatory relationships between amino acid transporters. Generally, these observations suggested that the grouping of SLCs by the gain-of-function-elicited metabolic fingerprint allowed for the identification of functional relationships.

In analogy to the analysis based on metabolites, we clustered SLCs based on their transcriptional profile. 447 SLC- and three GFP-expressing cell lines were clustered into 60 groups based on their similarity in the differential expression profile of 14,187 genes (Fig. 5A; Dataset EV7). To obtain a coarse overview of metabolic effects driven by each transcriptomic cluster, we calculated mean expression changes

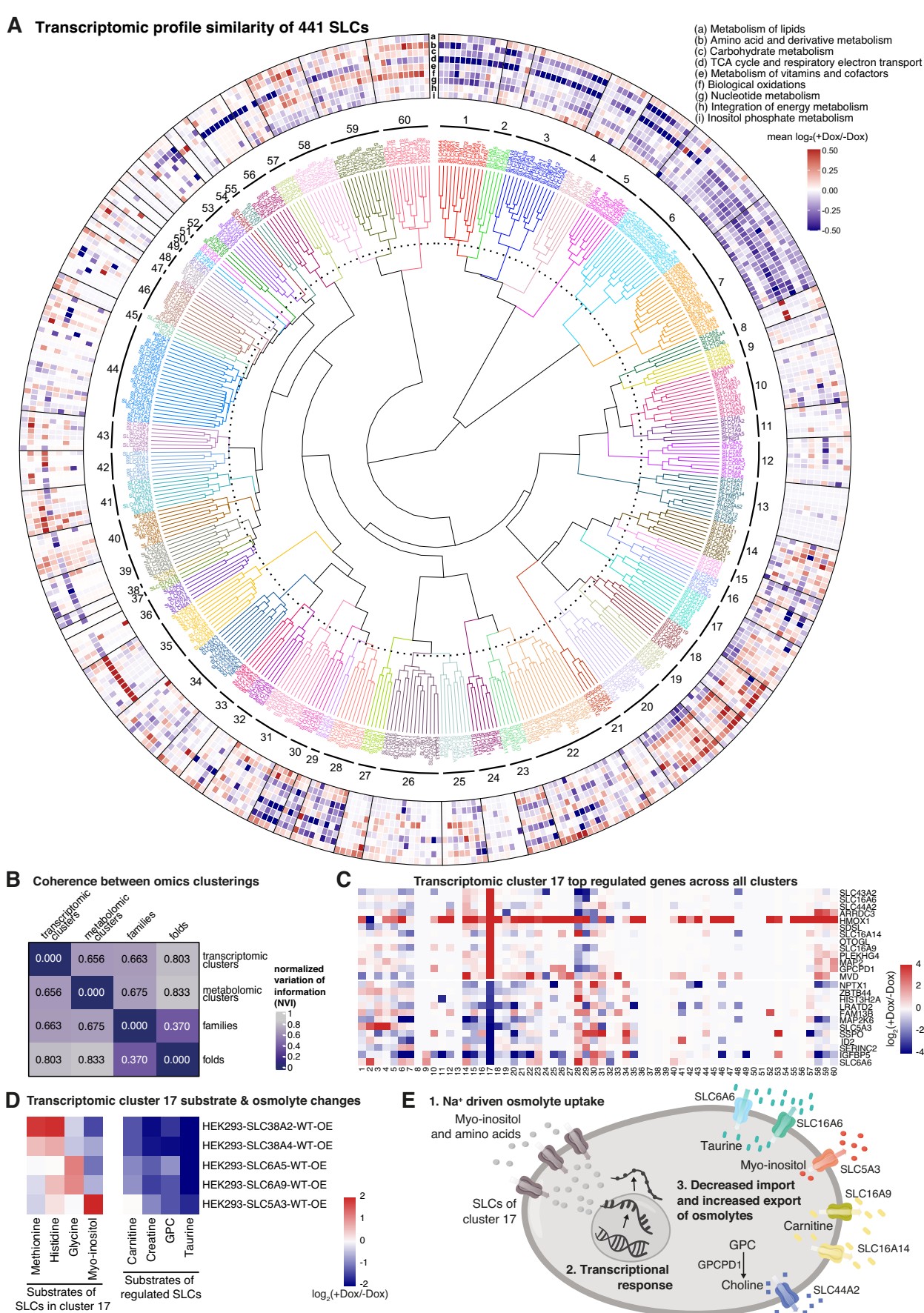

**A**  Transcriptomic profile similarity of 441 SLCs

(a) Metabolism of lipids
(b) Amino acid and derivative metabolism
(c) Carbohydrate metabolism
(d) TCA cycle and respiratory electron transport
(e) Metabolism of vitamins and cofactors
(f) Biological oxidations
(g) Nucleotide metabolism
(h) Integration of energy metabolism
(i) Inositol phosphate metabolism

mean log₂(+Dox/-Dox)

**B**  Coherence between omics clusterings

**C**  Transcriptomic cluster 17 top regulated genes across all clusters

**D**  Transcriptomic cluster 17 substrate & osmolyte changes

**E**  1. Na⁺ driven osmolyte uptake

2. Transcriptional response

3. Decreased import and increased export of osmolytes

**Figure 5.  Similarity clustering of transcriptomic profiles identifies differential metabolic impacts of SLC functional groups.**

(A) Dendrogram of hierarchical clustering based on the Euclidean distance matrix of standard-normalized transcriptional profiles of 450 analyses (transcriptomics data set of 441 SLC cell lines and 3 GFP cell lines; for 5 SLCs, two analyses with different induction times). The outer ring displays mean $\log_2$ fold change of metabolic genes in each of 9 top-level Reactome pathways. (B) Overview of coherence between transcriptomics clusters, metabolomics clusters, SLC families, and SLC structural folds. Coherence was quantified by calculating normalized variation of information (NVI) between different groupings, where a smaller NVI indicates higher coherence. (C) Heatmap showing doxycycline (Dox) vs uninduced average $\log_2$ fold changes in each cluster, for 25 most differentially expressed genes in cluster 17 compared to other clusters. Cluster numbers are indicated on $x$ axis. (D) Heatmap of Dox vs uninduced $\log_2$ fold changes of known substrates of cluster 17 members as well as substrates of SLCs and metabolic enzymes regulated by cluster 17. (E) Illustration of osmolyte uptake and transcriptional response to balance intracellular osmolyte concentrations and decrease osmotic pressure.

of genes associated with each Reactome pathway and examined the 9 top-level pathways (Fig. 5A, outer ring; Dataset EV8). Clusters 1–7 showed broad changes across metabolic pathways, with a general trend of decreased gene expression in these pathways. Other clusters had strong, specific signals for a single pathway, e.g., clusters 8, 35, 56 for biological oxidations and the TCA cycle. On the other hand, certain clusters did not appear to have strongly perturbed gene expression in any of the broad categories of metabolism, such as for example clusters 9 and 13.

We also evaluated each transcriptomic cluster for enrichment of shared functional properties of the overexpressed SLCs such as SLC family membership, structural fold, substrate class or subcellular localization (Appendix Fig. S2). In total, 13 clusters had at least one significantly enriched property; among these, cluster 17 (SLC6A19, SLC6A15, SLC6A12, SLC6A14, SLC6A9, SLC6A5, SLC5A3, SLC38A2, and SLC38A4) and cluster 6 (17 SLCs from diverse families) stood out with four enriched properties each (Fig. EV4D).

As a first step towards integrating metabolomics and transcriptomics data, we analyzed the coherence between SLC clusters derived from these two omics data sets, as well as their respective coherences with groupings based on SLC family and structural fold. Coherence was quantified by normalized variation of information (NVI), a distance metric for comparing clusterings (Meilă, 2007). Unsurprisingly, the lowest NVI (0.370), i.e., highest coherence, was obtained between SLC family and fold. The second-lowest NVI (0.656) was obtained between metabolomics and transcriptomics data clusterings, indicating that SLCs' effects on the metabolome and transcriptome tend to drive functional groupings that overlap between each other but are more distinct from structural relationships (Fig. 5B). We next examined the contribution of individual clusters to the overall NVI as well as mutual information (MI) between metabolomics and transcriptomics clusterings. Cluster contributions to NVI and MI were positively correlated with each other as well as with cluster size (Fig. EV4E). Clusters showing high mutual information contribute to the overall similarity between the clusterings, and clusters showing relatively low NVI contribute to a low overall distance between the clusterings. Combining both aspects allows identification of clusters representing potentially shared signatures in both metabolome and transcriptome profiles (Appendix Fig. S3). Metabolomics cluster 5 and transcriptomics clusters 6, 7, 17 were notable for high MI contributions relative to their contributions to NVI. We therefore focused on these clusters for further interpretation.

## An SLC-driven transcriptional response to altered intracellular osmolyte levels

Transcriptomic cluster 17 was significantly enriched for multiple functional properties: *sodium* (ion coupling), *SLC6* (family), *LeuT*

(fold) and *amino acids, peptides and derivatives* (substrate class) (Fig. 5B). SLCs in this cluster mostly transport amino acids and derivatives (glycine for SLC6A5 and SLC6A9, GABA and betaine for SLC6A12, and neutral amino acids for SLC38A2, SLC38A4, SLC6A14, SLC6A15, and SLC6A19) but curiously also included SLC5A3, which transports myo-inositol. Examination of the strongest gene expression changes in cluster 17 revealed that many of top differentially expressed genes unique to this cluster encode SLCs (henceforth referred to as a "regulated SLCs"): SLC5A3 (itself a cluster 17 member), SLC6A6, SLC16A6, SLC16A9, SLC16A14, SLC43A2 and SLC44A2 (Fig. 5C; Appendix Fig. S4). Hence, cluster 17 presented an example whereby the common metabolic effect of a group of SLCs led to a transcriptional response involving another group of SLCs. Substrates are known for SLC5A3 (myo-inositol), SLC6A6 (taurine), SLC16A9 (carnitine), SLC43A2 (neutral amino acids) and SLC44A2 (choline) (Goldmann et al, 2025). As previously noted, overexpression of the orphan SLC16A14 reduced intracellular carnitine, propionylcarnitine and butyrylcarnitine levels (Fig. 2D), compatible with a role in carnitine efflux. Although not demonstrated for human SLC16A6, the protein encoded by the mouse *Slc16a6* gene has been proposed to transport taurine (Higuchi et al, 2022).

Substrates of SLCs in cluster 17 and of the regulated SLCs all have roles as osmolytes (Burg and Ferraris, 2008; Steeves and Baltz, 2005; Bussolati et al, 2001). We hypothesized that overexpression of SLCs in cluster 17 increased intracellular accumulation of their respective osmolyte substrates, leading to activation of a common transcriptional response aimed at reducing intracellular osmolyte levels. This reduction was mediated by a series of changes: decrease in intracellular taurine by downregulation of taurine importer SLC6A6 and upregulation of taurine exporter SLC16A6 as well as decreases in intracellular carnitine likely to be caused by upregulation of carnitine exporter SLC16A9 and the putative acylcarnitine exporter SLC16A14. The likelihood of this interpretation is supported by the observed upregulation of the metabolic enzyme glycerophosphocholine phosphodiesterase 1 (GPCPD1), that converts glycerophophorylcholine (GPC) to choline. Upregulation of both GPCPD1 and SLC44A2 promotes GPC catabolism and choline export. Additionally, upregulation of SLC43A2, an amino acid uniporter, is expected to reduce excess intracellular amino acid levels.

To validate the above hypothesis, we performed targeted LC–MS/MS measurements of (1) substrates of cluster 17 members (SLC5A3, SLC6A5, SLC6A9, SLC38A2, SLC38A4) and (2) substrates or directly connected metabolites of the regulated SLCs (SLC5A3, SLC6A6, SLC16A6, SLC16A9, SLC16A14, SLC43A2, and SLC44A2), that were not measured in the original metabolomics panel. We confirmed that known substrates of cluster members

accumulated significantly upon doxycycline induction of the respective cell line: myo-inositol for SLC5A3, glycine for SLC6A5 and SLC6A9, and histidine as well as methionine for SLC38A2 and SLC38A4 (Fig. 5D). Substrates of the regulated SLCs were reduced too, aligning with a transcriptional program aimed at decreasing intracellular osmolyte levels: taurine (SLC6A6 and SLC16A6), carnitine and creatine (SLC16A9), and GPC (SLC44A2). Although regulation of transporter activity often resides on important translational and posttranslational mechanisms, these findings suggested that it was also possible to interpret changes in mRNA levels as part of metabolic regulatory circuits. In summary, SLCs in cluster 17 induced a transcriptional response to hypoosmotic stress, manifested in regulation of SLCs and other genes associated with reduction in intracellular osmolyte levels (Fig. 5E). This highlighted how observed transcriptional changes corresponded to metabolite changes, supporting the notion that the transcriptomics data set extended our coverage of metabolic pathways. The approach addresses properties in the regulation of metabolism that are complementary to those that can be inferred by our targeted metabolomics approach.

## A group of SLCs impacting the cellular glycosylation machinery

The enrichment analysis for functional properties revealed that transcriptomic clusters 6 and 7 were significantly enriched with members belonging to SLC35 and SLC30 families, respectively, as well as DMT fold and Golgi localization (Fig. EV4D). Due to the high similarity in the enrichment of functional properties between these two adjacent clusters, we considered them as a larger cluster of SLCs with a common transcriptomic effect. The SLC35 family represents the major family of endoplasmic reticulum (ER)- and Golgi-localized nucleotide sugar transporters that govern the availability of substrates for glycosylation reactions and carry the DMT protein fold (Ferrada and Superti-Furga, 2022). In agreement with the enriched functional properties, gene set enrichment analysis (GSEA) on differential expression profiles of metabolic genes for this transcriptomic cluster revealed GO Molecular Function terms related to glycosylation, including glycosyltransferase and hexosyltransferase activity, as well as Golgi-associated GO Cellular Compartment terms (Fig. 6A, upper right).

Inspection of the mutual information contributions from metabolomics versus transcriptomics clusters showed that transcriptomic clusters 6 and 7 shared high mutual information with metabolomic cluster 5 (Appendix Fig. S3A), consistent with a high overlap between members of the respective clusters (Fig. 6A, upper left). Differential metabolite abundance profiles of SLCs from metabolomics cluster 5 showed increased N-acetylglucosamine phosphate (GlcNAc-P) and N-acetylneuraminic acid (Neu5Ac), as well as decreased uridine triphosphate (UTP) and cytidine triphosphate (CTP) (Fig. 6A, lower left). GlcNAc-P and Neu5Ac belong to the connected metabolic pathways of hexosamine and sialic acid biosynthesis, which are linked to nucleotide metabolism through consumption of UTP and CTP to produce nucleotide-sugars UDP-N-acetylglucosamine (UDP-GlcNAc), UDP-N-acetylgalactosamine (UDP-GalNAc) and CMP-N-acetylneuraminic acid (CMP-Neu5Ac) that are substrates for glycosylation reactions in the ER and Golgi (Fig. 6B). Taken together, SLCs from transcriptomic clusters 6 and 7 as well as metabolomic cluster 5 (thereafter referred

to as the "glycosylation-related cluster") displayed a common gene expression signature and altered metabolism related to altered cellular glycosylation.

As seen from the enrichment analysis of functional properties, the glycosylation-related cluster consisted predominantly of Golgi-localized SLCs with known roles in transport of nucleotide sugars or metal ions, which are also known to affect glycosylation-related processes (Durin et al, 2023). An orphan SLC of unknown function, MFSD5 was also found in this cluster. To obtain an overview of specific glycosylation changes and gain insight into potential glycosylation-related functions, we selected MFSD5 for glycoproteomic analysis, along with two other SLCs from the cluster: TMEM241, recently identified as a Golgi UDP-GlcNAc transporter; and SLC17A9, a vesicular ATP transporter (Zhao et al, 2023; Sawada et al, 2008). Multiplexed mass spectrometry glycoproteomics identified 11,938 glycopeptides corresponding to 1093 proteins. The most abundant glycoproteins were enriched for the GO Cellular Compartment terms associated with secretory and endolysosomal pathways, as well as the GO Biological Process terms for vesicle-mediated transport, endocytosis and amide transport (Fig. EV5A). Principal component analysis of glycopeptide abundance profiles clearly separated Dox-induced from uninduced samples (Fig. EV5B). The detected glycopeptides spanned a wide range of 325 different glycan compositions, from the simplest O- or N-linked HexNAc(1) i.e., a single N-acetylglucosamine or N-acetylgalactosamine linked to serine/threonine or asparagine, to complex structures containing HexNAc, Hex (mannose, galactose or glucose), Fuc (fucose), NeuAc (N-acetylneuraminic acid). The distribution of glycopeptides across these 325 compositions was highly skewed, with 6 glycan compositions accounting for >50% of the total abundance signal in MFSD5 samples (7 overall for MFSD5, TMEM241, SLC17A9 combined) (Figs. 6C and EV5C) as well as >40% of the distinct glycopeptide species (Fig. EV5D). These highly abundant glycan compositions consisted of N-linked HexNAc(2)Hex(5–10), likely representing high-mannose N-glycan structures GlcNAc(2)Man(5–10) that reflect a minimally processed state following protein transit from the ER to Golgi; and O-linked HexNAc(1) which likely represents O-linked GalNAc, the starting point of mucin-type O-glycan synthesis. Across MFSD5, TMEM241 and SLC17A9 WT-OE cell lines, significant Dox-induced changes were almost exclusively in the positive direction (i.e. increases) for these high-abundance glycan compositions (Figs. 6C and EV5C).

To validate the glycoproteomic results of increased O-linked GalNAc and high-mannose N-glycans, we performed flow cytometry upon staining with *Vicia Villosa* Lectin (VVL), which binds terminal GalNAc, and *Galanthus Nivalis* Lectin (GNL), which specifically binds terminal mannose (Bojar et al, 2022). SLC35D3 and SLC35E4, which are not part of the glycosylation-related cluster were included as Golgi- and ER-localized reference controls, respectively. Cell surface VVL staining and intracellular GNL staining increased significantly upon overexpression of MFSD5 but not SLC35D3 or SLC35E4 (Fig. 6D,E). O-linked GalNAc is also known as Tn antigen, a recognized tumor marker (Chia et al, 2016) and we confirmed a corresponding Dox-induced increase in staining by the specific anti-Tn antibody 5F4 (Fig. 6F). We also further investigated intracellular distribution of GNL-stained high-mannose proteins by immunofluorescence confocal microscopy. In uninduced MFSD5 WT-OE cells, GNL-stained

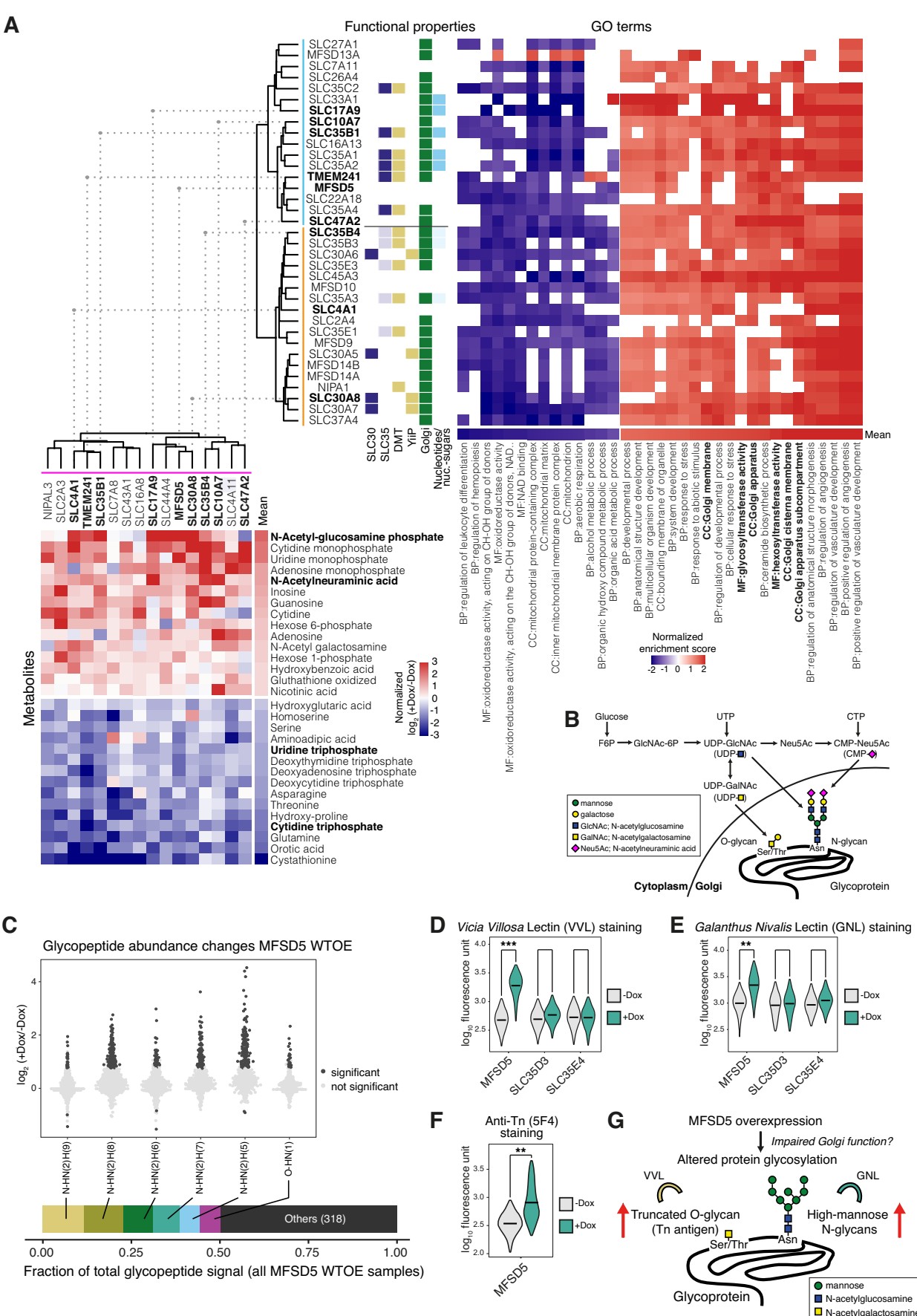

**Figure 6.   Multi-omics-based identification of an SLC cluster with effects on cellular N- and O-linked glycosylation.**

(A) Transcriptomics clusters 6 and 7 (left) and metabolomics cluster 5 (right) showing common SLCs between clusters from the two different data modalities. Upper right: Transcriptomic clusters 6 and 7 (thereafter referred to as combined "glycosylation-related cluster") with significantly enriched functional properties (Fisher's test $P < 0.2$) and heatmap of normalized enrichment scores from gene set enrichment analysis (GSEA) on $\log_2$ expression fold changes (+Dox/−Dox) of metabolic genes. Only GO terms significant ($P_{adj.}$ <0.05) from the mean $\log_2$ fold change profile are shown. Lower left: Heatmap of metabolomics data showing normalized doxycycline (Dox) vs uninduced $\log_2$ fold changes (+Dox/−Dox) for SLCs in glycosylation-related cluster. Only metabolites with 15 highest and lowest average $\log_2$ fold changes are shown. (B) Schematic representation of the hexosamine and sialic acid biosynthetic pathway that produces nucleotide sugar substrates for glycosyltransferase reactions in the Golgi. (C) Dox-induced vs uninduced glycopeptide abundance $\log_2$ fold changes for 6 most abundant glycan compositions across MFSD5-WT-OE samples. Each data point represents a glycopeptide abundance comparison between +/− Dox samples ($n = 3$). Individual glycopeptides are plotted as discrete points and color-coded by significance. Bottom: stacked barplot indicating fraction of total glycopeptide signal contributed by each glycan composition, showing that these six compositions account for over 50% of the total glycopeptide signal. (D) Cell surface Vicia Villosa Lectin (VVL) staining of +/− Dox cells following primary staining with biotinylated lectin and secondary staining with streptavidin-Alexa Flour 647. Each violin plot represents flow cytometry measurements of at least 30,000 cells pooled from three replicate wells, and horizontal bisecting lines indicate population geometric means. Effect sizes (Cohen's $d$) between uninduced and Dox-induced populations of each cell line are indicated by annotated brackets as follows: *$d > 0.5$; **$d > 1$; ***$d > 2$. The Cohen's $d$ values are: MFSD5, 1.922; SLC35D3, 0.325; SLC35E4, −0.0246. (E) Intracellular GNL staining of +/− Dox cells following permeabilization, primary staining with biotinylated lectin and secondary staining with streptavidin-Alexa Flour 647. Each violin plot represents flow cytometry measurements of at least 30,000 cells pooled from 3 replicate wells, and horizontal bisecting lines indicate population geometric means. Effect sizes (Cohen's $d$) between uninduced and Dox-induced populations of each cell line are indicated by annotated brackets as follows: *$d > 0.5$; **$d > 1$; ***$d > 2$. The Cohen's $d$ values are: MFSD5, 1.197; SLC35D3, 0.115; SLC35E4, 0.306. (F) Anti-Tn antibody staining of MFSD5 +/− Dox cells following primary staining with 5F4 and secondary staining with anti-mouse IgM Alexa Fluor 568. Each violin plot represents flow cytometry measurements of at least 30,000 cells pooled from 3 replicate wells, and horizontal bisecting lines indicate population geometric means. The bracket annotated with ** denotes an effect size (Cohen's $d$) >1; the value for MFSD5 is 1.234. (G) Schematic illustration of effects on N- and O-linked glycosylation by overexpression of MFSD5 as validated by lectin staining.

sharp punctate loci, which overlapped with the lysosomal marker LAMP1 indicating that GNL predominantly stained lysosome-targeted proteins. This was confirmed by colocalization of GNL staining with M6P-1, which detects the mannose-6-phosphate (M6P) modification that marks soluble proteins for lysosomal delivery (Müller-Loennies et al, 2010) (Fig. EV5E). Upon Dox induction, GNL and M6P-1 staining became diffused, suggesting that MFSD5 overexpression impacted M6P-dependent lysosomal targeting. Taken together, elevated levels of truncated O-glycans (reflecting either aberrant O-glycan initiation or failure to extend O-glycans) and high-mannose N-glycans (reflecting failure to trim mannose residues from nascent N-glycans and re-extend them with other sugar moieties) suggested that MFSD5 overexpression elicits a disruption to Golgi function.

Besides MFSD5, we tested the SLCs found in both the transcriptomic glycosylation cluster and the metabolomic cluster 5 (Fig. 6A, SLCs highlighted in black) as well as several additional cell lines from the transcriptomic cluster. VVL staining increased substantially upon Dox induction (Cohen's $d > 0.5$) for 12 out of 14 cell lines in the panel, excluding the reference controls SLC35D3 and SLC35E4 (Fig. EV5F; Table EV1), while GNL staining increased above the same threshold ($d > 0.5$) for 8 out of 14 cell lines (Fig. EV5G; Table EV2). Closer examination of SLC47A2 and SLC30A8 revealed bimodal distributions in VVL staining signal and the presence of subpopulations with substantially elevated signal (Fig. EV5H). This may suggest that these two SLCs affect glycosylation in a stochastic manner in only a subset of cells. In a transport-deficient mutant of SLC30A8 (Azzollini et al, 2024), the subpopulation with increased VVL staining was absent (Fig. EV5I). In line with this result, the transcriptional profile of the SLC30A8 mutant was distinct from the SLCs of the glycosylation-related cluster (Fig. EV5J), providing evidence that the transcriptional and glycosylation effects of SLC30A8 are dependent on transport functionality.

In summary, our family-wide omics data enabled the assignment and subsequent validation of a glycosylation-related function to the orphan SLC MFSD5 using the guilt-by-association principle. MFSD5 along with other SLCs in the cluster drove a common glycosylation effect characterized by increased O-linked GalNAc

and N-linked high-mannose glycans (Fig. 6G). The commonality of a group of SLCs with diverse annotated functions but sharing a metabolomic and gene expression signature, as well as effect on cellular glycosylation, offers avenues for future studies and mechanistic exploration.

# Discussion

By size, implication in human disease, and potential pharmacological exploitation, the target class of solute carriers (SLCs) compares well to GPCRs, kinases, proteases and ion channels (César-Razquin et al, 2015; Oprea et al, 2018). Most of the functional studies that focus on a whole target class using CRISPR/Cas9 or siRNA use one specific readout or a few readouts in parallel (DelRosso et al, 2023; Timms et al, 2023). Less frequently, unbiased functional readouts with hundreds of genes of related function or structure are employed (Liu et al, 2023; Johnson et al, 2023). This work provides a first, although coarse, functional annotation by multi-omics of the entire SLC superfamily under comparable conditions. The parallel evaluation of changes in metabolites and transcripts across hundreds of isogenic cell lines each bearing altered expression of only one gene, offers data for hundreds of SLC transporters for which no functional annotation has been available.

Selecting the cell line models was a major decision at the consortium level, aiming for a balance between physiological relevance, i.e., endogenous expression of the individual SLC gene, technical feasibility regarding the generation of cell lines, suitability for a commensurate data acquisition throughput, time, costs, and robustness across experimental locations. We selected the colon cancer cell line HCT 116 as our primary cell model because of its biological relevance in the transporter field and technical feasibility for large-scale genome engineering. For SLCs that are not expressed in HCT 116 or only at very low levels, compared to the other cancer cell lines in our panel, we chose a different cell model accordingly to provide a biologically more relevant model system (Fig. EV1A,B). For our SLC perturbation cell lines we chose SLC knock-out clones over knock-down pools to directly attribute observations to gene loss and ensure consistency across multiple measurements and

assays. Using multiple clones for each knock-out cell line and considering clonal effects within the statistical analysis of metabolomics data mitigated clonal variance. Additionally, we selected the Jump-In™ T-REx™ HEK293 cell line model to create a collection of overexpression cell lines for the entire SLC superfamily in an isogenic background. Overexpression models allowed rapid SLC expression changes in a loss-of-function background, minimizing the kind of metabolic adaptation and possible genetic drift associated with the generation of cell lines. Nonetheless, protein overexpression can present serious drawbacks such as overload of the cellular machinery and functional consequences unrelated to physiological function. Some of the transcriptional programs we observe are likely elicited by protein stress.

This study represents a first-pass survey of metabolic consequences of SLC function across the entire SLC superfamily, and several caveats must be observed. Nearly a quarter of the SLC superfamily were not studied in their natural cellular context, potentially affecting their functionality. However, we found literature-annotated substrates independently of whether an SLC was endogenously expressed in the given cell line. Using different cell line models each with their own standard culture conditions further challenged the complex metabolomics data analysis. A further constraint of our study was the necessity to limit to only standard culture conditions due to the already large sample number, fully aware that a unified standard media would unlikely represent optimal transport conditions for the different SLCs (Cantor et al, 2017). Without any doubt, future dedicated studies in more physiological settings will unravel subtleties and specificities not accessible in the present large-scale set-up.

We can consider targeted metabolomics as a quantitative "fingerprinting" approach that enables profiling and comparison at such a large scale. At the same time, metabolite coverage is limited and obviously fails to capture the full diversity of substrates transported by the SLC superfamily, or metabolic derivatives thereof. In addition, detection of a substrate is affected by multiple biological factors, such as transport kinetics, expression of the SLC, functional redundancy, metabolic conversion of the substrate, and metabolic need, which itself depends on cell type and cellular state. For example, it is striking that only metabolite decreases were detected in SLC16A14 (Fig. 2F). It is possible that SLC16A14 indeed mediates strong efflux of one or more molecules without a compensatory increase in other metabolites, but equally possible also is that there are compensatory metabolite increases that escaped detection due to limited coverage by our targeted metabolomics method. Since most metabolites are not significantly changed, it is less likely that overall metabolite levels were decreased due to less input material (e.g., from decreased viability/increased cell death, decreased cell volume etc), although this possibility could not be ruled out as we did not quantify the number of input cells separately for Dox-induced and uninduced conditions in our workflow. SLC35F6 is another example—although SLC35 is commonly known as a family of nucleotide sugar transporters, the nucleotide sugars detected by our targeted method uridine diphospho-glucose and uridine diphospho-galactose (collectively detected as uridine diphosphohexose) did not show significant change (Fig. 2G), and we regrettably did not cover the remaining seven known nucleotide sugars with our method despite the large number of SLC35 family members. However, SLC35 family members are also known to mediate transport of other compounds (for example, the anticancer drug YM155 by SLC35F2 (Winter et al, 2014)), and the metabolic changes observed for SLC35F6 may reflect transport of a novel compound that affects osmolyte and amino acid uptake/metabolism. This highlights the difficulty of setting up a targeted metabolomics panel that adequately covers all potential substrates of interrogated SLCs.

Despite these limitations, we found changes in annotated substrates or direct metabolic conversions for 102 SLCs (Fig. 1E,F; Dataset EV3). Many of the remaining significant metabolite hits might represent unannotated true substrates, and we anticipate that our study will accompany follow-up validation efforts that identify novel substrates for both orphan and well-studied SLCs. As clearly seen from our deorphanization efforts for SLC45A4, the initial metabolic insights provided by this data set can guide and motivate investigation into relevant metabolic pathways, but complementary assays to interrogate the remaining metabolic space will likely be required. We can also consider the metabolomics data presented by this study as providing an overview of the metabolic effects caused by SLC expression and insight on the cellular control of metabolite levels. The magnitude of changes across more than 370 cell lines can indicate a window of tolerance for individual metabolite fluctuations that might be tighter or looser (Fig. EV2A), possibly connecting to a metabolite's regulatory role in feedback or feedforward mechanisms. The data set offers the unique possibility to study correlations in metabolite abundance upon hundreds of perturbations in isogenic settings, perhaps leading to new insights into the plasticity of cellular metabolism in human cells.

Investigating functional profiles of individual SLCs revealed a novel role for SLC45A4 in mediating cellular export of polyamines, including putrescine and spermidine. So far, there is no annotated substrate for human SLC45A4, and evidence is scarce for the other three members of the SLC45 family. The heterologous expression of mouse *Slc45a2-4* increased sucrose uptake in yeast cells (Bartölke et al, 2014). In contrast, we did not observe increased sucrose uptake in our SLC45A4 overexpressing cells (Fig. EV3C). Although we could not rule out the possibility of sucrose uptake followed by rapid flux into glycolysis, this scenario is unlikely because sucrase-isomaltase (SI), the only known mammalian sucrose metabolic enzyme, is not expressed in HEK293 or HCT 116 cells (data from (Goldmann et al, 2025); ENA Project number PRJNA545487). Future uptake experiments using stable isotopically labeled sucrose will provide stronger confirmation whether sucrose is indeed a substrate of SLC45A4. Regardless of whether SLC45A4 mediates sucrose uptake, the polyamine efflux observed from SLC45A4 overexpressing cells is unlikely to be connected to sucrose transport, since (1) putrescine export occurs in serum-free media which does not contain even the trace amounts of sucrose potentially present in serum (Fig. 3D), and (2) our cell-free proteoliposome uptake assay demonstrating direct putrescine transport by SLC45A4 (Fig. 3F). While our proteoliposome uptake assay clearly deorphanizes SLC45A4 as a putrescine transporter, further understanding SLC45A4's contribution to cellular putrescine transport requires knowing the relative expression and selectivity of the various transporters in each tissue. These will hopefully be addressed in future studies with even higher throughput.

For the identification of functional relationships between SLCs, we tested several strategies with varied sample normalization, clustering method, and gene/metabolite sets, using the mean cluster stability and enrichment of functional SLC properties for evaluation. A limiting factor in the definition of functional clusters by

metabolic profiles was the sparseness of the data set (142 metabolites, 378 SLCs) as well as the relatively high level of noise, as no threshold for significance was set. We did not face these limitations in the clustering of transcriptional profiles and therefore used transcriptomic clusters as starting points for investigations and metabolomic analyses of the respective groups of SLCs for complementation and refinement. This underscores the importance of considering complementary sets of information to identify and validate clustering patterns effectively.

From transcriptional profiling of 441 SLCs, we identified cluster 17 of $Na^+$/osmolyte symporters with a shared regulation of osmolyte transporters, likely aimed at balancing intracellular osmolyte levels and relieving hypoosmotic stress. We speculate that $Na^+$ is the driving force in the accumulation of osmolytes by these $Na^+$-dependent cotransporters, since DMEM (Sigma) contains a high concentration of $Na^+$ ions (154 mM total $Na^+$ from sodium chloride and sodium bicarbonate). The intriguing question arises as to why other $Na^+$-dependent transporters of potential osmolytes, such as SLC6A6 (taurine), SLC6A8 (creatine), and SLC22A5 (carnitine), were not found in the same cluster. Among many possible explanations, we suspect that differences in substrate levels in the medium may be the cause: base DMEM contains 400 μM glycine, 200 μM of both histidine and methionine, and 40 μM myo-inositol, but no taurine, creatine or carnitine. Targeted metabolomics analysis of our culture media showed low levels of taurine (<10 μM) likely from fetal bovine serum, while creatine and carnitine were below the limit of detection (Dataset EV9). Therefore, substrate levels may be too low for substantial intracellular accumulation and subsequent induction of osmotic stress response upon SLC6A6, SLC6A8, or SLC22A5 overexpression. Under appropriate media conditions, it is likely that the transcriptional response described for cluster 17 will be elicited by even more SLCs transporting osmolyte substrates.

Among the cluster 17 regulated SLCs, taurine transporter SLC6A6 and myo-inositol transporter SLC5A3 are regulatory targets of the tonicity-responsive enhancer binding protein (TonEBP), a transcriptional regulator also known as NFAT5 (Burg and Ferraris, 2008). Other TonEBP-activated genes include SLC6A12 (betaine/GABA transporter 1), aldo-keto reductase family 1 member B (AKR1B1, functioning in sorbitol synthesis), patatin like phospholipase domain containing 6 (PNPLA6, functioning in GPC synthesis), heat shock protein family A member 2 (HSPA2), and insulin-like growth factor binding protein 5 (IGFBP5) (Choi et al, 2020; Lee et al, 2011). The downregulation of SLC6A6, SLC5A3 as well as IGFBP5 by cluster 17 (Figs. 5C and EV5E) is consistent with decreased TonEBP expression and activation of target genes under hypotonic conditions, as previously reported (Woo et al, 2000). On the other hand, the mechanism behind the upregulation of the osmolyte exporters SLC16A6, SLC16A9, SLC16A14 and SLC44A2, as well as GPCPD1, is currently unknown. In addition to its well-studied role in transcriptional activation, NFAT5 has been reported to suppress gene expression through direct DNA-binding mechanisms (Huerga Encabo et al, 2020; Lee et al, 2019). Although beyond the scope of the present paper, it would be intriguing to investigate whether upregulation of cluster 17 regulated SLCs also occur through TonEBP or another, novel mechanism.

We further identified an apparently diverse group of SLCs, including the UDP-GlcNAc transporter TMEM241, vesicular ATP transporter SLC17A9, zinc transporter SLC30A8, several members of the SLC35 family and the orphan MFSD5, with a shared effect on perturbing the cellular glycosylation machinery. It is not surprising that overexpression of SLC35 family members impact glycosylation, given their central role in transport of nucleotide sugars into the ER and Golgi for glycosyltransferase reactions (Hadley et al, 2019). Less recognized are the links between vesicular ATP transport (SLC17A9) or zinc transport by SLC30 family members, of which four are present in the cluster (SLC30A5–8). The SLC30 family is primarily involved in the transport of zinc out of the cytoplasm, either into the extracellular space or intracellular compartments (Chen et al, 2024). Overexpression of SLC30 family members possibly leads to $Zn^{2+}$ accumulation in the secretory pathway which may perturb glycosylation (Rømer et al, 2023). Given the important role of SLC17A9-mediated ATP transport in the secretory and endolysosomal pathways, both closely connected to the Golgi, where glycan biosynthesis and breakdown takes place, SLC17A9 overexpression may impact the crosstalk between these cellular compartments. The orphan transporter MFSD5 shows high structural similarity to SLC17A9 (Ferrada and Superti-Furga, 2022) and was reported to bind ATP (Jelcic et al, 2020), while plant homologs of this SLC are reported as Golgi-localized transporters of S-adenosylmethionine (Temple et al, 2022). Based on these observations, we speculate that MFSD5 may mediate transport of ATP or other adenosyl compounds, although this hypothesis remains to be tested.

Common glycosylation changes by this group of SLCs include elevated high-mannose N-glycans and Tn antigen levels. Increased high-mannose N-glycans are frequently upregulated in cancer and correlate with cancer grade (Chatterjee et al, 2021), while the Tn antigen is a well-recognized oncofetal antigen expressed in a large proportion of solid tumors including 70–90% of breast colon, lung, bladder, cervix, ovary, stomach, pancreatic, and prostate cancers (Ju et al, 2011; Chia et al, 2016; Mereiter et al, 2016) and supports tumor progression through immunomodulatory interactions (van Vliet et al, 2006). The mechanisms driving these cancer-associated glycosylation changes are not fully understood but hold potential for therapeutic intervention (Chia et al, 2016; Mereiter et al, 2019). We anticipate that the mechanistic elucidation of the SLC-driven glycosylation changes discovered in this study is expected to provide new insights into the roles of transporters in glycosylation and may reveal new biomarkers or targets for cancer therapy.

Despite our success in ascribing shared functions to groups of SLCs and in annotating novel functions to understudied SLCs based on the guilt-by-association principle, a clear shortcoming of this study is the lack of follow-up experiments to elucidate detailed mechanisms or investigate the physiological relevance of our findings in disease models. Yet, the analyses conducted serve as exemplary explorations of this rich resource, providing a foundation for numerous future studies. Given the significance of metabolism in physiology and disease, understanding the roles of individual metabolites, their regulation, and their function as regulators in an integrated manner is paramount. SLCs are attractive pivotal players to modulate metabolism as they often represent accessible "gates" and are eminently druggable (Lin et al, 2015; Wang et al, 2020; Dvorak and Superti-Furga, 2023).

The combination of metabolomic and transcriptomic data sets holds considerable promise for future research, once the connections between metabolism, transcription, and expression are better understood. We anticipate that these data sets will serve as valuable resources for understanding metabolic conditions in disease

contexts and the compensatory interactions influencing disease. This work together with the accompanying studies on the SLC knowledgebase (Goldmann et al, 2025), the SLC genetic interactions (Wolf et al, 2025) and the SLC protein-protein interactions (Frommelt et al, 2025) is a series of four publications which presents the largest study on the SLC superfamily to date and possibly a unique case of concerted functional assignment to a large superfamily of human genes. The underlying reagents and data sets are publicly accessible through a web portal and public repositories (https://re-solute.eu/; https://www.addgene.org/depositor-collections/re-solute/; https://www.atcc.org/cell-products/cell-models/solute-transporter-carrier-cells).

# Methods

## Reagents and tools table

| Reagent/resource | Reference or source | Identifier or catalog number |
| --- | --- | --- |
| **Experimental models** | | |
| HCT 116 cells | ATCC | RRID:CVCL_0291 |
| LS180 cells | ATCC | RRID:CVCL_0397 |
| Jump-In™ T-REx™ HEK293 cells | Thermo Fisher Scientific | RRID:CVCL_YL74 |
| Huh-7 cells | ATCC | RRID:CVCL_0336 |
| 1321N1 cells | ATCC | RRID:CVCL_0110 |
| SK-MEL-28 cells | ATCC | RRID:CVCL_0526 |
| Expi293F cells | Thermo Fisher Scientific | A14527 |
| **Recombinant DNA** | | |
| pLentiCRISPRv2 | Addgene | #52961 |
| Gateway™ pDONR™221 Vector | Thermo Fisher Scientific | 12536017 |
| pDONR221 entry clones | https://www.addgene.org/depositor-collections/re-solute/ | #131865–132311 |
| pCW57.1_TStrp_HA_Ct | Addgene | #194066 |
| pCW57.1_TStrp_HA_Nt | Addgene | #194065 |
| pJTI™ R4 CMV-TO MCS pA vector | Thermo Fisher Scientific | A15004 |
| **Antibodies** | | |
| Rat anti-HA antibody | Roche | 11867423001 |
| Rabbit anti-LAMP1 antibody | Cell Signaling Technology | #9091 |
| Biotinylated GNL | Vector Labs | B-1245-2 |
| Biotinylated VVL | Vector Labs | B-1235-2 |
| Mouse M6P-1 antibody | ABCD Antibodies | ABCD_AG949 |
| Anti-mouse Alexa Fluor 568 | Thermo Fisher Scientific | A-11004 |
| Anti-rabbit Alexa Fluor 488 | Thermo Fisher Scientific | A-11008 |
| Anti-Tn antibody (5F4 clone) | Ullu Mandel and Henrik Clausen (University of Copenhagen) | |

| Reagent/resource | Reference or source | Identifier or catalog number |
| --- | --- | --- |
| **Oligonucleotides and other sequence-based reagents** | | |
| gRNAs targeting SLCs | This study | See Table EV1 |
| NEBNext Multiplex Oligos for Illumina | New England Biolabs | #E7600 |
| **Chemicals, enzymes, and other reagents** | | |
| RPMI medium | Sigma | R8758 |
| DMEM medium | Sigma | D5796 |
| Fetal Bovine Serum | Gibco | 10270-106, Lot 42F8381K |
| Penicillin-Streptomycin | Gibco | 15140-122 |
| Doxycycline | Sigma-Aldrich | D9891-1G; CHEBI:50845 |
| PBS buffer | Sigma-Aldrich | D8537 |
| Poly-L-lysine | Sigma-Aldrich | P6282 |
| Ammonium carbonate | Sigma-Aldrich | 207861 |
| Methanol | Thermo Fisher Scientific | M-4000-17 |
| cOmplete Protease Inhibitor Cocktail tablets | Roche | 11836170001 |
| Dodecylmaltoside (DDM) | Anatrace | D310 |
| Cholesterol hemisuccinate (CHS) | Anatrace | CH210 |
| Strep-Tactin XT Superflow resin | IBA-Lifesciences | 2-4010-010 |
| TALON resin | Takara | 635504 |
| Soy phospholipids | Sigma-Aldrich | 11145 |
| Triton X-100 | Sigma-Aldrich | T8787 |
| TMTpro 18 plex | Thermo Fisher Scientific | A52045 |
| DTT | Roche | 10708984001 |
| IAA | Sigma-Aldrich | I6125 |
| Lys C | FUJIFILM Wako | 125 05061 |
| Trypsin Gold | Promega | V5280 |
| NEBNext Ultra II RNA Library Prep Kit | New England Biolabs | #E7760 |
| NEBNext Poly(A) mRNA Magnetic Isolation Module | New England Biolabs | #E7490 |
| Ampure XP beads | Beckman Coulter | A63880 |
| PhiX Control v3 | Illumina | |
| RNeasy 96 Kits | Qiagen | 74181 |
| QuantiFluor RNA System | Promega | PAE3310 |
| Sep-Pak cartridges | Waters | WAT054945 |
| TSKgel Amide-80 column | Sigma-Aldrich | 813071 |
| Microspin™ G-50 Columns | VWR | 27-5330-01 |
| **Software** | | |
| PeakBotMRM | https://github.com/christophuv/PeakBotMRM | |

| Reagent/resource | Reference or source | Identifier or catalog number |
|---|---|---|
| R | CRAN | Version 4.3.3 |
| Python | | Version 3.7 |
| Snakemake | | Version 6.6.1 |
| cutadapt | | Version 2.8 |
| STAR | | Version 2.7.9a |
| RSEM | | Version 1.3.2 |
| FastQC | | Version 0.11.9 |
| RSeQC | | Version 4.0.0 |
| MultiQC | | Version 1.11 |
| DESeq2 | Bioconductor | Version 1.42.1 |
| clusterProfiler | Bioconductor | Version 4.12.6 |
| FlowJo | BD Biosciences | v10.10.0 |
| Xcalibur | Thermo Fisher Scientific | v3.1 |
| Byonic | Protein Metrics | |
| Proteome Discoverer | Thermo Fisher Scientific | v3.0 |
| **Other** | | |
| 1290 Infinity II UHPLC system | Agilent Technologies | |
| ZORBAX RRHD Extend-C18 column | Agilent Technologies | 759700-902 |
| SecurityGuard ULTRA Cartridge UHPLC C18 | Phenomenex | |
| ACQUITY UPLC HSS T3 column | Waters | 186003539 |
| ACQUITY UPLC BEH HILIC column | Waters | 186004801 |
| ACQUITY UPLC BEH Amide column | Waters | 186003461 |
| 6470 Triple Quadrupole Mass Spectrometer | Agilent Technologies | |
| LSR Fortessa flow cytometer | BD Biosciences | |
| LSM 700 confocal microscope | Zeiss | |
| NovaSeq 6000 Sequencing System | Illumina | |
| Biomek i7 Hybrid workstation | Beckman Coulter | |
| Liposome Extruder | Avestin | LF-1 |

## Cell line generation

HCT 116 (RRID:CVCL_0291) and LS180 (RRID:CVCL_0397) cell lines were cultured in RPMI (R8758 Sigma) + 10% Fetal Bovine Serum (10270-106, Lot 42F8381K, Gibco) + Penicillin-Streptomycin (15140-122, Gibco). Jump-In™ T-REx™ HEK293 (RRID:CVCL_YL74), Huh-7 (RRID:CVCL_0336), 1321N1 (RRID:CVCL_0110) and SK-MEL-28 (RRID:CVCL_0526) in DMEM (D5796 Sigma) + 10% Fetal Bovine Serum (10270-106, Lot 42F8381K, Gibco) + Penicillin-Streptomycin (15140-122,

Gibco). All cell lines were grown at 37 °C and 5% $CO_2$. Transgene expression is induced via doxycycline treatment (1 µg/ml (D9891-1G, Sigma-Aldrich; CHEBI:50845).

To generate monoclonal KO clones, cell lines were stably transduced with the lentiviral vector pLentiCRISPRv2 (Addgene #52961) expressing gRNAs targeting SLCs (Dataset EV2). After 5–10 days of puromycin selection, cell pools were sorted to single cells using FACS or a limited dilution series and monoclonal cell lines were expanded and genotyped using an NGS-based approach (Veeranagouda et al, 2018). At least two independent KO clones were generated for each target. A detailed description of the KO generation can be found at: https://zenodo.org/records/7457297.

To re-express functional and tagged SLCs in the respective KO background, codon-optimized SLC cDNAs were cloned from pDONR221 gateway entry clones (https://www.addgene.org/depositor-collections/re-solute/) into doxycycline-inducible lentiviral gateway destination vectors containing C- or N-terminal HA-Twin-Strep® tags (Addgene # 194066, 194065) and stably transduced into KO cell lines. For N-terminally tagged SLCs, cDNAs with STOP codons at the end of the SLC ORF were used. Transgene expression was induced via doxycycline treatment at 1 µg/ml (D9891-1G, Sigma-Aldrich; CHEBI:50845) for 24 h. For each SLC, two pooled KO-OE cell lines were generated, each derived from an independent clonal KO cell line. A more detailed description of the KO-OE cell line generation can be found at: https://zenodo.org/records/7457295.

To generate cell lines overexpressing codon-optimized tagged SLCs in a WT background (WT-OE), SLC cDNAs were cloned from the same pDONR221 entry clones into modified pJTI™ R4 CMV-TO MCS pA vectors (Thermo Fisher Scientific) that contained C- or N-terminal HA-Twin-Strep® tags. Jump-In™ T-REx™ HEK293 cells (Thermo Fisher Scientific) were stably transfected with these constructs as recommended by the manufacturer's protocol. Transgene expression was induced via doxycycline treatment at 1 µg/ml for 24 h or the duration indicated. A more detailed description of the pooled WT-OE cell line generation and quality control can be found at: https://zenodo.org/records/7457221 and https://zenodo.org/records/5566805.

## Cell line selection for this study

For several SLC, multiple cell lines and transcriptomics data sets were generated. Based on expression analysis by western blot, immunofluorescence and AP-MS proteomics, we chose one cell line for each SLC to be included in this study. For six SLCs, we were not able to generate and include a functional cell line. For metabolite extraction, the appropriate cell line model for each SLC was chosen based on its expression across the panel of cell lines described above. In case the SLC is endogenously expressed in one of the five cancer cell lines and a respective knock-out cell line was available, the KO-OE cell line model was chosen for targeted metabolomics analysis. In case the SLC is not endogenously expressed in any of the cancer cell lines, or it was not possible to derive a respective knock-out cell line, the WT-OE model was chosen.

## Metabolite extraction

Cells were plated in quadruplicates in 6-well or 24-well plates in culture medium in the absence or presence of 1 µg/ml doxycycline

(150,000 cells per well for HCT 116 and LS180, 100,000 cells per well for 1321N1, 50,000 cells per well for Huh-7, 70,000 cells per well for SK-MEL-28, 750,000/150,000 cells per well in poly-L-lysine coated 6-well / 24-well plates for Jump-In™ T-REx™ HEK293). After 16-24 h, cells were first gently washed with room temperature ammonium carbonate buffer (75 mM NH$_4$HCO$_3$, pH 7.4). Then, cells were transferred to ice, where 300 μl/well in 24-well plate or 1500 μl/well in a 6-well plate of ice-cold 80:20 MeOH:H$_2$O was added. The cells were then scraped and transferred to a pre-cooled Eppendorf tube and immediately snap frozen in liquid nitrogen. Samples were thawed on ice before being centrifuged at 16,000×g for 10 min at 4 °C. The clarified metabolite-containing supernatants were moved into a high-performance liquid chromatography vial and stored at −80 °C until further preparation for LC–MS/MS analysis. Cell extracts were dried using a nitrogen evaporator. The dry residue was reconstituted in 16 μl H$_2$O, and 4 μl of sample extract was used for LC–MS/MS analysis. A mixture of isotopically labeled internal standards (Synthetic heavy isotope-labeled metabolites from Sigma-Aldrich and Cambridge Isotope Laboratories as well as Metabolite Yeast Extract (U-13C, 98%) from ISOtopic Solutions, Vienna, Austria) was added to each sample either with the extraction solvent or with the H$_2$O at reconstitution.

### Targeted metabolomics

A 1290 Infinity II UHPLC system (Agilent Technologies) was used for the chromatographic separation utilizing a ZORBAX RRHD Extend-C18, 2.1 × 150 mm, 1.8 μm analytical column (Agilent Technologies) and a SecurityGuard ULTRA Cartridge UHPLC C18 2.1 mm ID precolumn (Phenomenex). The columns were maintained at a temperature of 40 °C. The mobile phase A was 3% MeOH (v/v), 10 mM tributylamine, 15 mM acetic acid in H$_2$O, and mobile phase B was 10 mM tributylamine, 15 mM acetic acid in MeOH. The gradient elution with a flow rate 0.25 mL/min was performed for a total time of 24 min followed by back flushing of the column using a 6 port 2-position divert valve for 8 min using acetonitrile. At the end of the run 8 min of column equilibration with 100% mobile phase A was performed. Chromatographic separation was coupled to a 6470 triple quadrupole mass spectrometer (Agilent Technologies). The triple quadrupole mass spectrometer was operated in electrospray ionization negative mode, spray voltage 2 kV, gas temperature 150 °C, gas flow 1.3 L/min, nebulizer 45 psi, sheath gas temperature 325 °C, and sheath gas flow 12 L/min. The metabolites of interest were detected using a dynamic Multiple Reaction Monitoring (dMRM) mode. MRM transitions and expected retention times for target metabolites are provided in Dataset EV10.

### RNA isolation

For RNA isolation, $2.5 \times 10^5$ cells per well were seeded in four wells of a 12-well plate. In two of these wells, complete growth media is added, in the other two wells complete media containing 1 μg/ml doxycycline (Sigma-Aldrich; CHEBI:50845). Cell lines were harvested 24 h after doxycycline induction. At the time of harvest, media is removed from cells and cells are washed once with 1 ml PBS buffer (D8537, Sigma). After washing, 200 μl buffer RLT (Qiagen) is added directly to the cells. After complete cell lysis, lysates are transferred into matrix tubes and frozen at −80 °C until

further treatment. For RNA purification, RNeasy 96 Kits (Qiagen) were used. DNA is removed by on-column DNase treatment as described in the manual. After purification, RNA is eluted in 60–80 μl RNase-free water and stored at −80 °C until further usage. RNA concentrations are measured using the QuantiFluor® RNA System (Promega). For a selection of samples, RNA integrity was determined on a Bioanalyzer (Agilent). Total RNA samples were quantitatively and qualitatively assessed using the fluorescence-based Broad Range Quant-iT RNA Assay Kit (Thermo Fisher Scientific) and the Standard Sensitivity RNA Analysis DNF-471 Kit on a 96-channel Fragment Analyzer (Agilent), respectively.

### Library preparation and RNA sequencing

RNA samples were normalized on the MicroLab STAR automated liquid platform (Hamilton). Total RNA input of 250 ng was used for library construction with the NEBNext Ultra II Directional RNA Library Prep Kit for Illumina #E7760, together with the NEBNext Poly(A) mRNA Magnetic Isolation Module #E7490 upstream and the NEBNext Multiplex Oligos for Illumina #E7600 downstream (all New England Biolabs). The only deviation from the manufacturer's protocol was the use of Ampure XP beads (Beckman Coulter) for double-stranded cDNA purification, instead of the recommended SPRIselect Beads. The index PCR was performed with 12 cycles, while the final library was eluted in 35 μL. Library preparation was performed either manually or automatically using a Biomek i7 Hybrid workstation (Beckman Coulter) as described previously (Santacruz et al, 2022). mRNA libraries were then quantified by the High Sensitivity dsDNA Quanti-iT Assay Kit (Thermo Fisher) on a Synergy HTX (BioTek). mRNA libraries were also assessed for size distribution and adapter dimer presence (<0.5%) by the High Sensitivity Small Fragment DNF-477 Kit on a 96-channel Fragment Analyzer (Agilent).

All sequencing libraries were normalized on the MicroLab STAR (Hamilton), pooled and spiked in with PhiX Control v3 (Illumina). The library pools were subsequently clustered on S4 Flow Cell and sequenced on a NovaSeq 6000 Sequencing System (Illumina) with dual index, paired-end reads at 2 × 50 bp length (Read parameters: Rd1: 51, Rd2: 8, Rd3: 8, Rd4: 51), aiming for an average depth of 25 million Pass-Filter reads per sample.

### Metabolomics data processing and differential analysis

PeakBotMRM (https://github.com/christophuv/PeakBotMRM) was utilized for processing the raw LC–MS/MS data, followed by manual correction of some incorrectly recognized peaks and peak boundaries using the graphical user interface.

Metabolite peak areas were normalized to those of reference internal standards. Forty-seven metabolites had internal standards with the same chemical structures, and therefore these respective internal standards were used as references. For the remaining metabolites, we performed weighted linear regression on the expected metabolite concentrations against the calculated ratios for each metabolite and each internal standard. When internal standards achieved an adjusted $R^2$ greater or equal to 0.85 in at least 75% of the runs in which the metabolite was measured, the internal standard with the highest percentage was selected as the reference. If multiple internal standards achieved the same highest percentage, all were selected. For ten metabolites, an adjusted $R^2$

greater or equal to 0.85 was achieved in less than 75% and again the internal standard with the highest percentage was chosen. The internal standard(s) assigned to each target metabolite are provided in Dataset EV10. Individual injections were removed, or a reduced set of internal standards was used for normalization in case certain internal standards were not detected in certain runs. Normalization was performed by calculating the ratio of a metabolite peak area to the corresponding internal standard peak area, and subsequent log-transformation. For metabolites with multiple reference internal standards, the results were averaged. To also enable comparisons between different runs, we corrected for run-to-run differences (batch effects) by centering and scaling the internal standard-normalized values of each metabolite, per run, to its overall (i.e., across runs) median and interquartile range, respectively.

Finally, differential abundance upon SLC overexpression was tested for each metabolite by contrasting its normalized and batch-corrected values in doxycycline-induced samples to the corresponding uninduced samples (four replicates each) in a one-way ANOVA for WT-OE cell lines, or a two-way ANOVA for KO-OE cell lines, modeling the clone effect as additional factor. Resulting $P$ values were corrected for multiple testing per SLC, using the Benjamini–Hochberg procedure (Benjamini and Hochberg, 1995). Differentially abundant metabolites were called at a false discovery rate of 5%.

## Transcriptomics data processing

RNA-Seq data was processed using a Snakemake (version 6.6.1) pipeline running on Python (version 3.7). The pipeline comprised i) the read trimming using cutadapt (version 2.8), ii) the mapping of paired-end reads to the reference genome (GRCh38.p13 assembly with Ensembl 98 annotation) and to the overexpressed codon-optimized cDNA sequences for 447 SLCs using STAR (version 2.7.9a), iii) the quantification of reads per gene model using RSEM (version 1.3.2) as well as iv) the quality control of the sequencing data using FastQC (version 0.11.9) and the quality control before and after alignment using RSeQC (version 4.0.0) ensuring reliable downstream analyses. Quality control results after the alignment and quantification were compiled to individual reports using MultiQC (version 1.11).

## Differential expression analysis

To identify changes in gene expression levels upon overexpression of an SLC, differential expression analysis was performed by contrasting gene read counts between doxycycline-induced versus uninduced samples of the same cell line (two replicates each). Genes with a total of 10 or more reads across all samples of any cell line were considered for differential analysis. The codon-optimized cDNA sequences for 447 SLCs were excluded to consider only endogenously expressed SLC sequences. The DESeq2 package (version 1.42.1; (Love et al, 2014)) in R (version 4.3.3) was used to build a gene-level read count model, where counts were normalized by sample-specific size factors to account for differences in library size. Estimated gene-wide dispersions were used for the count modeling and a generalized linear model (GLM) was fit for each gene followed by hypothesis testing for differential expression using the Wald test. Resulting $P$ values were corrected for multiple testing using the Benjamini–Hochberg procedure (Benjamini and

Hochberg, 1995), and shrinkage of log fold changes was performed using the apeglm method (Zhu et al, 2019). Differentially expressed genes were called at a false discovery rate of 5%. Knock-out cell lines were analyzed for differential expression upon overexpression using the same procedure, with an added factor for the specific clone and a corresponding interaction factor with induction in the GLM. For differential expression between knock-out and wild-type, the GLM consisted of a factor for the genotype and its nested interaction with a factor for the clone.

## Uniqueness of differential gene expression

To identify genes with strong differential expression in specific cell lines, we calculated z-scores of the shrunken log fold change of each gene within a certain analysis and also across all analyses. Genes were then further filtered for significance of differential expression (adjusted $P$ value < 0.05) and for minimal signal (the gene had to have in one condition at least 50 read counts in both replica).

## Reactome pathway analyses

Pathways, reactions, compounds and proteins were extracted from the human Reactome pathway database (version 87), downloaded in BioPAX format. The data set was limited to "Metabolism" pathway (R-HSA-1430728) and its subcomponents.

Metabolites targeted in the metabolomics method were matched to compounds annotated in Reactome if they represented a more generic or a more specific term in the ChEBI ontology (release 229), considering the following relations: "is_a", "is_conjugate_base_of", "is_conjugate_acid_of", "is_tautomer_of", "is_enantiomer_of" and "has_role". Genes measured in transcriptomics analyses were matched to proteins annotated in Reactome via HGNC database (downloaded 2024-01-17).

To assess the regulation of individual metabolic pathways, we counted for each pathway the number of transcriptomics and metabolomics analyses featuring significant differentially expressed genes or differentially abundant metabolites that were matched to any reaction within the pathway. As pathways higher up in hierarchy or involving metabolites or genes present in many pathways will be matched very often, we assessed the significance of the frequency by permutation testing, shuffling the identities of the quantified metabolites and genes 200,000 times. The resulting significance levels were visualized on a Voronoi treemap of the hierarchical structure of all sub-pathways of human Metabolism pathway in Reactome, to show areas specifically affected by SLC overexpression.

To check whether we find known SLC substrates regulated in the targeted metabolomics experiments, we matched the targeted metabolites to the annotated substrates of SLCs (Goldmann et al, 2025), using the ChEBI ontology in the same way as described above.

Extending this approach to also consider metabolic conversion products of the annotated substrates, we searched for Reactome reactions that featured an annotated substrate (or a more specific term) matched to a reaction educt, and a targeted metabolite (or a more specific term) matched to a reaction product. Matching was done via ChEBI ontology using the same relations as listed above. Educts that were also products of the same reaction, as well as the substrate annotations of "hydron", "hydroxide" and "water" were not considered.

## Hierarchical clustering

Clustering was performed on 381 metabolic and 450 transcriptional profiles of metabolites and genes which were measured in all analyses (142 metabolites, 14,187 genes). The profiles of shrunken log fold changes were standard-normalized for each SLC, and Euclidean distance between the profiles was used for a hierarchical clustering (hclust function in R, version 4.3.3, with Ward's method).

As a measure of functionality, we tested clusters for enrichment of SLC functional properties of different classes (coupled ion, family, fold, location, substrate class). Respective classes and annotations are described in (Goldmann et al, 2025). Fisher's exact test for overrepresentation was performed for each functional property in each cluster, only considering properties with at least 3 annotations. Resulting $P$ values were corrected for multiple testing using the Benjamini–Hochberg procedure (Benjamini and Hochberg, 1995) for each class separately, and enrichments of functional SLC properties in clusters were called at a false discovery rate of 20%.

Guided by the mean silhouette width (cluster library, version 2.1.6; (Rousseeuw, 1987)) and the enrichment analysis, the metabolomics and transcriptomics dendrograms were divided into 48 and 60 clusters, respectively.

## SLC-level transcriptomic assessment of metabolic effects

To assess changes to the expression of metabolic genes driven by each SLC and transcriptomic cluster, we mapped genes to Reactome pathways as described above. For each cell line, we then calculated the mean expression $\log_2$ fold change of all associated genes belonging to each pathway. We defined top-level pathways as the set of direct child pathways of the human Metabolism pathway (R-HSA-1430728), and selected the nine pathways with the largest number of associated genes for plotting on Fig. 5A.

## Clusters coherence analysis

Variation of Information (VI) between clusterings was calculated by subtracting their mutual information from their joint entropy as defined in (Meilă, 2007):

$$VI = H_{1,2} - MI_{1,2}$$

Replacing entropy and mutual information with their respective definitions based on marginal probabilities (e.g., $p_i$) and joint probabilities (e.g., $p_{i,j}$) results in the decomposition of the contribution of each pair of clusters $i$ and $j$ in clusterings 1 and 2, respectively:

$$VI = \sum_{i,j} p_{i,j} \log \frac{p_i p_j}{p_{i,j}^2}$$

The total contribution of a cluster $i$ in clustering 1 was calculated by summing up its pairwise contribution with each cluster $j$ in clustering 2. Normalization of VI was performed by dividing VI by the clusterings' shared entropy $H_{1,2}$. Computations were performed in R (version 4.3.3).

## GO term enrichment analysis

To determine enriched GO terms within the overall gene expression profile of transcriptomic clusters 6 and 7, Gene Set Enrichment Analysis (GSEA) was performed in R using the *clusterProfiler* package (Yu et al, 2012). Standard-normalized $\log_2$ fold changes were averaged across SLCs of the two clusters, arranged in descending order and pre-ranked GSEA performed with the *gseGO* function using the following parameters: ont = "ALL", keyType = "ENSEMBL", minGSSize = 10, maxGSSize = 500, OrgDb = "org.Hs.eg.db", by= "fgsea".

## Targeted LC–MS/MS for metabolite quantification

For stable isotope labeling experiments, HCT116-SLC45A4-KO-OE cells were grown in RPMI media supplemented with either 4 mM $^{13}C^{15}N$-arginine or $^{13}C^{15}N$-glutamic acid for 3 days, then induced with 1 µg/mL Dox for 24 h. For other experiments with HCT116-SLC45A4 KO-OE cells, cells were seeded in regular RPMI media overnight, then switched to RPMI containing 10% regular FBS, dialyzed FBS, horse serum or no serum, with supplementation of 1 mM DFMO, 2 mM aminoguanidine, or 0.1 unit/mL porcine diamine oxidase (Sigma), and +/− 1 µg/mL Dox for 9 h. For experiments with HEK293 WT-OE cell lines from transcriptomic cluster 17, cells were seeded in regular DMEM overnight +/− 1 µg/mL Dox. Metabolite extraction was performed as described above, and additionally the protein pellets remaining after extraction were dissolved in 0.5 M KOH and subsequently quantified by BCA assay.

Targeted metabolite measurements were performed on a Waters Xevo TQ system utilizing an ACQUITY UPLC HSS T3 1.8-µm reversed-phase column (for SLC45A4-related metabolites except sucrose), ACQUITY UPLC BEH Amide 1.7 µm column (for SLC45A4 sucrose uptake quantification), or ACQUITY UPLC BEH HILIC 1.7 µm (for cluster 17-related metabolites). All columns were maintained at 40 °C. The mobile phase A was water with 0.1% formic acid, and mobile phase B was acetonitrile with 0.1% formic acid. For reversed-phase chromatography, the elution gradient was as follows: $t = 0$, 2% solvent B, flow rate 0.15 mL/min; $t = 2.5$, 2% solvent B, flow rate 0.15 mL/min; $t = 3.5$, 100% solvent B, flow rate 0.3 mL/min; $t = 6$, 100% solvent B, flow rate 0.3 mL/min; $t = 6.1$, 2% solvent B, flow rate 0.2 mL/min; $t = 8$, 2% solvent B, flow rate 0.15 mL/min. For hydrophilic interaction chromatography, the elution gradient was: $t = 0$, 100% solvent B, flow rate 0.25 mL/min; $t = 5$, 70% solvent B, flow rate 0.25 mL/min; $t = 8$, 25% solvent B, flow rate 0.3 mL/min; $t = 9.7$, 25% solvent B, flow rate 0.3 mL/min; $t = 10$, 100% solvent B, flow rate 0.25 mL/min; $t = 12$, 100% solvent B, flow rate 0.25 mL/min. The triple quadrupole mass spectrometer was operated in electrospray ionization mode with polarity determined according to the analyte. Other settings were as follows: capillary voltage 3 kV, source temperature 150 °C, desolvation temperature 350 °C, desolvation gas flow 650 L/h, collision gas flow 0.15 mL/min. MRM transitions were determined for each target metabolite using the IntelliStart function of MassLynx software and were as follows: GABA (+) 104.0953 → 69.0814; putrescine (+) 89.0681 → 72.06; spermidine (+) 146.2537 → 72.1248; $^{13}C^{15}N$-Glutamic acid (+) 154.1596 → 89.1233; sucrose (−) 341.1481 → 179.0127; carnitine (+) 162.201 → 103.0763; creatine (+) 132.1014 → 44.1745; glycerophosphorylcholine (+) 258.2624 → 104.1273; glycine (+) 76.064 → 30.1437; histidine (+) 156.1652 → 110.177; methionine (+) 150.083 → 104.0747; myo-inositol (+) 181.1592 → 109.019; taurine (+) 126.0466 → 44.1084.

Waters RAW data folders were converted to mzML files using Proteowizard MSConvert (Chambers et al, 2012) and peak areas were quantified using EL-MAVEN (Agrawal et al, 2019). With

exception of the stable isotope labeling experiment, peak areas were then normalized to both internal standard ($^{13}C^{15}N$-glutamic acid) and BCA protein quantification values and the results expressed as normalized peak areas.

## SLC45A4 expression, purification, proteoliposome preparation, and putrescine uptake assay

SLC45A4 was expressed and purified as previously described (Raturi et al, 2023). Briefly, full-length human SLC45A4 was expressed in Expi293F cells using the BacMam system. Cells were harvested after 48 h and resuspended in lysis buffer (300 mM NaCl, 50 mM HEPES pH 7.5) in the presence of cOmplete Protease Inhibitor Cocktail tablets (Roche) and lysed at 4 °C by two passes through an EmulsiFlex-C5 homogenizer (Avestin). The lysate was clarified by centrifugation at 10,000×g for 15 min, and membrane pelleted by ultracentrifugation at 100,000×g for 1 h at 4 °C. The resulting membrane pellet was solubilized in lysis buffer supplemented with 1% dodecylmaltoside (DDM) and 0.1% cholesterol hemisuccinate (CHS). The solubilized membranes were then incubated with pre-equilibrated Strep-Tactin XT Superflow resin (IBA-Lifesciences) for 1 h at 4 °C, washed with column buffer (300 mM NaCl, 50 mM HEPES pH 7.5, and 0.05% DDM/CHS (Anatrace)) supplemented with 10 mM MgCl$_2$ and 1 mM ATP, and protein was eluted with column buffer supplemented with 50 mM D-biotin. The affinity tag was cleaved with TEV protease overnight and the protein further purified by reverse IMAC purification using TALON resin (Takara). The tag-cleaved protein was concentrated using a 100 kDa cutoff centrifugal concentrator (Sartorius) and subjected to size exclusion chromatography using a Superdex 200 10/300 GL column (GE Healthcare) pre-equilibrated with gel filtration buffer (150 mM NaCl, 20 mM HEPES pH 7.5, 0.025% DDM/CHS). Peak fractions were pooled for subsequent experiments.

Liposomes were prepared by dissolving 120 mg soy lipids (Sigma-Aldrich) in 20 mL of chloroform and then drying as a thin film in a rotovap. The lipids were resuspended in 6 mL of encapsulation buffer (20 mM HEPES pH 7.5, 150 mM NaCl, 5 mM glycine and 5 mM valine), then subjected to 10 freeze-thaw cycles by transferring between liquid nitrogen and a 37 °C water bath. To form liposomes of uniform size, lipids were extruded 21 times through a 400 nm filter membrane using an LF-1 handheld syringe extruder (Avestin). Triton X-100 was added to a final concentration of 0.1%, then purified SLC45A4 protein added at a ratio of 0.1 mg protein per 4 mg lipid. For the control liposomes, an equivalent volume of encapsulation buffer was added to the liposomes. The mixture was incubated on ice for 1 h. Excess salt and detergent were then removed by passing through PD-10 desalting columns pre-equilibrated with encapsulation buffer. The eluted liposome suspension was then ultra-centrifuged at 100,000×g, 4 °C for 40 min, and the supernatant discarded. The (proteo)liposome pellets were then resuspended to a lipid concentration of 250 mg/mL in assay buffer (20 mM HEPES pH 7.5, 150 mM NaCl, 0.5 mM PIPES, and 0.5 mM Tris base). The uptake assay was initiated by the addition of 100 µM putrescine, followed by incubation at room temperature. At select time points, 20 µL aliquots of the assay suspension were filtered through Sephadex G-50 (fine) spin columns (Cytiva) pre-equilibrated with base buffer (20 mM HEPES pH 7.5, 150 mM NaCl) by centrifugation. Eluates were snap frozen in liquid nitrogen until LC–MS

analysis. For metabolite extraction, liposomes were dissolved in 400 µL 1:1 methanol/chloroform mixture, then 120 µL of water was added to achieve a final methanol/chloroform/water ratio of 5:5:3. The mixture was vortexed vigorously for 10 s, then centrifuged at 18,000×g, 4 °C for 10 min. The aqueous supernatant was transferred to HPLC vials, dried under a stream of nitrogen gas, and then stored at −80 °C until LC–MS analysis as described above.

## Flow cytometry

For all flow cytometry experiments, cells were seeded two days prior at an appropriate density such that cells are 80% confluent on the day of collection. Cells were induced +/− 1 µg/mL Dox 24 h before harvest. On the day of collection, cells are detached using 1 mM EDTA in PBS, washed with PBS, then fixed with 4% formaldehyde in PBS for 10 min. Fixed cells were washed once with PBS, resuspended in PBS + 1% BSA and stored at 4 °C for maximum 1 week. For intracellular GNL staining, $\sim3 \times 10^5$ cells/well were added to triplicate wells on a V-bottom 96-well plate, centrifuged, and supernatant decanted. Cells were permeabilized by resuspension in 100 µL 90% MeOH, 10% PBS and incubation on ice for 30 min. Cells were then aliquoted to V-bottom 96-well plate in triplicate wells, centrifuged at 350×g for 3 min, supernatant decanted, and blocked by resuspension in PBS + 10% FBS and incubation on a rotary shaker at room temperature for 1 h, followed by primary staining with biotinylated GNL 1:1000 (Vector Labs) in PBS + 10% FBS for 1 h. Cells were washed twice by resuspending in PBS + 10% FBS and incubating with shaking for 5 min each time. Secondary staining was performed with Alexa Fluor 647-conjugated streptavidin for 30 min at room temperature, following which cells were washed twice as above and resuspended in PBS for flow cytometry analysis. For VVL staining, cells were added to V-bottom 96-well plate, centrifuged and supernatant decanted. Cells were then incubated on ice for 30 min with biotinylated VVL (Vector Labs) diluted 1:1000 in TSM buffer, washed by pelleting and resuspension in PBS, secondary stained with Alexa Fluor 488-conjugated streptavidin on ice for 30 min, then wash again and resuspended in PBS. Staining for Tn antigen followed the same procedure, but with 1 h incubation on ice using anti-Tn antibody (5F4 clone) at 1:3 in PBS + 1% BSA and secondary staining with Alexa Fluor 594-conjugated anti-mouse IgM. Flow cytometry analysis was performed on LSR Fortessa (BD Biosciences) flow cytometer with a high-throughput microplate sampler.

Flow cytometry data files were exported in FCS 2.0 format and processed in FlowJo (BD Biosciences). For each experiment batch, a first gate was manually set on the forward scatter (FSC)-Area vs side scatter (SSC)-Area surface to isolate non-debris cells. A second gate was then set on the FSC-Area vs FSC-Height surface to isolate single cells. Fluorescence values for individual gated cells were exported and analyzed in R. Since standard statistical testing between flow cytometry populations (>10,000 data points per group) would lead to very small *P* values for every comparison, we instead used effect sizes, calculated as Cohen's *d* using the *effsize* package in R, for comparison of Dox-induced/uninduced samples of each cell line.

## Immunofluorescence

HCT116-SLC45A4-KO-OE or HEK293-MFSD5-WT-OE cells were seeded at 150,000 cells/well on polylysine-coated glass cover slips

placed in 24-well plates and induced with 1 μg/mL Dox overnight. Media was aspirated and cells were fixed in 4% formaldehyde in PBS for 10 min at room temperature. Cells were then blocked and permeabilized with IF blocking buffer (PBS + 10% FBS + 0.3% Triton X-100) for 1 h, then primary antibodies or lectins added at the following dilutions in IF blocking buffer: rat anti-HA 1:1000 (Roche), biotinylated GNL 1:1000 (Vector Labs), mouse M6P-1 1:50 (ABCD Antibodies), rabbit anti-LAMP1 1:200 (Cell Signaling). Following two hours of gentle rocking at room temperature, cells were washed three times with IF blocking buffer, then secondary staining performed for 1 h with the following antibodies: anti-mouse Alexa Fluor 568 1:1000, streptavidin-Alexa Fluor 647 1:1000, anti-rabbit Alexa Fluor 488 1:1000, DAPI 1:2000. Cells were washed three times, mounted on glass slides with Fluoromount-G (Thermo Fisher), then imaged with an LSM 700 confocal microscope (Zeiss).

## Glycoproteomics

Cells were seeded two days prior to harvest at an appropriate density such that cells are 80% confluent on the day of collection. Cells were induced +/− 1 μg/mL Dox 24 h before collection. On the day of harvest, cells are detached using 1 mM EDTA in PBS, washed twice with PBS to remove residual EDTA, then pelleted and stored at −80 °C until lysis. Cell pellets were lysed using a buffer containing 50 mM Tris-HCl (Trizma Base: Sigma-Aldrich, Cat#T6066; HCl: Sigma-Aldrich, Cat#258148), 8 M urea (VWR, Cat#0568), 150 mM NaCl (Merck, Cat#106404), pH 8, containing phosphatase inhibitors (Cell Signaling Technology, Cat#5870). Samples were kept on ice for 30 min before being sonicated using a water bath sonicator for 1 min at low intensity. Samples were centrifuged at 20,000× $g$ for 15 min at 4 °C. Resulting supernatants were transferred to a new tube, and protein contents were quantified using Pierce™ BCA Protein Assay Kit (Thermo Fischer, Cat#23225). For each sample, 300 μg of protein was reduced using DTT (Roche, Cat#10708984001; final concentration 10 mM) for 30 min at 37 °C, and subsequently alkylated using IAA (Sigma-Aldrich, Cat#I6125; final concentration 20 mM) for 30 min in the dark at room temperature. Samples were then diluted using a 50 mM Tris-HCl buffer to 4 M urea before adding 6 μg of Lys C (FUJIFILM Wako, Cat#125 05061) to each sample (1 μg Lys C: 50 μg sample protein). Samples were incubated for 2 h at 30 °C, before being further diluted by adding 50 mM Tris-HCl buffer to lower the urea concentration to 2 M. 6 μg of Trypsin Gold (Promega, Cat#V5280) were added to each sample (1 μg Trypsin: 50 μg sample protein), and samples were further incubated for 15 h at 37 °C. At this point, samples were acidified using TFA to a final concentration of 0.5% before being cleaned using 200 mg (3cc) Sep-Pak cartridges (Waters, Cat#WAT054945). The resulting elution fraction containing the cleaned peptides was dried under vacuum and stored at −20 °C. For TMT labeling, 200 μg of peptides of each sample were prepared in a final volume of 100 μL of 100 mM HEPES buffer pH 7.6. Samples were labeled using TMTpro 18 plex (Thermo Scientific, Cat#A52045) reagents, according to the manufacturer's instructions. The labeling efficiency was determined by LC–MS/MS on a small aliquot of each sample. After quenching the labeling reaction, samples were mixed in equimolar amounts, evaluated by LC–MS/MS. The mixed sample was acidified to a pH below 2 with 10% TFA and desalted using C18 cartridges (Sep-Pak Vac 1cc (200 mg), Waters, Cat#WAT054945). Peptides were eluted

with 2 × 600 μl 80% Acetonitrile (ACN) and 0.1% Formic Acid (FA), followed by freeze-drying. Glycopeptides from the resulting TMTpro mixes were enriched by performing off-line ion-pairing (IP) HILIC chromatography, as described previously (12). Briefly, the dried samples were taken up in 100 μL 75% acetonitrile containing 0.1% TFA and subjected to chromatographic separation on a TSKgel Amide-80 column (4.6 × 250 mm, particle size 5μ) using a linear gradient from 0.1% TFA in 80% acetonitrile to 0.1% TFA in 40% acetonitrile over 35 min (Dionex Ultimate 3000, Thermo). The 30 collected fractions were vacuum dried. Samples were resuspended using 0.1% trifluoroacetic acid (TFA, Thermo Scientific, Cat#28903). The IP-HILIC fractions were individually analyzed by LC–MS/MS. The nano HPLC system used was an UltiMate 3000 HPLC RSLC nano system (Thermo Scientific) coupled to a Q Exactive HF-X mass spectrometer (Thermo Scientific), equipped with a Proxeon nanospray source (Thermo Scientific). Peptides were loaded onto a trap column (Thermo Scientific, PepMap C18, 5 mm × 300 μm ID, 5-μm particles, 100 Å pore size) at a flow rate of 25 μL/min using 0.1% TFA as mobile phase. After 10 min, the trap column was switched in line with an analytical column (Thermo Scientific, PepMap C18, 500 mm × 75 μm ID, 2 μm, 100 Å). Peptides were eluted using a flow rate of 230 nl/min and a binary 180 min gradient. The two steps gradient started with the mobile phases: 98% A solution (water/formic acid, 99.9/0.1, v/v) and 2% B solution (water/acetonitrile/formic acid, 19.92/80/0.08, v/v/v), which was then increased to 35% B over the next 180 min, followed by a gradient of 90% B for 5 min, which was finally, in a 2 min period, decreased to the gradient 95% A and 2% B for equilibration at 30 °C. The following parameters were used for MS acquisition using an Exploris 480 instrument, operated in data-dependent mode: the instrument was operated in a positive mode; compensation voltages (CVs) used = CV-40, CV-50, CV-60; cycle time (per CV) = 1 s. The monoisotopic precursor selection (MIPS) mode was set to Peptide. Precursor isotopes and single charge state precursors were excluded. MS1 resolution = 60,000; MS1 Normalized AGC target = 300%; MS1 maximum inject time = 50 msec. MS1 restrictions were relaxed when few precursors were present. MS1 scan range = 400–1,600 m/z; MS2 resolution = 45 000; MS2 Normalized AGC target = 200%. Maximum inject time = 250 msec. Precursor ions charge states allowed = 2–7; isolation window = 1.4 $m/z$; fixed first mass = 110 m/z; HCD collision energies = 30,33,36; exclude isotopes = True; dynamic exclusion = 45 s. All MS/MS data were processed and analyzed using Xcalibur v3.1 (Thermo), and Byonic (Protein Metrics) included in Proteome Discoverer v3.0. Two default glycan databases (Mammalian O-glycans and Human N-glycans) were used with the latest human or mouse UniProt database to generate the glycopeptide search space. We used an in-house developed R script to filter out poorly matching spectra, considering a Byonic score higher than 200. Significant (1% and 5% FDR) changes in glycopeptide abundances were determined by considering $P$ value and magnitude of fold change simultaneously as previously described (Hein et al, 2015). Further data processing, analysis and visualization were performed in R.

## Data availability

Transcriptomics data can be accessed at the RESOLUTE web portal (https://re-solute.eu/resources/datasets) and at the European

Nucleotide Archive under accession number PRJEB81360. Targeted metabolomics data can be accessed at the RESOLUTE web portal (https://re-solute.eu/resources/datasets) and at MetaboLights under accession numbers MTBLS10077 and MTBLS11393. Immunofluorescence imaging data can be accessed at the BioImage Archive under accession numbers S-BIAD1422 and S-BIAD1423. Flow cytometry data can be accessed at Zenodo under accession numbers 14720859, 14720912, 14720951, and 14720966.

The source data of this paper are collected in the following database record: biostudies:S-SCDT-10_1038-S44320-025-00106-4.

## Peer review information

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

## Acknowledgements

This study received funding from the RESOLUTE consortium. RESOLUTE has received funding from the Innovative Medicines Initiative 2 Joint Undertaking under grant agreement No 777372. This Joint Undertaking receives support from the European Union's Horizon 2020 research and innovation programme and EFPIA. This article reflects only the authors' views and neither IMI nor the European Union and EFPIA are responsible for any use that may be made of the information contained therein. The last year of work, including validation of data and writing of the manuscript, was supported mainly by the Austrian Academy of Sciences. DBS and GC are supported by the Ontario Institute for Cancer Research, Royal Institution for the Advancement of Learning McGill University, Kungliga Tekniska Hoegskolan, Diamond Light Source Limited and by the Innovative Medicines Initiative 2 Joint Undertaking (JU) under grant agreement No 875510. The JU receives support from the European Union's Horizon 2020 research and innovation program and EFPIA. The lab of JMP received funding from the Medical University of Vienna, the T von Zastrow foundation. SM was supported by the European Union's Horizon 2020 research and innovation programme under the Marie Sklodowska-Curie grant agreement No. 841319 and the ESPRIT-Programme of the Austrian Science

Fund (FWF, Project number: ESP 166). GS-F was supported by the Austrian Academy of Sciences. We thank U Mandel and H Clausen (University of Copenhagen) for providing 5F4 hybridoma supernatant. We highly appreciate the feedback and critical reading of the manuscript by Ann-Katrin Hopp, Enrico Girardi, Gabriel Onea, Kai-Chun Li, Leonhard Heinz, Manuele Rebsamen, and Vojtech Dvorak.

## Author contributions

**Tabea Wiedmer**: Conceptualization; Data curation; Formal analysis; Supervision; Validation; Investigation; Visualization; Methodology; Writing—original draft; Project administration; Writing—review and editing. **Shao Thing Teoh**: Conceptualization; Data curation; Formal analysis; Supervision; Validation; Investigation; Visualization; Methodology; Writing—original draft; Writing—review and editing. **Eirini Christodoulaki**: Data curation; Formal analysis; Investigation; Visualization; Methodology. **Gernot Wolf**: Formal analysis; Supervision; Investigation; Methodology; Writing—review and editing. **Chengzhe Tian**: Data curation; Formal analysis; Investigation; Methodology. **Vitaly Sedlyarov**: Data curation; Formal analysis; Investigation; Methodology. **Abigail Jarret**: Data curation; Supervision; Investigation; Methodology. **Philipp Leippe**: Visualization; Writing—review and editing. **Fabian Frommelt**: Visualization; Writing—review and editing. **Alvaro Ingles-Prieto**: Visualization; Methodology. **Sabrina Lindinger**: Data curation; Investigation; Methodology. **Barbara M G Barbosa**: Data curation; Investigation; Methodology. **Svenja Onstein**: Data curation; Investigation; Methodology. **Christoph Klimek**: Data curation; Investigation; Methodology. **Julio Garcia**: Data curation; Investigation; Methodology. **Iciar Serrano**: Investigation; Methodology. **Daniela Reil**: Investigation; Methodology. **Diana Santacruz**: Investigation; Methodology. **Mary Piotrowski**: Investigation; Methodology. **Stephen Noell**: Investigation; Methodology. **Christoph Bueschl**: Methodology. **Huanyu Li**: Investigation; Methodology. **Gamma Chi**: Investigation; Methodology. **Stefan Mereiter**: Investigation; Methodology; Writing—review and editing. **Tiago Oliveira**: Investigation; Methodology. **Josef M Penninger**: Supervision; Funding acquisition. **David B Sauer**: Supervision; Funding acquisition; Writing—review and editing. **Claire M Steppan**: Supervision; Funding acquisition. **Coralie Viollet**: Supervision; Methodology. **Kristaps Klavins**: Supervision; Methodology. **J Thomas Hannich**: Supervision; Methodology. **Ulrich Goldmann**: Conceptualization; Data curation; Formal analysis; Supervision; Visualization; Methodology; Writing—review and editing. **Giulio Superti-Furga**: Conceptualization; Resources; Supervision; Funding acquisition; Methodology; Writing—original draft; Project administration; Writing—review and editing.

Source data underlying figure panels in this paper may have individual authorship assigned. Where available, figure panel/source data authorship is listed in the following database record: biostudies:S-SCDT-10_1038-S44320-025-00106-4.

## Disclosure and competing interests statement

GS-F is a co-founder and owns shares of Solgate GmbH, an SLC-focused company.

# Expanded View Figures

**Figure EV1. Comprehensive multi-omic coverage of the SLC superfamily.**

(**A**) Heatmap illustrating coverage of 447 SLCs across five cancer cell lines considered for KO and KO-OE (HCT 116, LS180, 1321N1, SK-MEL-28, Huh-7) as well as Jump-In™ T-REx™ HEK293. Tpm > 1 was used as threshold for expression. (**B**) Number of SLCs with targeted metabolomics and transcriptomics data sets and respective parental cell lines included in this study. (**C**) Coverage of transcriptomics data sets in this study compared to the SLC superfamily (the RESOLUTE list of 447 SLCs). The coverage is given both in absolute numbers per family (top) and percentage of members per family (bottom). (**D**) Coverage of targeted metabolomics data sets in this study compared to the SLC superfamily (the RESOLUTE list of 447 SLCs). The coverage is given both in absolute numbers per family (top) and percentage of members per family (bottom). (**E**) Principal component analysis of 378 targeted metabolomics differential analyses. Differential analysis was performed between $+$Dox/$-$Dox samples. Data points are colored according to the parental cell lines. Cell line models and number of analyses per cell line are given in brackets. (**F**) Comparison of the average $\log_2$ fold change and frequency of significant changes ($P_{adj.}$ <0.05) for each metabolite across all differential analyses performed (378 SLCs). Metabolite data points are shaded according to the frequency of detection above the calibration minimum (0–1). (**G**) Profiled SLCs by targeted metabolomics and divided according to the substrate class of their annotated substrates. Colored bars indicate proportion of SLCs with significant changes upon differential analysis of metabolite abundance $+/-$ doxycycline (Dox) induction, while gray bars indicate proportion of SLCs without significant metabolite changes. (**H**) Frequency of significant changes involving annotated substrates and metabolic conversions compared to frequency of all significant changes. All pairs of SLCs and targeted metabolites were grouped by a potential match of the SLC's annotated substrates to the targeted metabolite. The frequency of significant change was calculated per group and was found to be significantly higher for the 308 cases where a targeted metabolite could be directly matched to an annotated substrate for the overexpressed SLC. A smaller but also significant effect was found for the 2318 cases where a targeted metabolite could be matched via metabolic conversion to an annotated substrate for the overexpressed SLC (both Fisher's exact tests; error bars are the 95% confidence region of the perturbation frequency as calculated from the Fisher's test odds ratio estimate).

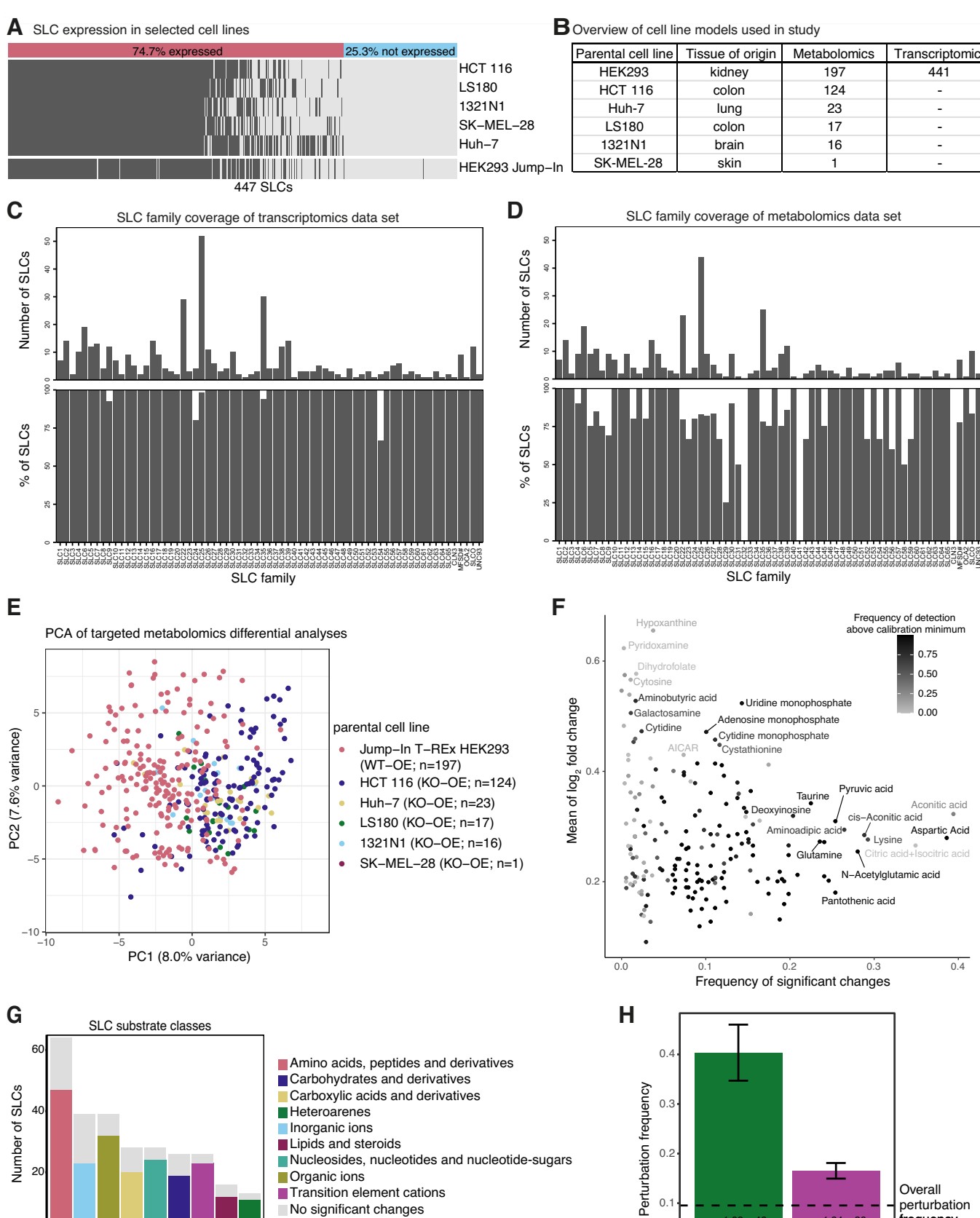

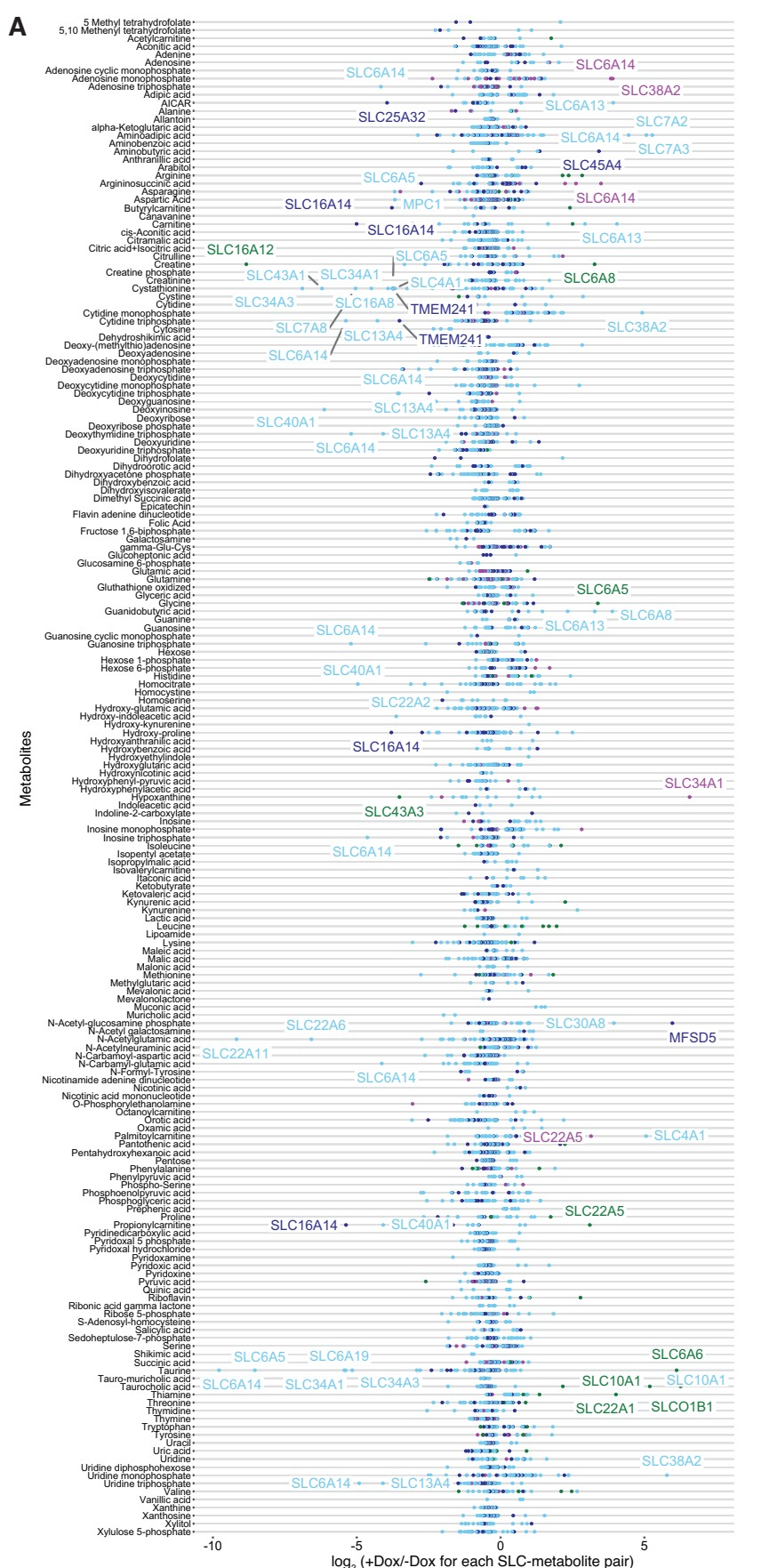

◀ **Figure EV2.  Overview of SLC-metabolite pairs by category and per metabolite.**

(A) Visualization of all significant SLC-metabolite pairs ($P_{adj.}$ <0.05) in a per-metabolite view with metabolites on the *y* axis and $\log_2$ fold change $+/-$doxycycline (Dox) for each SLC-metabolite pair on the *x* axis for the combined metabolomics data sets. Colors indicate the SLC-metabolite pair category ('annotated substrate', 'metabolic conversion', 'novel (non-orphan SLCs)', 'novel (orphan SLCs)'.

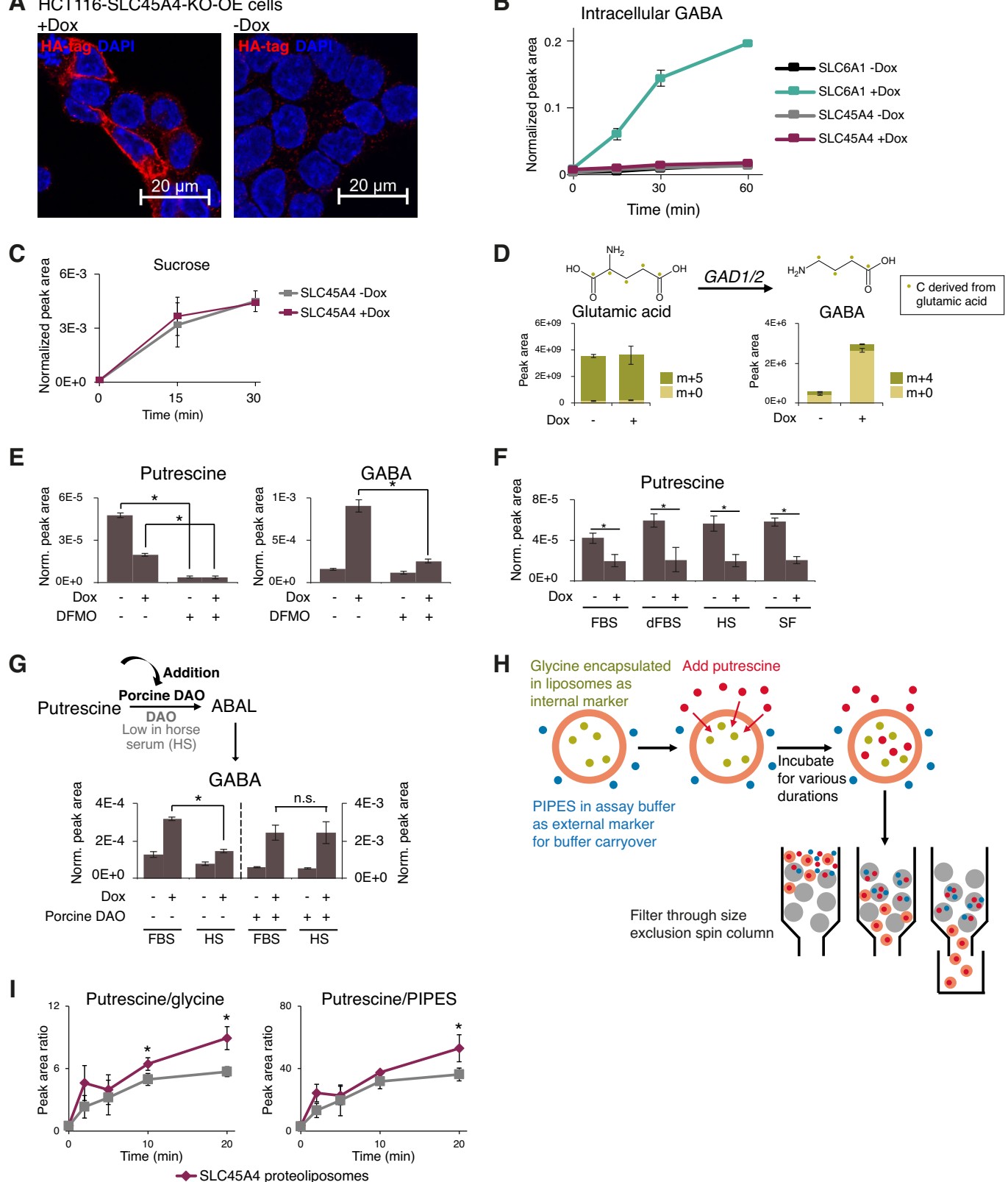

◀ **Figure EV3. SLC45A4 mediates GABA production via putrescine export and oxidation.**

(A) Representative immunofluorescence images of HCT116-SLC45A4-KO-OE cells +/− 24 h doxycycline (Dox) induction. Red and blue channels show HA-tagged SLC45A4 and DAPI nuclear counterstain, respectively. (B) Intracellular GABA levels in HCT 116 Renilla KO cells transduced with Dox-inducible SLC6A1 overexpression construct and HCT116-SLC45A4-KO-OE cells, following +/− 24 h Dox induction in OptiMEM media +dFBS and switch to OptiMEM +dFBS +100 μM GABA for the indicated durations. Data points represent means, error bars represent s.d. ($n = 3$). (C) Intracellular sucrose levels in HCT116-SLC45A4-KO-OE cells following +/− 24 h Dox induction and switch to media supplemented with 1 mM sucrose for the indicated durations. Data points represent means, error bars represent s.d. ($n = 3$). (D) Schematic representation of glutamic acid conversion to GABA, and labeled and unlabeled abundances of intracellular glutamic acid vs GABA in HCT116-SLC45A4-KO-OE cells +/− 24 h Dox induction. Carbon atoms potentially derived from glutamic acid (and hence labeled by $^{13}$C-glutamic acid) are marked by small green dots. Labeled species are as indicated for each metabolite. Bar heights represent means, error bars represent s.d. ($n = 6$). (E) Intracellular putrescine and GABA levels in HCT116-SLC45A4-KO-OE cells with and without ODC1 inhibition by 1 mM difluoromethylornithine (DFMO) +/− 9 h Dox induction. Bar heights represent means, error bars represent s.d. ($n = 3$). $P$ values (Welch's $t$ test): putrescine uninduced control vs DFMO, 1.50E-06; putrescine Dox control vs DFMO, 3.14E-05; GABA Dox control vs DFMO, 1.37E-04. (F) Intracellular putrescine levels in HCT116-SLC45A4-KO-OE cells cultured in regular FBS-containing media, media with dialyzed FBS (dFBS), media with horse serum (HS), or serum-free media (SF) +/− 9 h Dox induction. Bar heights represent means, error bars represent s.d. ($n = 3$). $P$ values (Welch's $t$ test): FBS uninduced vs Dox, 8.96E-03; dFBS uninduced vs Dox, 8.27E-03; HS uninduced vs Dox, 2.60E-03; SF uninduced vs Dox, 2.07E-04. (G) Intracellular GABA levels in HCT116-SLC45A4-KO-OE cells cultured in regular FBS-containing media vs horse serum (HS)-containing media, without (left) or with (right) porcine diamine oxidase (DAO) supplementation +/− 9 h Dox induction. Bar heights represent means, error bars represent s.d. ($n = 3$). $P$ values (Welch's $t$ test): FBS Dox vs HS Dox without DAO supplementation, 1.67E-05; FBS Dox vs HS Dox with DAO supplementation, 0.988. (H) Schematic of cell-free putrescine uptake assay. Glycine is encapsulated within liposomes inserted with SLC45A4 protein or protein-free control liposomes as an internal marker of liposome abundance. Liposomes are resuspended in assay buffer containing PIPES as an external marker of buffer carryover. Following addition of 100 μM putrescine and incubation for defined time points, liposome suspensions are filtered through Sephadex G-50 size exclusion chromatography (SEC) spin columns, which traps assay buffer components while allowing liposomes to flow through. The liposome-enriched eluates are then analyzed by LC–MS. (I) Putrescine/glycine and putrescine/PIPES peak ratios in SEC eluates of SLC45A4 proteoliposomes vs control liposomes over different uptake durations. Data points represent means, error bars represent s.d. ($n = 3$). $P$ values (Welch's $t$ test) of putrescine/glycine, SLC45A4 proteoliposomes vs control liposomes: $t = 0$, 0.907; $t = 2$, 0.119; $t = 5$, 0.583; $t = 10$, 0.0381; $t = 20$, 0.00960. $P$ values (Welch's $t$ test) of putrescine/PIPES, SLC45A4 proteoliposomes vs control liposomes: $t = 0$, 0.711; $t = 2$, 0.0556; $t = 5$, 0.669; $t = 10$, 0.110; $t = 20$, 0.0391. Where presented, asterisks (*) and n.s. on significance bars indicate $P < 0.05$ and $P \geq 0.05$, respectively. For panels (B, C, E, F and G), metabolite peak areas were normalized to internal standard compound peak area and further normalized to cellular protein content as described in "Methods".

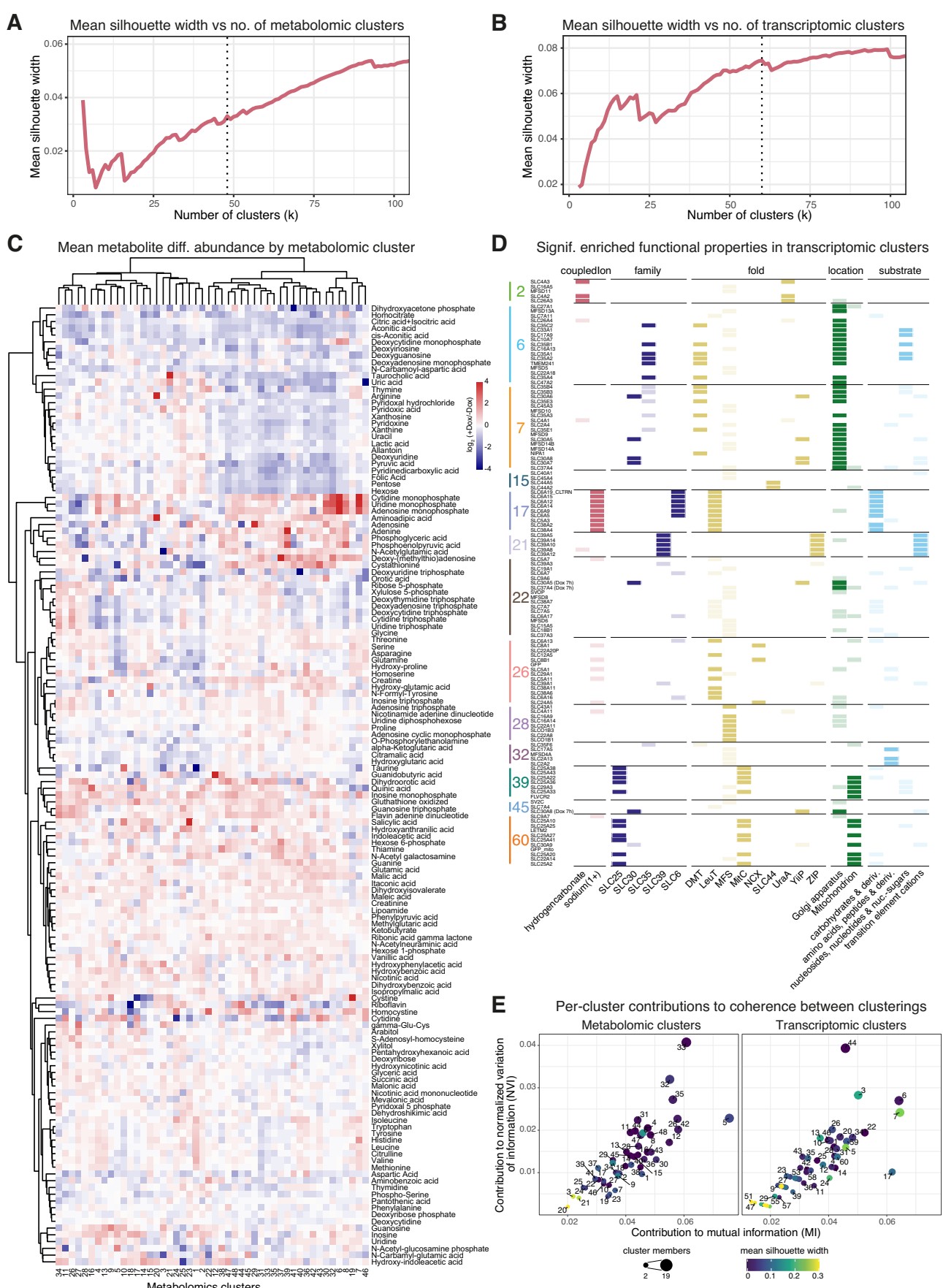

**A** Mean silhouette width vs no. of metabolomic clusters

**B** Mean silhouette width vs no. of transcriptomic clusters

**C** Mean metabolite diff. abundance by metabolomic cluster

**D** Signif. enriched functional properties in transcriptomic clusters

**E** Per-cluster contributions to coherence between clusterings

**Figure EV4.  Analyses of clustering results identify predominant metabolite changes in metabolomic clusters and reveals enriched functional properties in transcriptomic clusters.**

(A) Mean silhouette width across all metabolomic clusters at different cluster numbers (k = 2–100). The dashed line indicates the chosen k = 48. (B) Mean silhouette width across all transcriptomic clusters at different cluster numbers (k = 2–100). The dashed line indicates the chosen k = 60. (C) Heatmap of all clusters and metabolites based on the respective averaged $\log_2$ fold changes per cluster. Cluster numbers are indicated on $x$ axis. Metabolites are indicated on $y$ axis. (D) Transcriptomic clusters with significant SLC functional feature enrichment (Fisher's test $P < 0.2$). (E) Contributions of individual clusters to normalized variation of information and mutual information between metabolomics and transcriptomics clustering. Cluster size and discreteness (as quantified by silhouette width) are additionally indicated by size and color, respectively.

                                                                              

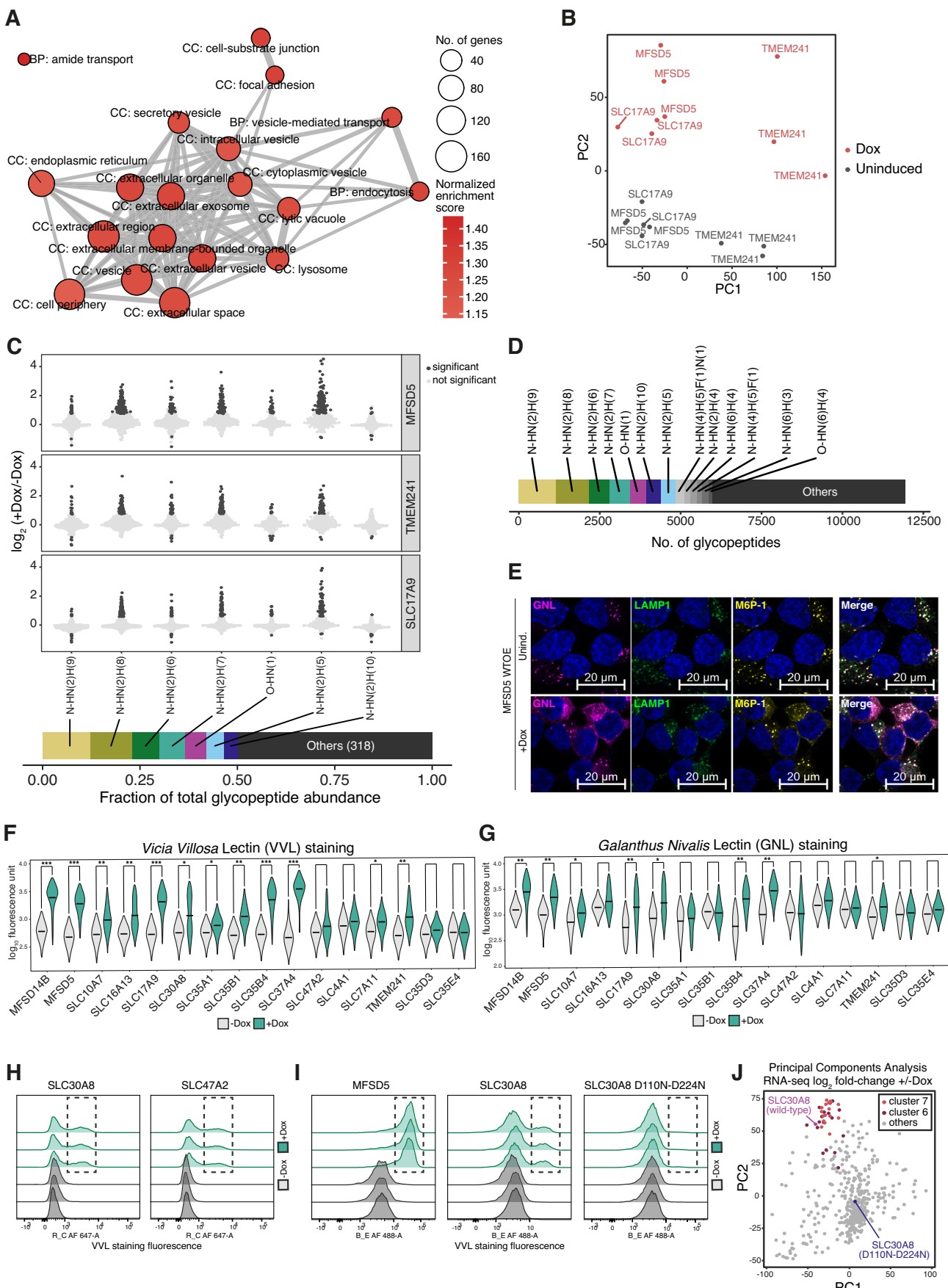

◀ **Figure EV5. SLC cluster alters both N-and O-linked glycosylation signatures.**

(A) GO terms significantly enriched ($P_{adj.}$ <0.05) by gene set enrichment analysis (GSEA) on most abundant glycoproteins detected in the data set. Abundance of each glycoprotein were calculated as sum of signal intensities of corresponding glycopeptides across 18 samples (3 replicates each of 3 cell lines – MFSD5 WT-OE, TMEM241 WT-OE, SLC17A9 WT-OE – and 2 conditions – Dox-induced and uninduced). (B) Principal components analysis performed on normalized glycopeptide abundances of 18 glycoproteomics samples, whereby to account for possible differences in protein amount, each glycopeptide was normalized to the total abundance of the corresponding protein. (C) Dox-induced vs uninduced glycopeptide abundance $\log_2$ fold changes for 7 most abundant glycan compositions across 18 samples. Individual glycopeptides are plotted as discrete points and color-coded by significance. Bottom: stacked barplot indicating fraction of total glycopeptide signal contributed by each glycan composition, showing that these 7 compositions account for over 50% of the total glycopeptide signal. (D) Stacked barplot indicating number of glycopeptides attributed to each glycan composition, showing that ~40.6% (4845 out of 11,938) identified glycopeptides belong to the 7 most abundant glycan compositions. (E) Representative immunofluorescence images of MFSD5 WT-OE uninduced and doxycycline (Dox)-induced cells showing individual and merged channels for *Galanthus Nivalis* Lectin (GNL), LAMP1 and M6P staining with DAPI nuclear counterstain. (F) Cell surface *Vicia Villosa* Lectin (VVL) staining of $+/-$ Dox cells for an extended panel of cell lines overlapping between transcriptomic and metabolomic clusters. Each violin plot represents flow cytometry measurements of at least 30,000 cells pooled from 3 replicate wells, and horizontal bisecting lines indicate population geometric means. Effect sizes (Cohen's $d$) between uninduced and Dox-induced populations of each cell line are indicated by annotated brackets as follows: *$d > 0.5$; **$d > 1$; ***$d > 2$. The Cohen's $d$ values are: MFSD14B, 1.036; MFSD5, 1.095; SLC10A7, 0.645; SLC16A13, 0.469; SLC17A9, 1.022; SLC30A8, 0.781; SLC35A1, 0.22; SLC35B1, -0.141; SLC35B4, 1.622; SLC37A4, 1.466; SLC47A2, -0.068; SLC4A1, 0.316; SLC7A11, 0.113; TMEM241, 0.856; SLC35D3, 0.122; SLC35E4, 0.312. (G) intracellular GNL staining of $+/-$ Dox cells for an extended panel of cell lines overlapping between transcriptomic and metabolomic clusters. Each violin plot represents flow cytometry measurements of at least 30,000 cells pooled from 3 replicate wells, and horizontal bisecting lines indicate population geometric means. Effect sizes (Cohen's $d$) between uninduced and Dox-induced populations of each cell line are indicated by annotated brackets as follows: *$d > 0.5$; **$d > 1$; ***$d > 2$. The Cohen's d values are: MFSD14B, 2.106; MFSD5, 2.437; SLC10A7, 1.001; SLC16A13, 1.287; SLC17A9, 2.19; SLC30A8, 0.915; SLC35A1, 0.614; SLC35B1, 1.464; SLC35B4, 2.244; SLC37A4, 3.125; SLC47A2, 0.436; SLC4A1, 0.331; SLC7A11, 0.75; TMEM241, 1.336; SLC35D3, 0.386; SLC35E4, −0.025. (H) Flow cytometry histograms comparing cell surface VVL staining of SLC30A8 and SLC47A2 WT-OE $+/-$ Dox cells. Dashed boxes indicate $+$Dox cell populations with increased VVL staining. (I) Flow cytometry histograms comparing cell surface VVL staining of MFSD5, SLC30A8 wild-type, and SLC30A8 D110N-D224N (transport-deficient mutant) $+/-$ Dox cells. Dashed boxes indicate $+$Dox cell populations with increased VVL staining, which is absent in the transport-deficient mutant. (J) Principal components analysis of gene expression $\log_2$ fold change profiles of SLC30A8 D110N-D224N along with the 450 SLCs used in transcriptomic clustering.

