## [Peer Review File · Molecular Systems Biology]

Metabolic mapping of the human solute carrier superfamily

Tabea Wiedmer, Shao Teoh, Eirini Christodoulaki, Gernot Wolf, Chengzhe Tian, Vitaly Sedlyarov, Abigail Jarret, Philipp Leippe, Fabian Frommelt, Alvaro Ingles-Prieto, Sabrina Lindinger, Barbara Barbosa, Svenja Onstein, Christoph Klimek, Julio Garcia Murias, Iciar Serrano, Daniela Reil, Diana Santacruz, Mary Piotrowski, Stephen Noell, Christoph Bueschl, Huanyu Li, Gamma Chi, Stefan Mereiter, Tiago Oliveira, Josef Penninger, David Sauer, Claire Steppan, Coralie Viollet, Kristaps Klavins, J. Hannich, Ulrich Goldmann, and Giulio Superti-Furga

Corresponding author(s): Giulio Superti-Furga (gsuperti@cemm.oeaw.ac.at)

Review Timeline:

Submission Date:	17th Oct 24
Editorial Decision:	29th Nov 24
Revision Received:	30th Jan 25
Editorial Decision:	6th Mar 25
Revision Received:	28th Mar 25
Accepted:	31st Mar 25

Editor: Jingyi Hou

Transaction Report:

29th Nov 2024

Manuscript Number: MSB-2024-12687

Title: Metabolic mapping of the human solute carrier superfamily

Author: Tabea Wiedmer

Shao Teoh

Eirini Christodoulaki

Gernot Wolf

Chengzhe Tian

Vitaly Sedlyarov

Abigail Jarret

Philipp Leippe

Fabian Frommelt

Alvaro Ingles-Prieto

Sabrina Lindinger

Barbara Barbosa

Svenja Onstein

Christoph Klimek

Julio Garcia Murias

Iciar Serrano

Daniela Reil

Diana Santacruz

Mary Piotrowski

Stephen Noell

Christoph Bueschl

Huanyu Li

Gamma Chi

Stefan Mereiter

Tiago Oliveira

Josef Penninger

David Sauer

Claire Steppan

Coralie Viollet

Kristaps Klavins

J. Hannich

Ulrich Goldmann

Giulio Superti-Furga

Dear Giulio,

Thank you for submitting your work to Molecular Systems Biology. We have now heard back from the three reviewers who agreed to evaluate your manuscript. As you will see from the reports below, the reviewers acknowledge the interest of the study. They raise, however, a series of concerns, which we would ask you to address in a major revision.

While Reviewers #1 and #3 are generally supportive, Reviewer #2 raised several significant concerns. Specifically, Reviewer #2 highlighted that the current manuscript lacks a clear, cohesive narrative to emphasize the novel biology, a point also noted by Reviewer #3. Additionally, Reviewer #2 identified various issues with data presentation and requested further technical details, justifications, and discussions. Notably, Reviewer #2 suggested better integration of metabolomic and transcriptomic data to enhance the biological insights, and Reviewer #3 mentioned that including follow-up studies would strengthen the manuscript. Given these comments, and taking into account the constraints on your ability to revise, we encourage you to consider performing some additional computational analyses to address the concerns raised, refine the structure and flow of the manuscript to better highlight the novel findings, and explicitly discuss the lack of follow-up studies as both a limitation and a potential direction for future research.

All other issues raised by the reviewers need to be satisfactorily addressed as well. As you may already know, our editorial policy allows in principle a single round of major revision, and it is therefore essential to provide responses to the reviewers' comments that are as complete as possible. Please feel free to contact me in case you would like to discuss in further detail any of the issues raised by the reviewers.

On a more editorial level, we would ask you to address the following issues:

- Please provide a .docx formatted version of the manuscript text (including legends for main figures, EV figures and tables). Please make sure that the changes are highlighted to be clearly visible.
- Please provide individual production quality figure files as .eps, .tif, .jpg (one file per figure).
- Please provide a .docx formatted letter INCLUDING the reviewers' reports and your detailed point-by-point responses to their comments. As part of the EMBO Press transparent editorial process, the point-by-point response is part of the Review Process File (RPF), which will be published alongside your paper.
- Please note that all corresponding authors are required to supply an ORCID ID for their name upon submission of a revised manuscript.
- We replaced Supplementary Information with Expanded View (EV) Figures and Tables that are collapsible/expandable online (see examples in <http://msb.embopress.org/content/11/6/812>). A maximum of 5 EV Figures can be typeset. EV Figures should be cited as 'Figure EV1, Figure EV2' etc... in the text and their respective legends should be included in the main text after the legends of regular figures.

Additional Tables/Datasets should be labeled and referred to as Table EV1, Dataset EV1, etc. Legends have to be provided in a separate tab in case of .xls files. Alternatively, the legend can be supplied as a separate text file (README) and zipped together with the Table/Dataset file.

For the figures and tables that you do NOT wish to display as Expanded View figures, they should be bundled together with their legends in a single PDF file called *Appendix*, which should start with a short Table of Content. Each legend should be below the corresponding Figure/Table in the Appendix. Appendix figures and tables should be referred to in the main text as: "Appendix Figure S1, Appendix Figure S2, Appendix Table S1" etc. See detailed instructions regarding expanded view here: <https://www.embopress.org/page/journal/17444292/authorguide#expandedview>.

- Before submitting your revision, primary datasets (and computer code, where appropriate) produced in this study need to be deposited in an appropriate public database (see <http://msb.embopress.org/authorguide-dataavailability> <https://www.embopress.org/page/journal/17444292/authorguide#dataavailability>). Please remember to provide a reviewer password if the datasets are not yet public. The accession numbers and database should be listed in a formal "Data Availability" section (placed after Materials & Method) that follows the model below (see also <https://www.embopress.org/page/journal/17444292/authorguide#dataavailability>). Please note that the Data Availability Section is restricted to new primary data that are part of this study.
- # Data availability
- The datasets (and computer code) produced in this study are available in the following databases:
- RNA-Seq data: Gene Expression Omnibus GSE46843 (<https://www.ncbi.nlm.nih.gov/geo/query/acc.cgi?acc=GSE46843>)
 - [data type]: [name of the resource] [accession number/identifier/doi] ([URL or identifiers.org/DATABASE:ACCESSION])
- *** Note - All links should resolve to a page where the data can be accessed. ***

- At EMBO Press we ask authors to provide source data for the main figures. Our source data coordinator will contact you to discuss which figure panels we would need source data for and will also provide you with helpful tips on how to upload and organize the files.

- Our journal encourages inclusion of *data citations in the reference list* to directly cite datasets that were re-used and obtained from public databases. Data citations in the article text are distinct from normal bibliographical citations and should directly link to the database records from which the data can be accessed. In the main text, data citations are formatted as follows: "Data ref: Smith et al, 2001". In the Reference list, data citations must be labeled with "[DATASET]". A data reference must provide the database name, accession number/identifiers and a resolvable link to the landing page from which the data can be accessed at the end of the reference. Further instructions are available at .

- We updated our journal's competing interests policy in January 2022 and request authors to consider both actual and perceived competing interests. Please review the policy <https://www.embopress.org/competing-interests> and update your competing interests if necessary. Please use the heading "Disclosure statement and competing interests".

- All Materials and Methods need to be described in the main text using our 'Structured Methods' format. According to this format, the Methods section includes a Reagents and Tools Table (listing key reagents, experimental models, software and relevant equipment and including their sources and relevant identifiers) followed by a Methods and Protocols section describing the methods, ideally using a step-by-step protocol format. The aim is to facilitate adoption of the methodologies across labs.

Please download and fill our Reagents and Tools Table template (.docx), which you can find in our author guidelines: <https://www.embopress.org/page/journal/17444292/authorguide#structuredmethods>.

An example of a Method paper with Structured Methods can be found here:
<https://www.embopress.org/doi/10.15252/msb.20178071>.

-Regarding data quantification:

Please ensure to specify the name of the statistical test used to generate error bars and P values, the number (n) of independent experiments (please specify technical or biological replicates) underlying each data point and the test used to calculate p-values in each figure legend. Discussion of statistical methodology can be reported in the materials and methods section, but figure legends should contain a basic description of n, P and the test applied.

Graphs must include a description of the bars and the error bars (s.d., s.e.m.).

- Please provide a "standfirst text" summarizing the study in one or two sentences (approximately 250 characters, including space), three to four "bullet points" highlighting the main findings and a "synopsis image" (550px width and 400-600 px height, PNG format) to highlight the paper on our homepage.

Here are a couple of examples:

<https://www.embopress.org/doi/10.15252/msb.20199356>

<https://www.embopress.org/doi/10.15252/msb.20209475>

<https://www.embopress.org/doi/10.15252/msb.209495>

When you resubmit your manuscript, please download our CHECKLIST (<https://www.embopress.org/pb-assets/embo-site/EMBO%20Press%20Author%20Checklist-1642513524327.xlsx>) and include the completed form in your submission.

Please note that the Author Checklist will be published alongside the paper as part of the transparent process (<https://www.embopress.org/page/journal/17444292/authorguide#transparentprocess>).

If you feel you can satisfactorily deal with these points and those listed by the referees, you may wish to submit a revised version of your manuscript. Please attach a covering letter giving details of the way in which you have handled each of the points raised by the referees. A revised manuscript will be once again subject to review and you probably understand that we can give you no guarantee at this stage that the eventual outcome will be favorable.

I look forward to receiving the revised manuscript soon.

Kind regards,
Jingyi

Jingyi Hou, PhD
Scientific Editor
Molecular Systems Biology

We realize that it is difficult to revise to a specific deadline. In the interest of protecting the conceptual advance provided by the work, we recommend a revision within 3 months (27th Feb 2025). Please discuss the revision progress ahead of this time with the editor if you require more time to complete the revisions. Use the link below to submit your revision:

IMPORTANT: When you send your revision, we will require the following items:

1. the manuscript text in LaTeX, RTF or MS Word format
 2. a letter with a detailed description of the changes made in response to the referees. Please specify clearly the exact places in the text (pages and paragraphs) where each change has been made in response to each specific comment given
 3. three to four 'bullet points' highlighting the main findings of your study
 4. a short 'blurb' text summarizing in two sentences the study (max. 250 characters)
 5. a 'thumbnail image' (550px width and max 400px height, Illustrator, PowerPoint or jpeg format), which can be used as 'visual title' for the synopsis section of your paper.
 6. Please include an author contributions statement after the Acknowledgements section (see <https://www.embopress.org/page/journal/17444292/authorguide>)
 7. Please complete the CHECKLIST available at (<https://bit.ly/EMBOPressAuthorChecklist>).
- Please note that the Author Checklist will be published alongside the paper as part of the transparent process

(<https://www.embopress.org/page/journal/17444292/authorguide#transparentprocess>).

See also figure legend guidelines: <https://www.embopress.org/page/journal/17444292/authorguide#figureformat>

9. Please note that corresponding authors are required to supply an ORCID ID for their name upon submission of a revised manuscript (EMBO Press signed a joint statement to encourage ORCID adoption).

(<https://www.embopress.org/page/journal/17444292/authorguide#editorialprocess>)

Currently, our records indicate that the ORCID for your account is 0000-0002-0570-1768.

Link Not Available

11. Include a Reagents and Tools Table as part of the Methods section, which can be downloaded from our author guidelines (<https://www.embopress.org/page/journal/17444292/authorguide#structuredmethods>)

*** PLEASE NOTE *** As part of the EMBO Press transparent editorial process initiative (see our Editorial at <https://dx.doi.org/10.1038/msb.2010.72>), Molecular Systems Biology publishes online a Review Process File with each accepted manuscripts. This file will be published in conjunction with your paper and will include the anonymous referee reports, your point-by-point response and all pertinent correspondence relating to the manuscript. If you do NOT want this File to be published, please inform the editorial office at msb@embo.org within 14 days upon receipt of the present letter.

Reviewer #1:

The paper Metabolic Mapping of the Human Solute Carrier Superfamily focuses on systematically evaluating the metabolic and transcriptional functions of the human solute carrier (SLC) superfamily. The authors utilize a comprehensive omics approach, including targeted metabolomics and transcriptomics, to assess the effects of SLC overexpression in different isogenic cell backgrounds.

The study undertakes a large-scale investigation, covering 378 SLC transporters and evaluating their impact on intracellular metabolites and gene expression. This breadth allows for metabolic fingerprinting of the SLC superfamily.

By integrating metabolomics and transcriptomics, the authors successfully identify functional relationships between SLCs. For instance, they identify SLC45A4 as a novel polyamine transporter.

Beyond known substrates, the study discovers putative substrates for several SLCs that had no previously annotated functions, which could help advance therapeutic targeting.

With the targeted metabolomics approach, only 189 metabolites were evaluated, which may not capture the full diversity of substrates transported by SLCs. This leaves room for more comprehensive assays.

The authors acknowledge that SLC overexpression can induce cellular stress, potentially confounding the transcriptional and metabolic data. This highlights the challenge of distinguishing physiological effects from artifacts of overexpression.

The experiments are performed under standard cell culture conditions, which may not fully reflect the physiological transport functions of the SLCs, particularly in different tissue environments or disease states.

The paper provides a potentially useful resource for future SLC research, offering insights into previously uncharacterized transporters and their roles in metabolism.

Reviewer #2:

Wiedmer et al's manuscript provides quantitative metabolomics data for 189 metabolites and transcriptome profiling across hundreds of over-expression conditions in wildtype or knock-out backgrounds for solute carrier proteins (SLCs). They include bioinformatics, some import/export studies, and do glycoproteomics.

Clearly, the authors would like to present this manuscript (and accompanying manuscripts) as a resource to the community. In its current form, however, the manuscript is largely a data dump, lacking a cohesive story to highlight any new biology. There is a multitude of superficially discussed observations, but the biological findings are rare. The reader is lost in confusion of what cell lines/conditions are used for what analyses, as well as the scope and the conclusion of findings at any systems-level. It is not clear why after collecting 2 omics data types for >400 transporters there is a need to do technically complex glycoproteomics.

Some figures/subfigures are clearly wrong and/or at odds with the authors' interpretations. The application of some bioinformatic/statistical approaches central to the paper are not well justified and may not be suited, particularly given the limited number of metabolites or transcripts altered for any given SLC. Given the amount of transcriptomics data (and KO conditions and metabolomics data), it seems like the authors could draw more numerous, and more significant conclusions through data integration.

As a stand-alone scientific work, the manuscript does not stand up. The paper needs significant re-working of analyses, figures and text to be considered at MSB. Less would be more.

Major points:

- For the sheer scale of data collected, it's striking that no attempt was made to link metabolomics and transcriptomics.
- It is unclear what cell lines are used for metabolomics analyses and figures (ie parental HEK, HCT, huh7, etc.) and how datasets were normalized or harmonized for very different starting numbers of cells, or between cell lines, or both? What figures are from what expression condition (WT-OE or KO-OE)? Why did WT-OE and KO-OE need to be done? All of the subfigures are mixed and it is not clear what is learned from each dataset by itself. Moreover, is fig2EV from KO-OE or WT-OE dataset? Both datasets? It's so difficult to just reconstruct what data is shown in which case.
- There is no discussion of phenotypic data, i.e.: do KO cells vs OE cells have differences in proliferation? Are the observed differences secondary effects? It's surprising that no KO seems to be lethal, and this should be discussed because it might point to genetic buffering and question the whole approach or, if growth defects are present, affect the interpretation of the results.
- Fig 1H, panel "HEK293-SLC39A10-WT-OE" is wrong. This figure shows that over-expression of SLC39A10 induces a -2.5 log₂ fold-change in SLC39A10. How can the overexpression of a protein lead to a nearly 6-fold decrease in that same protein's transcript level? It's also striking that we don't see any of the over-expressed genes in any subpanels of Fig 1H where these genes would lend credibility and serve as expected, internal validations. Something seems off with this analysis as many transcripts have high fold change but very low significance (-logP<1).
- It's clear that the underlying metabolomics data shows limited metabolic changes as seen in Fig EV2. Most transporter over-expression strains have few increased/decreased metabolites. For some transporters (like SLC16A14) they affect related metabolites (carnitines). This also holds true at the transcript-level as seen in Fig1H. Therefore, why is it appropriate to do hierarchical clustering throughout the manuscript if the majority of changes are driven by very few metabolites or transcripts? The current hierarchical analysis biases towards metabolites that change very little and are not statistically significant. Do the transcript/metabolite data lead natural, discrete clusters? Are there any control cell lines, like GFP-dox+ or empty vector, to rule out things that are simply due to dox-induction?
- How do we interpret the fact that, for SLCs with annotated metabolites that could be observed by the metabolomics pipeline, only 57 of 141 (40%) could see changes in an annotated substrate? The authors conclude that they can see substrate abundance changes for the direct transported metabolite/substrate. However, the majority of the time this is not true and no further biological (or technical) explanations or limitations are given. It's unclear if metabolites one reaction away from the annotated metabolite are from the same 40% where the transported metabolite changes, or how much overlap there is and for what type of transporters. Are they good at seeing changes for amino acid transporters, and not steroid transporters? Or maybe those SLCs are not expressed in parental cell lines? Etc.
- There is no discussion in the results section about limitations across the number of SLCs tested, nor justification for including many transporters in the omics studies. For example, transcriptomics clearly identified changes in lipid metabolism, and many transition metal transporters were analyzed, but this is out of the scope of the polar metabolomics analyses or without justification. Why not limit the scope of the paper to transporters for which the title, "Metabolic Mapping..." can comment on, particularly if there are companion papers? The authors could probably draw more meaningful biological conclusions by filtering conditions to those they can reasonably comment on or at least discuss limitations explicitly here.
- Throughout the manuscript, "deregulation" and "dysregulation" are misused. Over-expressing a protein and altered metabolome/transcriptome does not imply something loses regulation, but simply that due to over-expression a cell

compensates or has a regulated way to deal with this perturbation. This is clearly seen in the zinc transporter example, where cells are suggested to have a regulated compensation mechanism. There is no mechanistic data in this paper directly showing direct dysfunction in regulation.

- Analytical considerations and limitations of metabolomics are not discussed. For example, in Fig EV1F: are these "significant metabolites" found because of analytical chemistry constraints of the metabolomics pipeline (chromatography, best MRM transitions, best ionization) or real biology or are they just the most abundant metabolites? What does significant mean here?
- Figure 4a and Figure 5a are not legible (text size) and don't strengthen the manuscript even though they take up half a page each. Are clusters different from their structural family clusters? Are any clusters not expected from their structural family or from known biology? Again, unclear if this is even a reasonable analysis given how few metabolites in any condition drive the differences in volcano plots.
- Figure 2D panels are mentioned once in text without interpretation. How do we understand metabolites only decreasing in a condition without anything increasing biologically?
- The use of Reactome pathway analysis on metabolomics data does not provide any definition or clarity into the biology of SLCs, particularly when there are more pathways than metabolites, and many metabolites are shared across pathways. Why does such an analysis need to be done if there are no presented examples with large, global changes?
- Figure 3 should say explicitly what "labeled" metabolite is. What is the difference between peak area and normalized peak area shown in Figure 3? Why are some subfigures shown as peak area and some as norm peak area?
- For Figure 3, why not show the volcano plot for SLC45A4? It's not clear to the reader that you now move to the OE-KO condition for this Figure. How does the volcano plot of OE-KO compare to the volcano plot for SLC45A4 in Figure 2d? do we see spermidine or putrescine changing in Figure 2d? For Figure 3e, does AMG treatment globally alter metabolite levels and it is not specific? For example, does central carbon metabolism also change significantly? How can you rule out that aminoguanidine has intracellular effects as it's also a nitric oxide synthase inhibitor? For Figure 3d, why not measure DAO levels and activity in these serums directly, to rule out contributions from other aspects of serum? These are assumptive indirect experiments.
- Figure EV3: It is unreasonable to see changes in intracellular sucrose levels given rapid sucrose/glucose flux into glycolysis. The authors should have looked at flux of labeled sucrose to a glycolytic metabolite to rule it out as a sucrose transporter. They could have also looked at an annotated sucrose transporter (for which they probably have metabolomics data). The authors cannot directly say this transporter does not transport sucrose given the presented data and lack of controls and assumptive considerations.
- Figure 3F/Figure 3EVI: we understand SLC45A4 as an exporter, but all liposome studies are import studies. How do we rationalize this difference? Is there any precedent for this? Would be helpful to have another transporter control, not just empty liposomes. More description of controls for SEC should be included in the main text to make sense of this data.
- Figure 5 could be presented in a much clearer way to tell the story about cluster 17, and show transcriptomics data for the SLCs of interest, and not a half-page dendrogram. Are the transcriptomics findings for cluster 17 and upregulation of other SLCs seen as a metabolic signature from the metabolomics dataset? Why do even more metabolomics here? What do the existing metabolomics data say?
- Figure 6 conclusions could be strengthened or made much clearer and explicit. Subfigures from 6a/EV6 could be re-arranged to make a more convincing story for MFSD5. Currently, this final figure says that some SLCs located to the Golgi, including an orphan effect glycosylation. Why not say more about the orphan MFSD5? What is significant for staining in E/F? Why is the largest fraction of glycopeptides (~50%) not commented on (panel D)? is this one type of glycosylation? Why focus on the most common glycosylation events? Are these the ones that change the most?

Minor Points:

- The introduction could offer more precise exemplary functions of SLC proteins and their regulation or importance to human health.
- The section "pipeline for functional profiling of SLC family" could be significantly condensed, clarified, and most aspects in its current form moved to methods.
- Some aspects of figures have no reason to be included: greyed out "genetic interactions" and "interaction proteomics" on Figure 1a, c and d are not informative for this paper.
- It would be great if the authors outlined the study design (number of biological replicates, etc) and explained more concisely what comparisons are being drawn between cell lines (ie, KO vs OE, etc)
- Some text comes across as condescending: no readership should care about the "ethos of RESOLUTE" (see end of introduction section), but instead the veracity and quality of the data and its interpretation (ie its scientific merit). Any potential publisher would have authors provide data as supplemental files or resources for such a manuscript-otherwise there is no merit for its evaluation.

Reviewer #3:

By employing a comprehensive approach involving the knockout, overexpression, and post-knockout overexpression of SLC family proteins alongside metabolomics and transcriptomics analyses, the present article has reached several insightful conclusions.

Primarily, metabolite profiling reveals that metabolomic enrichment serves as a reliable means to confirm established SLC substrates and suggests potential substrates for less understood SLCs. Specifically, the confirmation of the substrate specificity for SLC45A2 substantiates the validity of this technique as an alternate method for substrate prediction, distinct from AI-driven

approaches. This discovery introduces a fresh dimension to the arsenal of methods available for substrate determination, broadening the scope of investigative tools in the field.

The latter part of the article, through the integration of transcriptomics and metabolomics, illuminates the functional correlations between SLCs. For instance, Cluster 17 is shown to influence osmoregulation, subsequently affecting the expression of metabolic enzymes and SLCs along with alterations in various metabolites, thereby maintaining osmotic balance. While the proposed explanation is logically sound and plausible, crucial evidence is lacking—specifically, the identification of osmosensors that detect changes in osmolarity and how these fluctuations trigger subsequent transcriptional modifications. The inclusion of such key insights would significantly enhance the integrity of the argument.

Additionally, I suggest that the author refine their writing style to improve readability. Clearer articulation of concepts and smoother transitions between ideas will make the article more accessible to a broader audience, enhancing its overall impact. By combining metabolomics and transcriptomics, researchers develop a powerful tool to get a more nuanced understanding of SLC function.

Point-by-point response for the manuscript: Metabolic mapping of the human solute carrier superfamily.

Response to reviewers of “Metabolic mapping of the human solute carrier superfamily”

Below, the reviewers' comments are represented in black, followed by our responses in green. To simplify the review, we added line numbers referring to the document with 'track changes' in the “simple markup” view. Additionally, we have appended a list of all figure panels that were revised or had minor edits.

Reviewer #1:

The paper Metabolic Mapping of the Human Solute Carrier Superfamily focuses on systematically evaluating the metabolic and transcriptional functions of the human solute carrier (SLC) superfamily. The authors utilize a comprehensive omics approach, including targeted metabolomics and transcriptomics, to assess the effects of SLC overexpression in different isogenic cell backgrounds.

The study undertakes a large-scale investigation, covering 378 SLC transporters and evaluating their impact on intracellular metabolites and gene expression. This breadth allows for metabolic fingerprinting of the SLC superfamily.

By integrating metabolomics and transcriptomics, the authors successfully identify functional relationships between SLCs. For instance, they identify SLC45A4 as a novel polyamine transporter.

Beyond known substrates, the study discovers putative substrates for several SLCs that had no previously annotated functions, which could help advance therapeutic targeting.

With the targeted metabolomics approach, only 189 metabolites were evaluated, which may not capture the full diversity of substrates transported by SLCs. This leaves room for more comprehensive assays.

The authors acknowledge that SLC overexpression can induce cellular stress, potentially confounding the transcriptional and metabolic data. This highlights the challenge of distinguishing physiological effects from artifacts of overexpression.

The experiments are performed under standard cell culture conditions, which may not fully reflect the physiological transport functions of the SLCs, particularly in different tissue environments or disease states.

The paper provides a potentially useful resource for future SLC research, offering insights into previously uncharacterized transporters and their roles in metabolism.

We thank the reviewer for the insightful comments and positive assessment of our manuscript, and for constructive feedback on shortcomings in the study. We agree that with only 189 metabolites evaluated, we cannot cover the full diversity of SLC substrates, a point that we have now taken extra effort to discuss as a limitation of the study (lines 649-671). However, as exemplified by our deorphanization efforts on SLC45A4 as a polyamine transporter, the initial insights provided by our dataset can guide and motivate investigation into relevant metabolic pathways using complementary assays to interrogate the remaining metabolic space. This has also been added to the Discussion (lines 673-678).

We also agree that differential profiles acquired under standard cell culture conditions may not reflect physiological transport functions; or, conversely, some transport functions may fail to be reflected in the data under non-physiological conditions. We have taken care to mention this limitation in the Discussion (lines 644-648).

We hope that our changes sufficiently address the points raised, and thank the reviewer again for the constructive comments.

Overview of edits: lines 644-648, 649-671, 673-678

Reviewer #2:

Wiedmer et al's manuscript provides quantitative metabolomics data for 189 metabolites and transcriptome profiling across hundreds of over-expression conditions in wildtype or knock-out backgrounds for solute carrier proteins (SLCs). They include bioinformatics, some import/export studies, and do glycoproteomics.

Clearly, the authors would like to present this manuscript (and accompanying manuscripts) as a resource to the community. In its current form, however, the manuscript is largely a data dump, lacking a cohesive story to highlight any new biology. There is a multitude of superficially discussed observations, but the biological findings are rare. The reader is lost in confusion of what cell lines/conditions are used for what analyses, as well as the scope and the conclusion of findings at any systems-level. It is not clear why after collecting 2 omics data types for >400 transporters there is a need to do technically complex glycoproteomics.

Some figures/subfigures are clearly wrong and/or at odds with the authors' interpretations. The application of some bioinformatic/statistical approaches central to the paper are not well justified and may not be suited, particularly given the limited number of metabolites or transcripts altered for any given SLC. Given the amount of transcriptomics data (and KO conditions and metabolomics data), it seems like the authors could draw more numerous, and more significant conclusions through data integration.

As a stand-alone scientific work, the manuscript does not stand up. The paper needs significant re-working of analyses, figures and text to be considered at MSB. Less would be more.

We thank the reviewer for the thorough and detailed review of our manuscript. We greatly appreciate the time and effort invested in evaluating our work and providing constructive feedback. The 23 major and minor points that were outlined have been instrumental in guiding our revisions towards an improved manuscript.

While we understand the concerns regarding the clarity and focus of the manuscript, we would like to emphasize that studies presenting large-scale -omics datasets often prioritize creating a resource, with detailed biological findings emerging from subsequent follow-up studies. The biological findings presented in this study (SLC45A4 function in polyamine transport, SLC cluster with osmotic stress response, SLC cluster and orphan MFSD5 with glycosylation impact) were intended as examples to showcase the potential value of the data. The attempt to juggle presenting the data as a resource and to highlight multiple discrete biological findings has affected clarity and depth, as was rightfully pointed out. To address this, we have undertaken substantial revisions.

We hope that the extensively revised manuscript addresses the concerns sufficiently and convinces the reader of the scientific value of this work. We thank the reviewer again for the insightful comments and suggestions. Please find our detailed point-by-point response below:

Major points:

1. For the sheer scale of data collected, it's striking that no attempt was made to link metabolomics and transcriptomics.

Data-driven integration of different resources on SLCs, including the metabolomic profiles and transcriptomic profiles presented in this manuscript, is the focus of an accompanying manuscript (Goldmann et al). In addition, knowledge-driven integration using genome-scale metabolic models and knowledge graphs is currently worked on in a follow-up project. Because of this and the already dense content of the manuscript, we did not pursue data integration in-depth in this manuscript. However, we agree that a more systematic comparison of the metabolomics and transcriptomics datasets would also strengthen the current manuscript. Accordingly, we performed an analysis of the coherence between SLC clusterings derived from these two omics datasets, as well as their respective coherences with groupings based on SLC family and structural fold. We quantified coherence using normalized variation of information (NVI; (Meilă, 2007); new Fig. 5B). We also performed a decomposition analysis to find the contributions of individual pairs of clusters to the coherence to highlight the clusters showing the strongest overlaps between transcriptomics and metabolomics data sets (Fig. EV4E, Appendix Figure S3). The new analyses have been added to the Results at the end of the section on transcriptomics data clustering (lines 453-475) and details of the approach have been added to Methods (lines 1216-1226)

Overview of edits: lines 453-475, 1216-1226, Fig. 5B, Fig. EV4E, Appendix Figure S3

2. It is unclear what cell lines are used for metabolomics analyses and figures (ie parental HEK, HCT, huh7, etc.) and how datasets were normalized or harmonized for very different starting numbers of cells, or between cell lines, or both? What figures are from what expression condition (WT-OE or KO-OE)? Why did WT-OE and KO-OE need to be done? All of the subfigures are mixed and it is not clear what is learned from each dataset by itself. Moreover, is fig2EV from KO-OE or WT-OE dataset? Both datasets? It's so difficult to just reconstruct what data is shown in which case.

We re-structured and clarified Fig. 1B to provide a clearer overview of all the generated cell lines and data sets, as well as the subsets which were selected for this study and all the following analyses. The text was re-phrased and expanded accordingly (re-structured and re-phrased part: lines 149-239; expansion in lines 152-156, 161-165, 166-169, 177-181).

The rationale to use KO-OE and WT-OE cell line models is given in the results section (lines 166-169), as well as the criteria for selection in the methods section (lines 1035-1044). In brief, the appropriate cell line model for each SLC was chosen based on its expression across the panel of cancer cell lines. In case the SLC is endogenously expressed in one of the five cancer cell lines and a respective knock-out cell line was available, the KO-OE cell line model was chosen. Since cell line generation and metabolomics data measurements were performed in parallel, cases exist where a KO-OE model came available at a later point in time but was not measured in metabolomics – an unavoidable consequence of the work being carried out in a large consortium with many moving parts. In case the SLC is not endogenously expressed in any of the cancer cell lines, or it was not possible to derive a respective knock-out cell line, the WT-OE model was chosen. Due to budget and time constraints, it was not possible to acquire targeted metabolomics for each SLC-OE across all generated cell lines. We therefore aimed for measuring the effects of each SLC-OE in at least one cell line.

For all SLC superfamily-wide analyses, the data set included both analyses from 197 WT-OE and 181 KO-OE cell lines. Figure captions for Fig. 1, EV2, 4 and 5 were extended for clarification. We additionally

performed a principal component analysis on the entire data set (Fig. EV1E and Appendix Figure S1B, lines 204-209).

Metabolic profiles were only compared upon differential abundance analysis between induced and uninduced samples, which accounts for different cell numbers during seeding. Additionally, peak areas were normalized to internal standards and batch corrected as described in the methods.

Overview of edits: Fig. 1B, EV1E; Appendix Figure S1B; lines 149-239; Figure captions for Fig. 1, EV2, 4 and 5

3. There is no discussion of phenotypic data, i.e.: do KO cells vs OE cells have differences in proliferation? Are the observed differences secondary effects? It's surprising that no KO seems to be lethal, and this should be discussed because it might point to genetic buffering and question the whole approach or, if growth defects are present, affect the interpretation of the results.

Only a few SLCs are essential and therefore their knockout lethal. In the accompanying manuscript on genetic interactions of SLCs (Wolf and Leippe, et al. Figure EV2C therein), we found that six SLCs (SLC25A26, SLC7A1, MTCH2, SLC30A9, SLC25A3 and SLC35B1) are essential in HCT 116, out of a total of 258 expressed SLCs. This is based on both Cas12a and Cas9 pooled KO screening data (both Cas systems were employed in separate screens), specifically sgRNA depletion over three weeks (cutoff LFC < -2). Of those six essential SLCs, we managed to generate HCT 116 KO clones for only one of these (SLC7A1). Aside from those 6, the effects of single SLC KOs on proliferation rate were small, even when measured three weeks after lentiviral KO transduction (Wolf and Leippe et al. Figure EV2A and B therein). This information was added to the manuscript (lines 154-156). As detailed in the methods section (lines 1035-1044) and now additionally in the Results (lines 177-182), we proceeded with the SLC WT-OE whenever KO-OE cell line generation failed.

We always compare the effects of doxycycline-induced SLC-overexpression to uninduced cells in either WT or KO background. The induction is for a relatively short time frame (overnight) to minimize secondary effects, such as from differences in proliferation rates, adaptive or stress responses. However, we cannot categorically exclude secondary effects or stress responses due to high transgene expression levels. These points are discussed in the manuscript (lines 288-290 and lines 634-638). To fully exclude secondary effects, overexpression of transport deficient SLC mutants would be necessary. Information about transport deficient SLC mutants across the whole family does not exist to the best of our knowledge. The term "secondary effect" is also somewhat subjective as it is difficult to determine whether the downstream metabolic, transcriptomic or biological consequence of expressing an SLC is related to its 'natural' or 'intended' function. In any case, our metabolomics approach does detect concentration changes in both directly transported substrates and their metabolic conversions for overexpression of SLCs that have annotated substrates (lines 273-279, Figure EV1H), providing some assurance that our data does contain information related to the SLCs' transported substrates.

We re-structured and clarified Fig. 1B to provide a clearer overview of data generation.

Overview of edits: Fig. 1B, lines 154-156; 177-182

4. Fig 1H, panel "HEK293-SLC39A10-WT-OE" is wrong. This figure shows that over-expression of SLC39A10 induces a -2.5 log₂ fold-change in SLC39A10. How can the overexpression of a protein lead to a nearly 6-fold decrease in that same protein's transcript level? It's also striking that we

don't see any of the over-expressed genes in any subpanels of Fig 1H where these genes would lend credibility and serve as expected, internal validations. Something seems off with this analysis as many transcripts have high fold change but very low significance ($-\log P < 1$).

We respectfully disagree with the reviewer's strong assertion that the panels in Fig. 1H are wrong. As stated in the results (lines 156–159) and methods (lines 1027-1031), codon-optimized SLC sequences were used for the generation of overexpression cell lines. As such, we can differentiate overexpressed from endogenous SLC transcripts. These codon-optimized sequences were excluded during the differential expression analysis as the transcript counts for those overexpressed sequences are much higher than the endogenous ones and would not lead to a meaningful differential expression analysis. The method section on Transcriptomics data processing detailed that transcripts were mapped both against the human reference genome and against cDNA overexpression sequences. This data processing was already stated in the methods section, Differential expression analysis: "Genes with a total of less than 10 reads in all analyses were excluded as well as the codon-optimized cDNA sequences for 447 SLCs to consider only endogenously expressed SLC sequences." (lines 1148–1150).

The observed decrease in endogenous SLC39A10 transcript levels represents a biological observation, indicative of negative feedback regulation at the transcriptional level, rather than an error in the analysis. To ensure clarity for readers, we now additionally state in Figure caption 1H and line 291 that this analysis only considered endogenous gene transcripts.

Regarding the significance levels and fold changes, transcripts with high fold changes but low significance arise naturally in RNA-seq data due to variability in expression levels or lower statistical power for less abundant genes. Such cases reflect the inherent characteristics of transcriptomic datasets and are corrected for in the DESeq statistical analysis. However, we noticed that the figure panel 1H unintentionally presented the log₂ fold changes instead of the shrunken log₂ fold changes. We thank the reviewer for pointing this out and corrected the figure panel 1H accordingly.

We hope this clarification addresses the reviewer's concerns.

Overview of edits: line 291, Caption Figure 1H, Figure 1H

5. It's clear that the underlying metabolomics data shows limited metabolic changes as seen in Fig EV2. Most transporter over-expression strains have few increased/decreased metabolites. For some transporters (like SLC16A14) they affect related metabolites (carnitines). This also holds true at the transcript-level as seen in Fig1H. Therefore, why is it appropriate to do hierarchical clustering throughout the manuscript if the majority of changes are driven by very few metabolites or transcripts? The current hierarchical analysis biases towards metabolites that change very little and are not statistically significant. Do the transcript/metabolite data lead natural, discrete clusters? Are there any control cell lines, like GFP-dox+ or empty vector, to rule out things that are simply due to dox-induction?

While Fig. EV2 summarizes the largest metabolite changes among the 378 SLCs profiled, which provides a good starting point for inspecting SLCs eliciting changes in metabolites of interest, it does not allow to identify and visualize relationships (similarities) between the SLC profiles – hierarchical clustering is a widely used and conceptually uncomplicated approach that provided a means of accomplishing this.

We agree that by forcing the overall fold change magnitudes to be similar between cell lines we do run the risk of amplifying noise in cell lines with overall small Dox/uninduced fold changes. This noisiness also contributes to high within-cluster variability, i.e. low 'cluster discreteness', as can be seen from the rather low mean cluster silhouette widths (Fig. EV4A, B). On the other hand, biologically meaningful patterns do not need to arise from large, statistically significant changes in individual genes/metabolites only, but may also be reflected in small but concerted changes in many genes/metabolites – the principle behind methods such as GSEA and Ingenuity Pathway Analysis. Our analysis therefore represents a compromise between potential noise in the clustering and the ability to pick up small changes that contribute to SLC similarities.

We can argue from the results that our clustering analysis approach is not solely biased towards metabolites that change very little. For example, clusters 3, 20, 21 (discussed in Fig. 4B-D) are each clearly driven by one or two strongly changed metabolites. On the other hand, some clusters (for example metabolomics cluster 5 shown in Fig. 6A) are driven by a large number of smaller metabolite changes. That our analysis can find both types of clusters, whose metabolite changes are explainable biologically (the former as direct transported substrates or metabolic derivatives thereof, described in lines 420-430; the latter as metabolites connected to the perturbed biological process, described in lines 533-543) justifies our approach.

Overview of edits/relevant sections: lines 420-430, 533-543

6. How do we interpret the fact that, for SLCs with annotated metabolites that could be observed by the metabolomics pipeline, only 57 of 141 (40%) could see changes in an annotated substrate? The authors conclude that they can see substrate abundance changes for the direct transported metabolite/substrate. However, the majority of the time this is not true and no further biological (or technical) explanations or limitations are given. It's unclear if metabolites one reaction away from the annotated metabolite are from the same 40% where the transported metabolite changes, or how much overlap there is and for what type of transporters. Are they good at seeing changes for amino acid transporters, and not steroid transporters? Or maybe those SLCs are not expressed in parental cell lines? Etc.

In practice, as the reviewer will certainly appreciate, it is much easier to explain the presence of an observation than its absence, given the endless number of biological and technical factors that can contribute to a lack of detection. Moreover, the SLC-substrate annotations were manually curated from primary literature and critically evaluated (see lines 246-248 and accompanying manuscript by Goldmann, Wiedmer et al); however, some annotations are based on little evidence, relying on a single publication. In other words, a fraction of the SLC-substrate pairs from primary literature that we are accepting in good faith as "truth set" might be wrong or their annotation incomplete, for SLCs that are substrate annotated might transport additional substrates. Regardless, interpreting the absence of observed substrate changes would require a systematic experimental investigation into the biological and technical determinants affecting SLC-substrate pair detection in our steady-state metabolomics workflow. This would be technical mass-spectrometry analysis beyond the scope of this study. Instead, we focus on the many observations we successfully made rather than speculating on the ones we did not.

We clarified the wording in the beginning and end of this paragraph to point out that the approach did not intend and was not tailored towards the direct identification of novel substrates (lines 251-253 and 275-279).

We acknowledge the limitations of the approach, including restricted metabolite coverage, the measurement of steady-state conditions under adapted metabolic states, and cell culture conditions that may lack certain substrates. These points are explicitly mentioned in the results and discussion sections (lines 246-248, 276-277, and 640-655).

We added additional information about the overlap between SLCs for which we measured annotated substrates and metabolic conversions and the type of transporters that we found (lines 267-273).

Overview of edits: lines 251-253, 267-273, 275-279

7. There is no discussion in the results section about limitations across the number of SLCs tested, nor justification for including many transporters in the omics studies. For example, transcriptomics clearly identified changes in lipid metabolism, and many transition metal transporters were analyzed, but this is out of the scope of the polar metabolomics analyses or without justification. Why not limit the scope of the paper to transporters for which the title, "Metabolic Mapping..." can comment on, particularly if there are companion papers? The authors could probably draw more meaningful biological conclusions by filtering conditions to those they can reasonably comment on or at least discuss limitations explicitly here.

The Introduction was expanded and the Results sections describing Figure 1 and EV1 were re-structured, re-phrased and expanded to convey these points more clearly.

Briefly, the study was intentionally designed to perform -omics analyses across the entire SLC superfamily, which comprised 447 members at the time of conception. We measured 378 SLCs in metabolomics and 441 SLCs in transcriptomics. The only limitations related to inclusion or exclusion of SLCs were given by the cell line generation and methodology (costs and time).

We chose a fingerprinting approach and two data sets to complement each other to expand the coverage of metabolic pathways. Therefore, it is desirable that transcriptomic data cover pathways that are not covered by polar metabolomics. The observed transcriptional and metabolic changes may not be directly/metabolically related to certain substrates, but we hypothesize that SLCs with similar substrates and/or biological functions will lead to similar metabolic and/or transcriptomics profiles. Furthermore, given the methodological approach and its limitations we mainly focus on the consequences downstream of SLC expression. Limitations due to the coverage of the metabolomics method are discussed. Despite the limited coverage of metabolites, we found significant metabolite changes across all substrate classes. Furthermore, limiting the analysis to a subset of SLCs would risk cherry-picking data, potentially introducing bias and undermining the broader goal of creating an unbiased resource and analysis across the full SLC superfamily. Critiquing the "metabolic fingerprinting" rationale in hindsight is understandable; however, it is important to recognize that this study represents the first attempt to perform such analyses for SLCs at this large scale.

Overview of edits: lines 134-138; 149-239

8. Throughout the manuscript, "deregulation" and "dysregulation" are misused. Over-expressing a protein and altered metabolome/transcriptome does not imply something loses regulation, but simply that due to over-expression a cell compensates or has a regulated way to deal with this perturbation. This is clearly seen in the zinc transporter example, where cells are suggested to have a regulated compensation mechanism. There is no mechanistic data in this paper directly showing direct dysfunction in regulation.

We agree that our usage of the terms 'deregulation' and 'dysregulation' to indicate changes in metabolite or transcript levels was confusing, and in many cases do not reflect true mechanistic dysfunction in regulation. We have therefore replaced these terms with alternative terms e.g. 'change', 'increase/decrease', 'differential abundance' or 'differential expression' in the text.

Overview of edits: Replaced terms 'deregulation' and 'dysregulation' throughout the manuscript and figures.

9. Analytical considerations and limitations of metabolomics are not discussed. For example, in Fig EV1F: are these "significant metabolites" found because of analytical chemistry constraints of the metabolomics pipeline (chromatography, best MRM transitions, best ionization) or real biology or are they just the most abundant metabolites? What does significant mean here?

Metabolites are considered significantly changed if the adjusted p-value of the differential analysis of dox-induced and uninduced samples is below 0.05, as described in the methods. In other words, the first bar on aconitic acid in Figure panel EV1F, now moved to Appendix Figure S1C, means that its concentration is significantly different (meeting the p value cutoff) between induced and uninduced cells in about 40% of all measured SLC-OE cell lines.

Clearly, the targeted metabolomics and chromatography method influences the outcome of meeting significance testing, and the measurements are a result of both biological changes and technical constraints. We agree that the previous Fig. EV1F may be confusing, and the respective analysis should be examined and elaborated in more detail with respect to analytical limitations. To visualize and discuss dependencies on the constraints of the method in more detail, we moved the previous Fig. EV1F to the Appendix Figure S1C and added new analyses in Fig. EV1F and Appendix Figure S1D, which are discussed in lines 224-239.

In summary, the measurement of metabolite levels was influenced by the method and by the abundance of the metabolites in the samples. However, we did not observe a correlation of respective parameters with the frequency of significant change (Fig. EV1F), pointing towards underlying biological reason, which is for instance supported by transcriptomics for the TCA metabolites (described in lines 240-245).

Overview of edits: lines 224-239, EV1F, Appendix Figures 1C/D.

10. Figure 4a and Figure 5a are not legible (text size) and don't strengthen the manuscript even though they take up half a page each. Are clusters different from their structural family clusters? Are any clusters not expected from their structural family or from known biology? Again, unclear if this is even a reasonable analysis given how few metabolites in any condition drive the differences in volcano plots.

Figures 4 and 5 are anchors to discuss our clustering analyses of metabolomic and transcriptomic profiles, respectively. The dendrograms serve to visualize the similarities between individual SLCs as well as between clusters. Fig. 5A additionally shows Reactome metabolic pathway-related gene expression in transcriptomic clusters. As such, we do not think that these dendrograms can be omitted without significantly changing the underlying message of these figures. While it is difficult to read SLC names on the dendrograms due to the small size, we have ensured that the figures' resolutions are sufficient for the names to be readable when zoomed in and data tables containing the underlying cluster memberships are provided. The dendrograms are also anchors for discussion of individual clusters.

As mentioned above, we have performed a coherence analysis between metabolomics and transcriptomics clusterings, and also between these clusterings and SLC groupings by family or structural fold (new Fig. 5B). The results indicate that there is high coherence (low NVI) between SLC family and structural fold, weaker coherence (moderate NVI) between the two omics clusterings, as well as between either omics clustering and SLC family, and very weak coherence (high NVI) between omics clusterings with protein fold. In other words, metabolomics and transcriptomics clusters tend to be *not* expected from their structural families as defined by protein folds. This new analysis has been added to the manuscript (lines 459-475). Regarding known biology, the glycosylation-related cluster described in Fig. 6 is perhaps the best example containing SLCs (metal transporters, vesicular ATP transporter SLC17A9, orphan MFSD5 etc) whose links to glycosylation are poorly understood or completely unknown.

Overview of edits: lines 459-475, Fig. 5B

11. Figure 2D panels are mentioned once in text without interpretation. How do we understand metabolites only decreasing in a condition without anything increasing biologically?

We have added more elaboration on the four orphan SLCs highlighted in Fig. 2C (lines 334-344), including suggesting that SLC16A14 may have a role in carnitine efflux, which is later discussed in the section on the transcriptomic cluster 17 showing an osmolyte transcriptional response. This later section was also modified accordingly (lines 488-490). For ease and clarity when referring to specific SLCs among these four, we split Fig. 2D into four separate panels – Fig. 2D (SLC16A14), E (SLC35F6), F (SLC45A4), G (MFSD5). We added to the discussion some possible reasons why only metabolite decreases were detected for SLC16A14, including limited method coverage potentially missing compensatory increased metabolites, and decreased cell number/volume. We additionally mention the inadequacy of coverage of our analytical method to detect potential nucleotide sugar changes by SLC35F6 (lines 654-671).

Overview of edits: lines 334-344, 488-490, 654-671, Fig. 2D-G

12. The use of Reactome pathway analysis on metabolomics data does not provide any definition or clarity into the biology of SLCs, particularly when there are more pathways than metabolites, and many metabolites are shared across pathways. Why does such an analysis need to be done if there are no presented examples with large, global changes?

The aim of Fig. 1C is to present general regulatory effects across all cell lines / SLCs, which is of course limited by the coverage of the method. Ideally, a larger metabolite panel would provide better resolution on a pathway level, but a low-resolution picture is not necessarily useless as it still narrows down to multiple metabolic pathways that can be affected by a given metabolite. Importantly, these analyses

enabled us to visualize the complementarity of the metabolomics and transcriptomics data sets in their coverage of the metabolic pathways which are affected by SLC expression and measured by the respective method. We re-arranged and added further clarification to the description of this analysis and results (lines 210-223).

Overview of edits: lines 210-223

13. Figure 3 should say explicitly what "labeled" metabolite is. What is the difference between peak area and normalized peak area shown in Figure 3? Why are some subfigures shown as peak area and some as norm peak area?

We modified the figure legends for Fig. 3B and Fig. EV3D to indicate the specific isotopomers, eg. m+5 for ¹³C¹⁵N-GABA (from ¹³C¹⁵N-arginine) or m+4 for ¹³C-GABA (from ¹³C-glutamic acid). We additionally added annotations on the chemical structures in Fig. 3A and Fig. EV3D (upper part) to indicate the C and N atoms that are potentially labeled from ¹³C¹⁵N-Arg or ¹³C-Glu. An explanation of 'normalized peak area' (clarifying that it represents peak area values normalized to both internal standard compound and cellular protein content) was added to the figure legends.

Overview of edits: Fig. 3B, Fig. EV3D

14. For Figure 3, why not show the volcano plot for SLC45A4? It's not clear to the reader that you now move to the OE-KO condition for this Figure. How does the volcano plot of OE-KO compare to the volcano plot for SLC45A4 in Figure 2d? do we see spermidine or putrescine changing in Figure 2d? For Figure 3e, does AMG treatment globally alter metabolite levels and it is not specific? For example, does central carbon metabolism also change significantly? How can you rule out that aminoguanidine has intracellular effects as it's also a nitric oxide synthase inhibitor? For Figure 3d, why not measure DAO levels and activity in these serums directly, to rule out contributions from other aspects of serum? These are assumptive indirect experiments.

SLC45A4 volcano plot in Fig. 2D (now **Fig. 2F**) is indeed HCT 116 SLC45A4 KO-OE; we reworded the opening line of the SLC45A4 section to make this more explicitly clear (lines 350-351). Spermidine and putrescine were not covered in the original metabolomics data and were additionally measured with a different targeted method; we added to the text to make this explicitly clear (lines 355-357). We agree that there is a possibility of AMG having other effects on metabolism and we have now added it to text (lines 391-392); however, we do not think it is necessary to experimentally investigate this possibility since our main interest is in the polyamine transport function of SLC45A4, which is clearly unaffected by AMG (Fig. 3E). Regarding direct measurement of DAO activity, we do not think it is necessary for two reasons: first, the putrescine export function of SLC4A4 was clearly unaffected by serum presence or type (Fig. EV3F); and second, addition of porcine DAO to media with either FBS or horse serum clearly and strongly boosted SLC45A4-dependent GABA production, directly evidencing the role of DAO in GABA synthesis (Fig. EV3G).

Overview of edits: lines 350-351, 355-357, 391-392

15. Figure EV3: It is unreasonable to see changes in intracellular sucrose levels given rapid sucrose/glucose flux into glycolysis. The authors should have looked at flux of labeled sucrose to a glycolytic metabolite to rule it out as a sucrose transporter. They could have also looked at an annotated sucrose transporter (for which they probably have metabolomics data). The authors

cannot directly say this transporter does not transport sucrose given the presented data and lack of controls and assumptive considerations.

We agree with the reviewer that our data does not disprove sucrose transport by SLC45A4, and have added to the Discussion the rapid glycolytic assimilation of uptaken sucrose as a possible reason for not observing differences in sucrose accumulation in SLC45A4 OE, as well as the reviewer's suggestion of uptake experiments using stable isotopically labelled sucrose to probe sucrose uptake (lines 692-697). As it is beyond the scope of the current paper to conclusively disprove sucrose transport, this will not be experimentally followed up; however, we have made sure to modify the Discussion text to avoid any negative claims of sucrose transport (lines 690-692). We also clarified in the Discussion that SLC45A4 mediated polyamine efflux is unlikely to be connected to sucrose transport, since 1) putrescine export occurred in serum-free media which does not contain sucrose (Fig. 3D), and 2) cell-free proteoliposome uptake demonstrated direct putrescine transport (Fig. 3F), again in the absence of sucrose (lines 695-700).

Overview of edits: lines 690-700

16. Figure 3F/Figure 3EVI: we understand SLC45A4 as an exporter, but all liposome studies are import studies. How do we rationalize this difference? Is there any precedent for this? Would be helpful to have another transporter control, not just empty liposomes. More description of controls for SEC should be included in the main text to make sense of this data.

The direction of transport in the proteoliposomes is not expected to be a major concern since transporters are reversible, and the direction of transport is defined by the electrochemical gradient of the substrate(s) (Keller *et al*, 2008). Furthermore, the direction of transport for some protein molecules may align with the native protein, as proteins insert in both inward- and outward-facing orientations using the standard freeze-thaw method (Scalise *et al*, 2013). Therefore, by following well-established protocols, we have focused on measuring fundamental transport activity in vitro (Vickers *et al*, 1999; Komatsu *et al*, 2011).

While, at first appearance, proteoliposomes with another SLC might provide an interesting point of comparison, this actually probes a subtly different hypothesis that is more difficult to address. Guided by our earlier experiments, we have focused on the testable scientific question of deorphanizing the target by asking, "Can SLC45A4 transport putrescine?" To address this, we used empty liposomes as the negative control, allowing us to directly validate that SLC45A4 can catalyze transport across the otherwise putrescine-impermeable lipid bilayer. This mimics the native state/function of the protein in a cellular membrane and follows examples from literature where SLC proteoliposomes are compared to empty liposomes for measuring the transport activity in vitro, e.g. (Komatsu *et al*, 2011; Wang *et al*, 2015; Hiasa *et al*, 2014; Juge *et al*, 2006). On the other hand, comparing SLC45A4's putrescine transport to another SLC asks the subtly different question "Is SLC45A4's transport of putrescine biologically significant, relative to another transporter?" However, fundamentally, putrescine transport will vary with each SLC's selectivity. Therefore, interpreting such in vitro results will depend highly on the arbitrary SLC selected for comparison. Furthermore, the biological value of such results is limited, as cellular transport of putrescine results from the expression level and selectivity for all SLCs expressed, which will vary by tissue. Therefore, an additional SLC used for comparison is inappropriate for our in vitro studies. We have added a brief discussion of this to the manuscript (lines 700-703).

We have added more details of the internal liposome marker (glycine) and external buffer marker (PIPES) to the Results (lines 397-403).

Overview of edits: lines 397-403, 700-703

17. Figure 5 could be presented in a much clearer way to tell the story about cluster 17, and show transcriptomics data for the SLCs of interest, and not a half-page dendrogram. Are the transcriptomics findings for cluster 17 and upregulation of other SLCs seen as a metabolic signature from the metabolomics dataset? Why do even more metabolomics here? What do the existing metabolomics data say?

The main purpose of Figure 5 was to show our clustering analysis of transcriptomic profiles, with cluster 17 as one picked example to illustrate the kind of hypothesis we can derive from the clusters. We cannot remove the clustering dendrogram as it keeps the focus on the overall clustering, in addition to showing other information such as Reactome pathway-related gene expression.

Transcriptomic cluster 17 members are not found as a single cluster in metabolomics data, but are instead found in several clusters: cluster 1 (SLC6A5, SLC6A15, SLC6A19), cluster 4 (SLC6A9, SLC6A14) and cluster 7 (SLC38A2, SLC38A4, SLC6A12). It is unsurprising that there is some difference in the metabolomics clustering, which is more affected by the SLCs' specific substrates in addition to the perturbed osmolytes; in addition, as mentioned below, some osmolytes were not measured in the original metabolomics data. However, there is still a substantial overlap in information between transcriptomic cluster 17 and the metabolomics clusters, as shown by transcriptomic cluster 17 having one of the highest mutual information contributions to the inter-clustering coherence analysis (new Fig. EV4E, Appendix Figure S3).

More metabolomics was done because some relevant metabolites e.g. myo-inositol and glycerophosphorylcholine (GPC) were not all covered in the original metabolomics data set. We have added to the text to clarify this (lines 504-507). Also, the transcriptomics data was from HEK293 WT-OE cell lines but different cell lines were used for some metabolomics analyses: (SLC5A3 was an HCT 116 KO-OE cell line and SLC6A9 was a 132N1 KO-OE) and we wanted to validate our transcriptomic findings using the same HEK293 cell lines.

Overview of edits: Fig. EV4E, Appendix Figure S3, lines 504-507

18. Figure 6 conclusions could be strengthened or made much clearer and explicit. Subfigures from 6a/EV6 could be re-arranged to make a more convincing story for MFSD5. Currently, this final figure says that some SLCs located to the Golgi, including an orphan effect glycosylation. Why not say more about the orphan MFSD5? What is significant for staining in E/F? Why is the largest fraction of glycopeptides (~50%) not commented on (panel D)? is this one type of glycosylation? Why focus on the most common glycosylation events? Are these the ones that change the most?

Subfigures have been revised and rearranged, including a complete rework of panel 6A to clearly show the overlap between relevant transcriptomic and metabolomic clusters. A heatmap of GSEA results for the transcriptomic cluster members has been added to highlight pathway-level transcriptomic changes in Golgi and glycosylation related terms. Please note also that Fig. EV6 has been reassigned as EV5 due to the editorial limit on number of Extended View figures.

The aim of this figure/section is to show an example whereby combined analysis of transcriptomic and metabolomic data, utilizing information from both the clustering patterns of the SLCs (i.e. feature enrichment in the clusters) and the altered genes and metabolites themselves, reveals a molecular or biological process that a group of SLCs have an impact on – in this case, cellular glycosylation. It is indeed a good idea to highlight that the orphan MFSD5 is found among this group of SLCs, pointing to a potential role in Golgi function or other glycosylation-related process. Accordingly, the final panel has been reworked to focus on MFSD5. The entire section on the glycosylation cluster has been rewritten to reflect the restructuring, as well as incorporate findings from our new clustering coherence analysis (lines 522-609). However, further investigation into the molecular function of MFSD5 is beyond the scope of this paper.

Traditional statistical testing between flow cytometry populations (>10000 data points per group) would lead to very small p-values for every comparison, so we opted to show effect sizes instead. We have now added annotation of the effect size (Cohen's d) on the violin plots. Clarifications were added to the legends of Fig. 6 and EV5, as well as the Methods section (lines 1335-1338).

The 'largest fraction (~50%)' in Fig. 6D (now Fig. EV5C) is not one type of glycosylation. In this panel, we wanted to highlight the top 7 most common glycan compositions, which together make up > 50% of total signal intensity. The remaining glycopeptides consist of 318 different glycan compositions but together account for only < 50% of the total signal and were lumped together. We reasoned that looking at changes in the most abundant compositions would provide insight on the SLCs' overall impact on glycosylation, so we examined log₂FCs for the 7 most abundant compositions and found the corresponding glycopeptides are broadly increased in +Dox cells.

Overview of edits: Fig. 6, lines 522-609, 1335-1338

Minor Points:

19. The introduction could offer more precise exemplary functions of SLC proteins and their regulation or importance to human health.

We expanded the introduction to contain a prime example of SLC regulation, that is the insulin-stimulated SLC2A4 translocation, relevant to insulin resistance and type 2 diabetes (lines 97-100).

Overview of edits: lines 97-100

20. The section "pipeline for functional profiling of SLC family" could be significantly condensed, clarified, and most aspects in its current form moved to methods.

In response to the major point #2 we re-structured and clarified Fig. 1B to provide a clearer overview of all the cell lines and data sets, as well as the subsets which were selected for this study and the following analyses. The text was re-phrased accordingly to make it more concise (section "Pipeline for functional profiling of SLC superfamily"). We reasoned that it is important to clearly explain the basis for the study in this section as the lack of clarity was mentioned in several major points.

Overview of edits: lines 149-209

21. Some aspects of figures have no reason to be included: greyed out "genetic interactions" and "interaction proteomics" on Figure 1a, c and d are not informative for this paper.

We revised Fig. 1A to illustrate more clearly its purpose, i.e. to provide an overview of the RESOLUTE paper collection. The respective figure was included the accompanying manuscripts as well (Frommelt, Ladurner et al; Wolf, Leippe et al; Goldmann, Wiedmer et al.) to highlight their complementarity and connectivity.

The aim of Fig. 1C and 1D is to present general regulatory effects across all cell lines / SLCs, as detailed in the response to point 12. These analyses enabled us to visualize the complementarity of the metabolomics and transcriptomics data sets in their coverage of the metabolic pathways which are affected by SLC expression and measured by the respective method. We re-arranged and added further clarification to the description of this analysis and results to explain their relevance more clearly (lines 210-223).

Overview of edits: lines 210-223, Fig. 1A, B, C

22. It would be great if the authors outlined the study design (number of biological replicates, etc) and explained more concisely what comparisons are being drawn between cell lines (ie, KO vs OE, etc)

In response to the major point #2 we re-structured and clarified Fig. 1B to provide a clearer overview of all the cell lines and data sets, as well as the subsets which were selected for this study and all the following analyses. The comparisons being made are indicated under "Conditions". Additionally, we extended the respective figure captions and text to include more information such as biological replicates).

Overview of edits: lines 166-209, Figure caption 1B, Fig. 1B

23. Some text comes across as condescending: no readership should care about the "ethos of RESOLUTE" (see end of introduction section), but instead the veracity and quality of the data and its interpretation (ie its scientific merit). Any potential publisher would have authors provide data as supplemental files or resources for such a manuscript-otherwise there is no merit for its evaluation.

We rephrased the referenced text at the end of the introduction section (line 139). Supplemental files of the data as well as links to the analyses, presented as dashboards on the webpage, have been provided with the initial manuscript submission (listed in lines 1469-1482). Raw data is deposited at public repositories.

Overview of edits: line 139

Reviewer #3:

By employing a comprehensive approach involving the knockout, overexpression, and post-knockout overexpression of SLC family proteins alongside metabolomics and transcriptomics analyses, the present article has reached several insightful conclusions.

Primarily, metabolite profiling reveals that metabolomic enrichment serves as a reliable means to confirm established SLC substrates and suggests potential substrates for less understood SLCs. Specifically, the confirmation of the substrate specificity for SLC45A2 substantiates the validity of this technique as an alternate method for substrate prediction, distinct from AI-driven approaches. This

discovery introduces a fresh dimension to the arsenal of methods available for substrate determination, broadening the scope of investigative tools in the field.

The latter part of the article, through the integration of transcriptomics and metabolomics, illuminates the functional correlations between SLCs. For instance, Cluster 17 is shown to influence osmoregulation, subsequently affecting the expression of metabolic enzymes and SLCs along with alterations in various metabolites, thereby maintaining osmotic balance. While the proposed explanation is logically sound and plausible, crucial evidence is lacking—specifically, the identification of osmosensors that detect changes in osmolarity and how these fluctuations trigger subsequent transcriptional modifications. The inclusion of such key insights would significantly enhance the integrity of the argument.

Additionally, I suggest that the author refine their writing style to improve readability. Clearer articulation of concepts and smoother transitions between ideas will make the article more accessible to a broader audience, enhancing its overall impact.

By combining metabolomics and transcriptomics, researchers develop a powerful tool to get a more nuanced understanding of SLC function.

We thank the reviewer for the insightful comments and positive assessment of our manuscript, and for constructive feedback on shortcomings in the study. We agree that further investigation into the mechanism of osmosensing and regulatory mechanisms for the putative osmoregulated genes of Cluster 17 would be interesting topics for further investigation. However, detailed investigation into this subject was beyond the scope of the present study, since the purpose was to highlight examples of SLC-driven biology that could be discovered from our data set. Instead, we have added discussion of a possible mechanism driven by the known tonicity-responsive enhancer binding protein (TonEBP / NFAT5) to the Discussion (lines 728-742) in hopes of motivating future studies into the mechanism.

Overview of edits: lines 728-742

In response to the suggestion to refine the writing and improve readability, we re-structured, re-phrased and expanded all the Results sections with a focus to make concepts clearer and improve the connectivity between different parts of the study.

Overview of edits: Most significant revision of text in lines 149-279, 459-491, 522-609, Figures 1A/B, EV1, EV4, 6, EV5 and Appendix Figure S1.

We hope that our changes sufficiently address the points raised, and thank the reviewer again for the constructive comments.

References

- Hiasa M, Miyaji T, Haruna Y, Takeuchi T, Harada Y, Moriyama S, Yamamoto A, Omote H & Moriyama Y (2014) Identification of a mammalian vesicular polyamine transporter. *Sci Rep* 4: 6836
- Juge N, Yoshida Y, Yatsushiro S, Omote H & Moriyama Y (2006) Vesicular glutamate transporter contains two independent transport machineries. *J Biol Chem* 281: 39499–39506
- Keller T, Schwarz D, Bernhard F, Dötsch V, Hunte C, Gorboulev V & Koepsell H (2008) Cell free expression and functional reconstitution of eukaryotic drug transporters. *Biochemistry* 47: 4552–4564
- Komatsu T, Hiasa M, Miyaji T, Kanamoto T, Matsumoto T, Otsuka M, Moriyama Y & Omote H (2011) Characterization of the human MATE2 proton-coupled polyspecific organic cation exporter. *Int J Biochem Cell Biol* 43: 913–918
- Meilä M (2007) Comparing clusterings—an information based distance. *J Multivar Anal* 98: 873–895
- Scalise M, Pochini L, Giangregorio N, Tonazzi A & Indiveri C (2013) Proteoliposomes as tool for assaying membrane transporter functions and interactions with xenobiotics. *Pharmaceutics* 5: 472–497
- Vickers MF, Mani RS, Sundaram M, Hogue DL, Young JD, Baldwin SA & Cass CE (1999) Functional production and reconstitution of the human equilibrative nucleoside transporter (hENT1) in *Saccharomyces cerevisiae*. Interaction of inhibitors of nucleoside transport with recombinant hENT1 and a glycosylation-defective derivative (hENT1/N48Q). *Biochem J* 339 (Pt 1): 21–32
- Wang S, Tsun Z-Y, Wolfson RL, Shen K, Wyant GA, Plovanich ME, Yuan ED, Jones TD, Chantranupong L, Comb W, *et al* (2015) Metabolism. Lysosomal amino acid transporter SLC38A9 signals arginine sufficiency to mTORC1. *Science* 347: 188–194

6th Mar 2025

Manuscript Number: MSB-2024-12687R

Title: Metabolic mapping of the human solute carrier superfamily

Author: Tabea Wiedmer

Shao Teoh

Eirini Christodoulaki

Gernot Wolf

Chengzhe Tian

Vitaly Sedlyarov

Abigail Jarret

Philipp Leippe

Fabian Frommelt

Alvaro Ingles-Prieto

Sabrina Lindinger

Barbara Barbosa

Svenja Onstein

Christoph Klimek

Julio Garcia Murias

Iciar Serrano

Daniela Reil

Diana Santacruz

Mary Piotrowski

Stephen Noell

Christoph Bueschl

Huanyu Li

Gamma Chi

Stefan Mereiter

Tiago Oliveira

Josef Penninger

David Sauer

Claire Steppan

Coralie Viollet

Kristaps Klavins

J. Hannich

Ulrich Goldmann

Giulio Superti-Furga

Dear Giulio,

Thank you for submitting your revised manuscript to Molecular Systems Biology. We have now received the enclosed report from two Reviewers who agreed to re-assess your work. Since the original Reviewer #2 is unable to re-review the manuscript, we asked Reviewer #1 to also review the your responses to Reviewer #2's main points.

As you will see below, Reviewer #1 thinks that the concerns of Reviewer #2 have been adequately addressed. Therefore, I am pleased to inform you that we will be able to accept your manuscript pending the following amendments:

1. Please remove the figures from the manuscript file. Figure legends should remain replaced below the References.
2. Remove the Authors contribution section from the manuscript file.
3. Data availability: Please provide specific URLs for MTBLS10077; MTBLS11393, PRJEB81360 datasets in the data availability statement.
4. Funding: All the information from the Comments box needs to be included in the list of funders using the "More Funders" option,
5. The synopsis image is too large. Please provide an updated version with the specified dimension(550px width and 400-600 px height, PNG format).
6. "Material and Methods" should be renamed to "Methods".

7. Appendix: In Table of Content, please add page numbers for the listed items.
8. "Supplementary information" should be removed from manuscript file.
9. Section order should be corrected: Title page - Abstract & Keywords - Introduction - Results - Discussion - Methods - Data Availability - Acknowledgements - Disclosure and Competing Interests Statement - References - Figure Legends - Table(s) - Expanded View Figure Legends.
10. Tables EV1-EV3 and EV6-EV7 should be renamed to Dataset EV1-EV5 . Source file names, titles, legends and manuscript callouts all need to be updated to Dataset EV1-EV5; legends should remain as separate tabs/sheets in each Excel file.
11. Please address the following issues in figure legends
 - Please note that the exact p values are not provided in the legends of figures 1G, 2A, 3C, D, E F; EV3 E, F, G, I.
 - Please indicate the statistical test used for data analysis in the legends of figures 1E, F, G, H; 2B-G.
 - Please indicate what */ **/ ***/ **** represents; if this represents p value(s), please indicate the statistical test used and where appropriate the exact p value in the legend(s) of figure(s) 6D-F; EV5 F, G.
 - Please note that the box plots need to be defined in terms of minima, maxima, centre, bounds of box and whiskers, and percentile in the legends of figures 2A
 - Please note that information related to n is missing in the legends of figures 2A-C, 6C, D, E, EV5 F
 - Please note that the error bars are not defined in the legends of figures EV1 H.

When you resubmit your manuscript, please download our CHECKLIST (<https://bit.ly/EMBOPressAuthorChecklist>) and include the completed form in your submission. *Please note* that the Author Checklist will be published alongside the paper as part of the transparent process (<https://www.embopress.org/page/journal/17444292/authorguide#transparentprocess>)

Click on the link below to submit your revised paper.

Thank you for submitting this interesting paper to Molecular Systems Biology.

Yours sincerely,
Jingyi

Jingyi Hou, PhD
Senior Editor
Molecular Systems Biology

If you do choose to resubmit, please click on the link below to submit the revision online before 5th Apr 2025.

IMPORTANT: When you send your revision, we will require the following items:

1. the manuscript text in LaTeX, RTF or MS Word format
2. a letter with a detailed description of the changes made in response to the referees. Please specify clearly the exact places in the text (pages and paragraphs) where each change has been made in response to each specific comment given
3. three to four 'bullet points' highlighting the main findings of your study
4. a short 'blurb' text summarizing in two sentences the study (max. 250 characters)
5. a 'thumbnail image' (550px width and max 400px height, Illustrator, PowerPoint or jpeg format), which can be used as 'visual title' for the synopsis section of your paper.

6. Please include an author contributions statement after the Acknowledgements section (see <https://www.embopress.org/page/journal/17444292/authorguide#manuscriptpreparation>)

7. Please complete the CHECKLIST available at (<https://bit.ly/EMBOPressAuthorChecklist>). Please note that the Author Checklist will be published alongside the paper as part of the transparent process (<https://www.embopress.org/page/journal/17444292/authorguide#transparentprocess>).

See also figure legend guidelines: <https://www.embopress.org/page/journal/17444292/authorguide#figureformat>

9. Please note that corresponding authors are required to supply an ORCID ID for their name upon submission of a revised manuscript (EMBO Press signed a joint statement to encourage ORCID adoption).

(<https://www.embopress.org/page/journal/17444292/authorguide#editorialprocess>)

Currently, our records indicate that the ORCID for your account is 0000-0002-0570-1768.

Link Not Available

10. Include a Reagents and Tools Table as part of the Methods section, which can be downloaded from our author guidelines (<https://www.embopress.org/page/journal/17444292/authorguide#structuredmethods>)

*** PLEASE NOTE *** As part of the EMBO Press transparent editorial process initiative (see our Editorial at <https://dx.doi.org/10.1038/msb.2010.72> , Molecular Systems Biology will publish online a Review Process File to accompany accepted manuscripts. When preparing your letter of response, please be aware that in the event of acceptance, your cover letter/point-by-point document will be included as part of this File, which will be available to the scientific community. More information about this initiative is available in our Instructions to Authors. If you have any questions about this initiative, please contact the editorial office (msb@embo.org).

Reviewer #1:

The authors have addressed my original concerns and the paper is acceptable.

Overall I think the authors have made a significant effort to address reviewer #2's concerns. The readability has improved and many of points (in my opinion) were adequately addressed. I believe this would be a useful paper, and is suitable for MSB.

Reviewer #3:

(This reviewer did not provide additional comments but rated the study as suitable for publication.)

Point by point response letter for MSB-2024-12687R

Editorial comments

1. Please remove the figures from the manuscript file. Figure legends should remain replaced below the References.

We removed the figures from the manuscript file.

2. Remove the Authors contribution section from the manuscript file.

We removed the authors contribution section.

3. Data availability: Please provide specific URLs for MTBLS10077; MTBLS11393, PRJEB81360 datasets in the data availability statement.

The URLs are there, but the MetaboLights studies are not online yet. The MetaboLights URLs are fixed and will not change when they pass curation and become public.

4. Funding: All the information from the Comments box needs to be included in the list of funders using the "More Funders" option,

We included the information from the box in the list of funders.

5. The synopsis image is too large. Please provide an updated version with the specified dimension(550px width and 400-600 px height, PNG format).

We modified the synopsis image accordingly.

6. Material and Methods" should be renamed to "Methods".

We renamed the section.

7. Appendix: In Table of Content, please add page numbers for the listed items.

The page numbers were added.

8. "Supplementary information" should be removed from manuscript file.

We removed the section from the manuscript file.

9. Section order should be corrected: Title page - Abstract & Keywords - Introduction - Results - Discussion - Methods - Data Availability - Acknowledgements - Disclosure and Competing Interests Statement - References - Figure Legends - Table(s) - Expanded View Figure Legends.

The section order was corrected accordingly.

10. Tables EV1-EV3 and EV6-EV7 should be renamed to Dataset EV1-EV5. Source file names, titles, legends and manuscript callouts all need to be updated to Dataset EV1-EV5; legends should remain as separate tabs/sheets in each Excel file.

Since there were already 5 existing Datasets EV, after renaming the indicated Tables EV1-3, 6, 7 to Datasets EV there are now 10 total Datasets EV. These have been re-numbered according to order of appearance in the text and reuploaded.

11. Please address the following issues in figure legends

- Please note that the exact p values are not provided in the legends of figures 1G, 2A, 3C, D, E F; EV3 E, F, G, I.

Exact p values and the statistical tests used have been added to the legends of Fig. 2A, 3C, D, E F; EV3 E, F, G, I. Based on our judgement, it is not possible to indicate exact p values for 1G. All data points represented in the figure are significant, as described in the legend: "Genes were then further filtered for significance of differential expression ($p\text{-adj.} < 0.05$) and for minimal signal (the gene had to have in one condition at least 50 read counts in both replica)."

- Please indicate the statistical test used for data analysis in the legends of figures 1E, F, G, H; 2B-G.

The statistical tests were included in the legends of respective figures.

- Please indicate what */ **/ ***/ **** represents; if this represents p value(s), please indicate the statistical test used and where appropriate the exact p value in the legend(s) of figure(s) 6D-F; EV5 F, G.

As explained in the Methods, standard statistical testing between flow cytometry populations (>10000 data points per group) would lead to very small p-values for every comparison. Therefore, instead of testing statistical significance, we evaluated effect sizes (Cohen's d) in comparisons of Dox/uninduced samples. These explanations have been added to both Fig. 6 and EV5 legends, along with an explanation of what effect size level corresponds to each asterisk annotation. The exact effect size values have also been provided in the legends.

- Please note that the box plots need to be defined in terms of minima, maxima, centre, bounds of box and whiskers, and percentile in the legends of figures 2A

We added the respective information to the figure legend.

- Please note that information related to n is missing in the legends of figures 2A-C, 6C, D, E, EV5 F

We added the respective information to the figure legend.

- Please note that the error bars are not defined in the legends of figures EV1 H. The legend was modified to define the error bars.

Reviewer's comments

Reviewer #1:

The authors have addressed my original concerns and the paper is acceptable.

Overall I think the authors have made a significant effort to address reviewer #2's concerns. The readability has improved and many of points (in my opinion) were adequately addressed. I believe this would be a useful paper, and is suitable for MSB.

Reviewer #3:

(This reviewer did not provide additional comments but rated the study as suitable for publication.)

We thank the reviewers for their assessment of the revised manuscript and their positive feedback.

31st Mar 2025

Manuscript number: MSB-2024-12687RR

Title: Metabolic mapping of the human solute carrier superfamily

Dear Giulio,

Thank you again for sending us your revised manuscript. We are now satisfied with the modifications made and I am pleased to inform you that your paper has been accepted for publication.

Yours sincerely,

Sincerely,
Jingyi

Jingyi Hou, PhD
Senior Editor
Molecular Systems Biology
